# GRAPH NEURAL BANDITS

## ABSTRACT

Contextual bandits aim to choose the optimal arm with the highest reward out of a set of candidates based on their contextual information, and various bandit algorithms have been applied to personalized recommendation due to their ability of solving the exploitation-exploration dilemma. Motivated by online recommendation scenarios, in this paper, we propose a framework named **Graph Neural Bandits (GNB)** to leverage the collaborative nature among users empowered by graph neural networks (GNNs). Instead of estimating rigid user clusters, we model the "fine-grained" collaborative effects through estimated user graphs in terms of exploitation and exploration individually. Then, to refine the recommendation strategy, we utilize separate GNN-based models on estimated user graphs for exploitation and adaptive exploration. Theoretical analysis and experimental results on multiple real data sets in comparison with state-of-the-art baselines are provided to demonstrate the effectiveness of our proposed framework.

## 1 INTRODUCTION

Contextual bandits are a specific type of multi-armed bandit problem where the additional contextual information (contexts) related to arms are available at each round, and the learner intends to refine its selection strategy based on the received arm contexts and rewards. Various contextual bandit algorithms have been applied in real-world recommendation tasks, such as online content recommendation and advertising (Li et al., 2010; Wu et al., 2016), and clinical trials (Durand et al., 2018; Villar et al., 2015). Meanwhile, collaborative effects among users provide us the opportunity to design better recommender strategies, since the target user's preference can be inferred based on other similar users. Such effects have been studied by many bandit works (Gentile et al., 2014; Li et al., 2019; Gentile et al., 2017; Li et al., 2016; Ban & He, 2021). Different from the conventional collaborative filtering methods (He et al., 2017; Wang et al., 2019), bandit-based approaches focus on more dynamic environments (such as news, short-video platform) and the exploitation-exploration dilemma inherently existed in the decisions of recommendation.

Existing works for clustering of bandits (Gentile et al., 2014; Li et al., 2019; Gentile et al., 2017; Li et al., 2016; Ban & He, 2021; Ban et al., 2022a) have been proposed to model the user correlations (collaborative effects) by clustering users into rigid groups, and assigning each formed group with an estimator to learn the assumed reward functions combined with an Upper Confidence Bound (UCB) strategy for exploration. However, these works only consider the "coarse-grained" user correlations. To be specific, they assume that users from the same group would share identical preferences, i.e., the users from the same group are compelled to make equal contributions to the final decision (arm selection) with regard to the target user. Such formulation of user correlations ("coarse-grained" collaborative effects), evidently fails to comply with real-world application scenarios, since users within the same group tend to have similar but subtly different preferences instead of sharing completely identical tastes. Therefore, given a target user, it is more practical to assume that the rest of the users would impose different levels of (collaborative) effects on this user.

Motivated by aforementioned limitations of existing works, in this paper, we propose a novel framework, named **Graph Neural Bandits (GNB)**, to formulate the "fine-grained" collaborative effects, where the correlation of each user pair is preserved by user graphs. Given a target user, other users are allowed to make different contributions to the final decision based on the strength of their correlation to the target user, which therefore corresponds to the "fine-grained" collaborative effects. In particular, in GNB, we propose a novel approach to construct two kinds of user graphs with distinct purposes, called "user exploitation graphs" and "user exploration graphs". Then, we apply

two separate graph neural network (GNN) models on these two kinds of user graphs, to incorporate the collaborative effects for both exploitation and exploration purposes in the final decision-making process. Our main contributions can be summarized as follows:

1. Different from existing works that only formulate the "coarse-grained" collaborative effects by neglecting the divergence within user groups, we introduce a new problem setting to model the "fine-grained" user collaborative effects via user graphs. In our setting, the pair-wise user correlations are preserved to contribute differently to the decision-making.
2. We propose a framework named GNB, which has the novel ways to build two kinds of user graphs with two different purposes, i.e., exploitation and adaptive exploration, respectively. Then, GNB utilizes GNN-based models for a refined arm selection strategy by leveraging the user correlations encoded in these two kinds of user graphs.
3. With standard assumptions, we provide the theoretical analysis showing that GNB can achieve the regret upper bound of complexity $\mathcal{O}(\sqrt{T \log(Tn)})$, where $T$ is the number of rounds and $n$ is the number of users. This bound is sharper than the existing related works.
4. Extensive experiments comparing GNB with nine state-of-the-art algorithms are conducted on various real data sets, which demonstrate the effectiveness of our proposed method.

After introducing the problem definition in Section 2, we provide the details of our proposed framework in Section 3. Then, we present the theoretical analysis in Section 4, and the experiments in Section 5. Finally, we conclude the paper in Section 6. Due to page limit, we will leave the review of related works to the Section A in the Appendix.

## 2 GRAPH NEURAL BANDITS: PROBLEM DEFINITION AND NOTATION

Suppose there are a total of $n$ users with the user set $\mathcal{U} = \{1, \cdots, n\}$. At each time step $t \in [T]$, the learner will receive a user $u_t \in \mathcal{U}$ to serve, along with a set of candidate arms $\mathcal{X}_t = \{\boldsymbol{x}_{i,t}\}_{i \in [a]}$ for selection. The cardinality of this arm set is $|\mathcal{X}_t| = a$, and each arm is described by a $d$-dimensional context vector $\boldsymbol{x}_{i,t} \in \mathbb{R}^d$ with $\|\boldsymbol{x}_{i,t}\|_2 = 1$. Meanwhile, each arm $\boldsymbol{x}_{i,t} \in \mathcal{X}_t$ is associated with a reward $r_{i,t}$. As the user correlation is one important factor in determining the reward, we define the following reward function:

$$r_{i,t} = h(\boldsymbol{x}_{i,t}, u_t, \mathcal{G}_{i,t}^{(1),*}) + \epsilon_{i,t} \tag{1}$$

where $h(\cdot)$ is the unknown reward mapping function, and $\epsilon_{i,t}$ stands for some zero-mean noise such that $\mathbb{E}[r_{i,t}] = h(\boldsymbol{x}_{i,t}, u_t, \mathcal{G}_{i,t}^{(1),*})$. Here, we have $\mathcal{G}_{i,t}^{(1),*} = (\mathcal{U}, E, W_{i,t}^{(1),*})$ being the **unknown** user graph induced by arm $\boldsymbol{x}_{i,t}$, which encodes the "fine-grained" user correlations in terms of **expected rewards**. In graph $\mathcal{G}_{i,t}^{(1),*}$, each user $u \in \mathcal{U}$ corresponds to a graph node; meanwhile, $E = \{e(u, u')\}_{\forall u,u' \in \mathcal{U}}$ refers to the set of edges, and the set $W_{i,t}^{(1),*} = \{w_{i,t}^{(1),*}(u, u')\}_{\forall u,u' \in \mathcal{U}}$ stores the weights for each edge from $E$. Under real-world application scenarios, users sharing the same preference for certain arms (e.g., sports news) may have distinct tastes over other arms (e.g., political news). Thus, we allow each arm $\boldsymbol{x}_{i,t} \in \mathcal{X}_t$ to induce different user collaborations $\mathcal{G}_{i,t}^{(1),*}$.

Then, motivated by various real applications (e.g., online recommendation with normalized ratings), we consider $r_{i,t}$ to be bounded $r_{i,t} \in [0, 1]$ in this paper, which is standard in existing works (e.g., Gentile et al. (2014; 2017); Ban & He (2021); Ban et al. (2022a)). Note that as long as $r_{i,t} \in [0, 1]$, we do not impose any distribution assumption (e.g., sub-Gaussian distribution) on noise term $\epsilon_{i,t}$.

**Comparison with Existing Problem Definitions.** The problem definition of existing user clustering works (e.g., Gentile et al. (2014); Li et al. (2019); Gentile et al. (2017); Ban & He (2021); Ban et al. (2022a)) only can formulate "coarse-grained" user correlations. In their settings, given a user group $\mathcal{N} \subseteq \mathcal{U}$, all the users in $\mathcal{N}$ are forced to share the same reward function given an arm $\boldsymbol{x}_{i,t}$, i.e., $\mathbb{E}[r_{i,t} \mid u, \boldsymbol{x}_{i,t}] = h_{\mathcal{N}}(\boldsymbol{x}_{i,t}), \forall u \in \mathcal{N}$. In contrast, our definition of the reward function enables us to model the pair-wise fine-grained user correlations by introducing another two important factors $u$ and $\mathcal{G}_{i,t}^{(1),*}$. With our formulation, each user here is allowed to produce different rewards facing the same arm, i.e., $\mathbb{E}[r_{i,t} \mid u, \boldsymbol{x}_{i,t}] = h(\boldsymbol{x}_{i,t}, u, \mathcal{G}_{i,t}^{(1),*}), \forall u \in \mathcal{N}$. Here, with different users $u$, the corresponding expected reward $h(\boldsymbol{x}_{i,t}, u, \mathcal{G}_{i,t}^{(1),*})$ can be different. Therefore, our definition of the reward function is more generic, and it can also readily generalize to existing user clustering algorithms (with "coarse-grained" user correlations) by allowing each single user group to form an isolated sub-graph in $\mathcal{G}_{i,t}^{(1),*}$ with no connections across different sub-graphs.

To bridge user collaborative effects with user preferences (rewards), we consider the following constrain for the reward function in **Eq.** 1. The intuition is that for any two users with comparable user correlations, they would share similar tastes over the items with a high probability. For arm $\boldsymbol{x}_{i,t}$, we consider the difference of expected rewards between any two users $u, u' \in \mathcal{U}$ to be governed by

$$|h(\boldsymbol{x}_{i,t}, u, \mathcal{G}_{i,t}^{(1),*}) - h(\boldsymbol{x}_{i,t}, u', \mathcal{G}_{i,t}^{(1),*})| \leq \Psi(\mathcal{G}_{i,t}^{(1),*}[u,:], \mathcal{G}_{i,t}^{(1),*}[u',:]) \tag{2}$$

where $\mathcal{G}_{i,t}^{(1),*}[u,:]$ stands for the adjacency matrix row of $\mathcal{G}_{i,t}^{(1),*}$ that corresponds to user (node) $u$, and $\Psi : \mathbb{R}^n \times \mathbb{R}^n \mapsto \mathbb{R}$ denotes an unknown mapping function. The reward function definition (**Eq.** 1) and the constraint (**Eq.** 2) motivate us to design the GNB framework, to be introduced in Section 3.

Since the true user graph $\mathcal{G}_{i,t}^{(1),*}$ in **Eq.** 1 reflects user correlations in terms of the expected reward, it is exactly referring to the user exploitation correlation, where users with high correlations tend to have similar expected rewards (**Eq.** 2). Then, we proceed to give the formulation of $\mathcal{G}_{i,t}^{(1),*}$ below.

**Definition 1** (User Correlation for Exploitation). *In round $t$, for any two users $u, u' \in \mathcal{U}$, their exploitation correlation score $w_{i,t}^{(1),*}(u, u')$ w.r.t. a candidate arm $\boldsymbol{x}_{i,t} \in \mathcal{X}_t$ is defined as*

$$w_{i,t}^{(1),*}(u, u') = \Psi^{(1)}\big(\mathbb{E}[r_{i,t}|u, \boldsymbol{x}_{i,t}], \ \mathbb{E}[r_{i,t}|u', \boldsymbol{x}_{i,t}]\big)$$

*where $\mathbb{E}[r_{i,t}|u, \boldsymbol{x}_{i,t}], i \in [a]$ is the expected reward in terms of the user-arm pair $(u, \boldsymbol{x}_{i,t})$. Given two users $u, u' \in \mathcal{U}$, the function $\Psi^{(1)} : \mathbb{R} \times \mathbb{R} \mapsto \mathbb{R}$ maps from their expected rewards $\mathbb{E}[r_{i,t}|u, \boldsymbol{x}_{i,t}]$ to their user exploitation score $w_{i,t}^{(1),*}(u, u')$.*

Given an arm $\boldsymbol{x}_{i,t} \in \mathcal{X}_t$, the user correlation for exploitation measures the user preference (i.e., expected reward) correlation between two users $u, u' \in \mathcal{U}$, and the corresponding exploitation score $w_{i,t}^{(1),*}(u, u')$ refers to the edge weight between these two users (nodes) $u, u'$ in exploitation graph $\mathcal{G}_{i,t}^{(1),*}$. In this paper, we consider the mapping functions $\Psi^{(1)}$ as the prior knowledge, and it can be functions such as the radial basis function (RBF) kernel or normalized absolute difference in practice.

**Modeling User Exploration Correlations with the User Exploration Graph $\mathcal{G}_{i,t}^{(2),*}$.** In order to deal with the exploration-exploitation dilemma, for each candidate arm $\boldsymbol{x}_{i,t} \in \mathcal{X}_t$, we propose to formulate the user exploration graph $\mathcal{G}_{i,t}^{(2),*} = (\mathcal{U}, E, W_{i,t}^{(2),*})$ in order to model the user correlations in terms of the uncertainty of the reward estimation model, which are formulated by the set of edge weights $W_{i,t}^{(2),*} = \{w_{i,t}^{(2),*}(u, u')\}_{\forall u, u' \in \mathcal{U}}$. Inspired by Ban et al. (2022b), before defining the second kind of user correlation (i.e., user exploration correlation), we first introduce the definition of expected potential gain for reward estimation, which measures the prediction uncertainty of reward estimators. Note that our formulation is distinct from Ban et al. (2022b), since they only focus on the single-bandit setting with no user collaborations, and all the users will be treated identically.

**Definition 2** (Expected Potential Gain). *Given user $u \in \mathcal{U}$ at time step $t$, given a candidate arm $\boldsymbol{x}_{i,t} \in \mathcal{X}_t, i \in [a]$ and a reward estimation function $f_u(\cdot)$ corresponding to user $u$, the expected potential gain for the reward estimation $f_u(\boldsymbol{x}_{i,t})$ is defined as $\mathbb{E}[r_{i,t}|u, \boldsymbol{x}_{i,t}] - f_u(\boldsymbol{x}_{i,t})$.*

Here, the potential gain for reward estimation essentially formulates the uncertainty of model $f_u(\cdot)$ by measuring the difference between the expected reward $\mathbb{E}[r_{i,t}|u, \boldsymbol{x}_{i,t}]$ and the prediction $f_u(\boldsymbol{x}_{i,t})$. Next, we proceed to introduce the second kind of user correlation, i.e., user exploration correlation.

**Definition 3** (User Correlation for Exploration). *In round $t$, given two users $u, u' \in \mathcal{U}$ and an arm $\boldsymbol{x}_{i,t} \in \mathcal{X}_t$, their underlying exploration correlation score $w_{i,t}^{(2),*}(u, u')$ is*

$$w_{i,t}^{(2),*}(u, u') = \Psi^{(2)}\big(\mathbb{E}[r_{i,t}|u, \boldsymbol{x}_{i,t}] - f_u(\boldsymbol{x}_{i,t}), \ \mathbb{E}[r_{i,t}|u', \boldsymbol{x}_{i,t}] - f_{u'}(\boldsymbol{x}_{i,t})\big)$$

*with $\mathbb{E}[r_{i,t}|u, \boldsymbol{x}_{i,t}] - f_u(\boldsymbol{x}_{i,t}), i \in [a]$ being the potential gain for the user-arm pair $(u, \boldsymbol{x}_{i,t})$. Here, $f_u(\cdot)$ is the reward estimation function specified to user $u$, and $\Psi^{(2)} : \mathbb{R} \times \mathbb{R} \mapsto \mathbb{R}$ is the mapping from user potential gains $\mathbb{E}[r_{i,t}|u, \boldsymbol{x}_{i,t}] - f_u(\boldsymbol{x}_{i,t})$ to their exploration correlation score.*

For the arm $\boldsymbol{x}_{i,t}$ and two users $u, u' \in \mathcal{U}$, the user exploration correlation score $w_{i,t}^{(2),*}(u, u')$ refers to the correlation of prediction uncertainty between two user-specific functions $f_u(\cdot)$ and $f_{u'}(\cdot)$. Then,

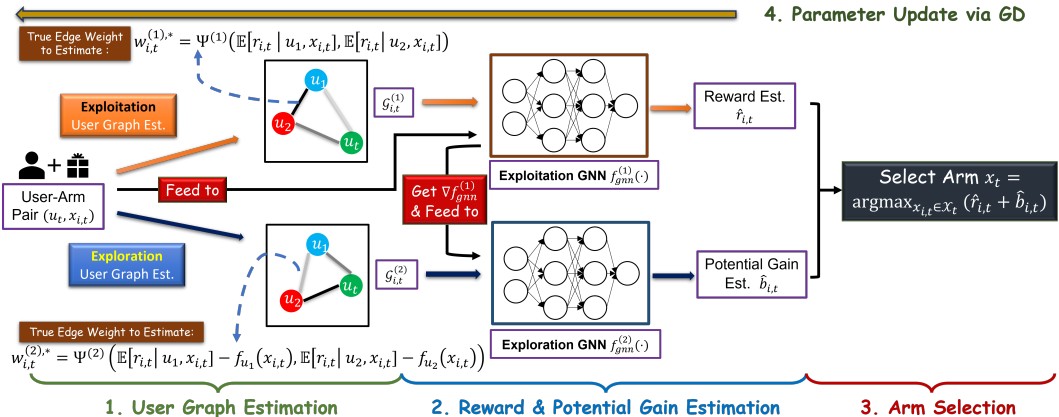

Figure 1: Workflow of the proposed Graph Neural Bandits (GNB) framework.

the exploration score $w_{i,t}^{(2),*}(u, u')$ will be considered as the edge weight between these two nodes (users) $u, u'$ in the true user exploration graph $\mathcal{G}_{i,t}^{(2),*}$. Analogous to previous $\Psi^{(1)}$, we also consider the mapping functions $\Psi^{(2)}$ is given as the prior knowledge. Intuitively, when the exploration score $w_{i,t}^{(2),*}(u, u')$ is high, we can apply similar exploration strategies for both users $u, u'$. For example, given arm $\boldsymbol{x}_{i,t}$, if the reward estimation error (i.e., prediction uncertainty) is large for both $u$ and $u'$, we may want to explore these two user-arm pairs $(u, \boldsymbol{x}_{i,t}), (u', \boldsymbol{x}_{i,t})$ more for additional knowledge.

**Learning Objective.** For the received user $u_t$ in round $t$, the learner is expected to recommend an arm $x_t \in \mathcal{X}_t$ (with reward $r_t$) in order to minimize the cumulative pseudo-regret $R(T) = \mathbb{E}[\sum_{t=1}^{T}(r_t^* - r_t)]$ where $r_t^*$ is the reward for the optimal arm $\mathbb{E}[r_t^*|u_t, \mathcal{X}_t] = \max_{\boldsymbol{x}_{i,t} \in \mathcal{X}_t} h(\boldsymbol{x}_{i,t}, u_t, \mathcal{G}_{i,t}^{(1),*})$.

**Notation.** Denoting $\mathcal{T}_{u,t} \subseteq [t]$ as the collection of time steps that user $u \in \mathcal{U}$ is served up to round $t$, we use $\mathcal{P}_{u,t} = \{(\boldsymbol{x}_\tau, r_\tau)\}_{\tau \in \mathcal{T}_{u,t}}$ to represent the collection of received arm-reward pairs associated with user $u$, and $T_{u,t} = |\mathcal{T}_{u,t}|$ refers to the number of rounds that user $u$ has been served. Here, $\boldsymbol{x}_\tau \in \mathcal{A}_\tau, r_\tau \in \mathbb{R}$ separately refer to the chosen arm and actual received reward in round $\tau \in \mathcal{T}_{u,t}$. Similarly, we use $\mathcal{P}_t = \{(\boldsymbol{x}_\tau, r_\tau)\}_{\tau \in [t]}$ to denote all the past records (i.e., arm-reward pairs), up to round $t$. For any graph $\mathcal{G}$, we denote $\boldsymbol{A} \in \mathbb{R}^{n \times n}$ as its adjacency matrix (with added self-loops), and $\boldsymbol{D} \in \mathbb{R}^{n \times n}$ as its degree matrix. Then, we will introduce our proposed solution, the GNB framework.

## 3 GRAPH NEURAL BANDITS: PROPOSED FRAMEWORK

The workflow of our proposed GNB framework is illustrated by Figure 1, and it consists of four major components: (1) estimating the user exploitation graph $\mathcal{G}^{(1),*}$, denoted by $\mathcal{G}^{(1)}$, and user exploration graph $\mathcal{G}^{(2),*}$, denoted by $\mathcal{G}^{(2)}$ to model the user correlations in terms of exploitation and exploration respectively; (2) applying GNN models $f_{gnn}^{(1)}(\cdot), f_{gnn}^{(2)}(\cdot)$ on the estimated user graphs $\mathcal{G}^{(1)}$ and $\mathcal{G}^{(2)}$, to collaboratively derive the estimated reward for exploitation, and potential gain for exploration; (3) selecting the arm $x_t$ based on estimated reward and potential gain; and (4) training parameters for GNN models and user neural networks with gradient descent (GD).

### 3.1 USER GRAPH ESTIMATION WITH USER NETWORKS

Based on the definition of unknown true user graphs $\mathcal{G}_{i,t}^{(1),*}, \mathcal{G}_{i,t}^{(2),*}$ w.r.t. arm $\boldsymbol{x}_{i,t} \in \mathcal{X}_t$ (**Definitions 1, 3**), we proceed to derive their estimations $\mathcal{G}_{i,t}^{(1)}, \mathcal{G}_{i,t}^{(2)}, i \in [a]$ with individual user networks $f_u^{(1)}$, $f_u^{(2)}, u \in \mathcal{U}$. With these two kinds of estimated user graphs $\mathcal{G}_{i,t}^{(1)}$ and $\mathcal{G}_{i,t}^{(2)}$, we can thus model the user behaviors under the exploitation setting and the exploration setting separately. Due to page limit, pseudo-code summarizing the workflow is presented in **Alg. 2** in Section D of the Appendix.

**User Exploitation Network $f_u^{(1)}$.** For each user $u \in \mathcal{U}$, we propose to apply an exploitation network $f_u^{(1)}(\cdot)$ to learn user $u$'s preference for arm $\boldsymbol{x}_{i,t}$, i.e., $\mathbb{E}[r_{i,t}|u, \boldsymbol{x}_{i,t}]$. Here, $f_u^{(1)}(\cdot)$ will be trained on the past records (arm contexts and rewards) $\mathcal{P}_{u,t}$ from user $u$, and the loss function will be the quadratic loss between the predicted reward and the actual reward. Following the **Definition 1**, we will then be able to construct the exploitation graph $\mathcal{G}_{i,t}^{(1)}$ by estimating the user exploitation correlation

based on user preferences. In $\mathcal{G}_{i,t}^{(1)}$, we consider the edge weight between two user nodes $u, u'$ to be $w_{i,t}^{(1)}(u, u') = \Psi^{(1)}\big(f_u^{(1)}(\boldsymbol{x}_{i,t}), f_{u'}^{(1)}(\boldsymbol{x}_{i,t})\big)$, where $\Psi^{(1)}(\cdot, \cdot)$ is the mapping function mentioned in **Definition** 1 (line 11, **Alg.** 2).

**User Exploration Network** $f_u^{(2)}$. To estimate the potential gain (i.e., the uncertainty for the reward estimation) $\mathbb{E}[r|u, \boldsymbol{x}_{i,t}] - f_u^{(1)}(\boldsymbol{x}_{i,t})$, we adopt an additional user exploration network $f_u^{(2)}(\cdot)$ inspired by Ban et al. (2022b). As the confidence interval of reward estimation can be expressed as a function of network gradients (Zhou et al., 2020; Qi et al., 2022), we apply $f_u^{(2)}(\cdot)$ to directly learn the confidence bound with the gradient of $f_u^{(1)}(\cdot)$. Here, the input of $f_u^{(2)}(\cdot)$ is the network gradient of $f_u^{(1)}(\cdot)$ given arm $\boldsymbol{x}_{i,t}$, denoted as $\nabla f_u^{(1)}(\boldsymbol{x}_{i,t})$. For the training process, $f_u^{(2)}(\cdot)$ will be trained with the past gradients of $f_u^{(1)}$, i.e., $\{\nabla f_u^{(1)}(\boldsymbol{x}_\tau)\}_{\tau \in \mathcal{T}_{u,t}}$ as inputs; and the uncertainty of reward prediction $\{r_\tau - f_u^{(1)}(\boldsymbol{x}_\tau)\}_{\tau \in \mathcal{T}_{u,t}}$ will be the labels. Analogously, for the estimated user exploration graph $\mathcal{G}_{i,t}^{(2)}$ and given two user nodes $u, u'$, we let their edge weight be $w_{i,t}^{(2)}(u, u') = \Psi^{(2)}\bigg(f_u^{(2)}\big(\nabla f_u^{(1)}(\boldsymbol{x}_{i,t})\big), f_{u'}^{(2)}\big(\nabla f_{u'}^{(1)}(\boldsymbol{x}_{i,t})\big)\bigg)$, where $\nabla f_u^{(1)}(\boldsymbol{x}_{i,t})$ stands for the gradient of $f_u^{(1)}(\cdot)$ given arm $\boldsymbol{x}_{i,t}$ as the input (line 12, **Alg.** 2), and $\Psi^{(2)}(\cdot, \cdot)$ is the mapping function in **Definition** 3.

**Network Architecture.** For the theoretical analysis and experiments, we apply separate $L$-layer ($L \geq 2$) fully-connected (FC) networks for both kinds of user networks. Their weight matrix entries for the first $L - 1$ layers are drawn from the Gaussian distribution $N(0, 2/m)$. The entries of the last layer ($L$-th layer) are sampled from $N(0, 1/m)$. Complementary details are in Appendix Section C.

## 3.2 Exploitation and Exploration with User Graphs

With two kinds of estimated user graphs encoding user correlations, we apply two GNN models to separately estimate arm rewards and potential gains for a refined arm selection strategy, which enables us to utilize the **past records from all the users** compared with single-bandit algorithms (i.e., methods with no user collaboration).

### 3.2.1 Architecture of GNN Models

In round $t$, with user exploitation graph $\mathcal{G}_{i,t}^{(1)}$ for each arm $\boldsymbol{x}_{i,t} \in \mathcal{X}_t$, we apply the exploitation GNN model $f_{gnn}^{(1)}(\boldsymbol{x}_{i,t}, \mathcal{G}_{i,t}^{(1)}; \Theta_{gnn}^{(1)})$ to collaboratively estimate the arm reward $\hat{r}_{i,t}$ for the received user $u_t \in \mathcal{U}$. We start from learning the aggregated hidden representation over the user graph, denoted as

$$\boldsymbol{H}_{agg} = \sigma\big((\boldsymbol{S}_{i,t}^{(1)})^k \cdot (\boldsymbol{X}_{i,t}\boldsymbol{\Theta}_{agg}^{(1)})\big) \in \mathbb{R}^{n \times m} \tag{3}$$

where $\boldsymbol{S}_{i,t}^{(1)} = (\boldsymbol{D}_{i,t}^{(1)})^{-\frac{1}{2}} \boldsymbol{A}_{i,t}(\boldsymbol{D}_{i,t}^{(1)})^{-\frac{1}{2}}$ is the symmetrically normalized adjacency matrix of $\mathcal{G}_{i,t}^{(1)}$, and $\sigma$ represents the ReLU activation function. With $m$ being the network width, we have $\boldsymbol{\Theta}_{agg}^{(1)} \in \mathbb{R}^{nd \times m}$ as the trainable weight matrix. After propagating the information for $k$ hops over the user graph, each row of $\boldsymbol{H}_{agg}$ corresponds to the aggregated $m$-dimensional hidden representation for one specific user-arm pair $(u, \boldsymbol{x}_{i,t}), u \in \mathcal{U}$. In this way, the propagation of multi-hop information can provide a global perspective over the users, since it also involves the neighborhood information of users' neighbors (Zhou et al., 2004). Here in **Eq.** 3, the embedding matrix $\boldsymbol{X}_{i,t}$ for the arm $\boldsymbol{x}_{i,t} \in \mathcal{X}_t, i \in [a]$ is defined as

$$\boldsymbol{X}_{i,t} = \begin{pmatrix} \boldsymbol{x}_{i,t}^{\mathsf{T}} & \boldsymbol{0} & \cdots & \boldsymbol{0} \\ \boldsymbol{0} & \boldsymbol{x}_{i,t}^{\mathsf{T}} & \cdots & \boldsymbol{0} \\ \vdots & & \ddots & \vdots \\ \boldsymbol{0} & \boldsymbol{0} & \cdots & \boldsymbol{x}_{i,t}^{\mathsf{T}} \end{pmatrix} \in \mathbb{R}^{n \times nd} \tag{4}$$

to partition the weight matrix $\boldsymbol{\Theta}_{gnn}^{(1)}$ for different users. In this way, it is designed to generate individual $m$-dimensional representations w.r.t. each user-arm pair $(u, \boldsymbol{x}_{i,t}), u \in \mathcal{U}$, which correspond to the rows of the matrix multiplication $(\boldsymbol{X}_{i,t}\boldsymbol{\Theta}_{agg}^{(1)}) \in \mathbb{R}^{n \times m}$.

Then, with $\boldsymbol{H}_0 = \boldsymbol{H}_{agg}$, we feed aggregated representations to the $L$-layer ($L \geq 2$) FC network as

$$\boldsymbol{H}_l = \sigma(\boldsymbol{H}_{l-1} \cdot \boldsymbol{\Theta}_l^{(1)}) \in \mathbb{R}^{n \times m}, \ l \in [L-1], \qquad \hat{\boldsymbol{r}}_{all}(\boldsymbol{x}_{i,t}) = \boldsymbol{H}_{L-1} \cdot \boldsymbol{\Theta}_L^{(1)} \in \mathbb{R}^n \tag{5}$$

where $\widehat{r}_{all}(\boldsymbol{x}_{i,t}) \in \mathbb{R}^n$ represents the reward estimation for all the users in $\mathcal{U}$, given the arm $\boldsymbol{x}_{i,t}$. Received user $u_t$ in round $t$, the reward estimation for the user-arm pair $(u_t, \boldsymbol{x}_{i,t})$ would be the corresponding element in $\widehat{r}_{all}$ (line 8, **Alg.** 1), represented by:

$$\hat{r}_{i,t} = f_{gnn}^{(1)}(\boldsymbol{x}_{i,t}, \mathcal{G}_{i,t}^{(1)}; \boldsymbol{\Theta}_{gnn}^{(1)}) = [\widehat{r}_{all}(\boldsymbol{x}_{i,t})]_{u_t}. \tag{6}$$

For the FC network, the weight matrices for the first $L-1$ layers are $\boldsymbol{\Theta}_l \in \mathbb{R}^{m \times m}, l \in [1, \cdots, L-1]$, and for the $L$-th layer, we have $\boldsymbol{\Theta}_L \in \mathbb{R}^m$. Here, we use $\boldsymbol{\Theta}_{gnn}^{(1)} = [\text{vec}(\boldsymbol{\Theta}_{agg}^{(1)})^\intercal, \text{vec}(\boldsymbol{\Theta}_1^{(1)})^\intercal, \ldots, \text{vec}(\boldsymbol{\Theta}_L^{(1)})^\intercal]^\intercal \in \mathbb{R}^p$ to represent the trainable parameters of the GNN exploitation model. The exploitation GNN model $f_{gnn}^{(1)}(\cdot)$ will be trained with GD based on all the received records $\mathcal{P}_t$. Then we apply the quadratic loss function between the reward prediction $\{f_{gnn}^{(1)}(\boldsymbol{x}_\tau, \mathcal{G}_\tau^{(1)}; \boldsymbol{\Theta}_{gnn}^{(1)})\}_{\tau \in [t]}$ of chosen arms $\boldsymbol{x}_\tau$, and the actual received rewards $\{r_\tau\}_{\tau \in [t]}$.

**Connection with the Reward Function Definition (Eq. 1) and Constraint (Eq. 2).** It is known that when width $m$ is large enough, the FC network is naturally Lipschitz continuous with respect to the input (Allen-Zhu et al., 2019). In our case, with aggregated hidden representations $\boldsymbol{H}_{agg}$ being the input to the FC network (**Eq.** 5), we will have the difference of reward estimations $\hat{r}_{i,t}$ bounded by the distance of rows in matrix $\boldsymbol{H}_{agg}$ (i.e., aggregated hidden representations). Therefore, given arm $\boldsymbol{x}_{i,t} \in \mathcal{X}_t$ and two users $u_i, u_j \in \mathcal{U}$, the difference of their estimated rewards $|[\widehat{r}_{all}(\boldsymbol{x}_{i,t})]_{u_i} - [\widehat{r}_{all}(\boldsymbol{x}_{i,t})]_{u_j}|$ can be bounded by the distance of their estimated correlation vectors (i.e, the corresponding rows in $\boldsymbol{S}_{i,t}$). This matches the reward function definition and the constraint presented in **Eq.** 1-2.

**Exploration GNN Model.** To achieve adaptive exploration with user collaborations, we apply a second GNN model $f_{gnn}^{(2)}(\nabla[f_{gnn}^{(1)}]_{i,t}, \mathcal{G}_{i,t}^{(2)}; \boldsymbol{\Theta}_{gnn}^{(2)})$ to evaluate the potential gain $\hat{b}_{i,t}$ of the reward estimation $f_{gnn}^{(1)}(\boldsymbol{x}_{i,t}, \mathcal{G}_{i,t}^{(1)}; \boldsymbol{\Theta}_{gnn}^{(1)})$ (line 8, **Alg.** 1). Here, the input is the user exploration graph $\mathcal{G}_{i,t}^{(2)}$, and the corresponding input graph signal is the gradient of the exploitation GNN model $\nabla[f_{gnn}^{(1)}]_{i,t} = \nabla_{\boldsymbol{\Theta}_{gnn}^{(1)}} f_{gnn}^{(1)}(\boldsymbol{x}_{i,t}, \mathcal{G}_{i,t}^{(1)}; \boldsymbol{\Theta}_{gnn}^{(1)})$. Analogous to $f_{gnn}^{(1)}(\cdot)$, the architecture of $f_{gnn}^{(2)}(\cdot)$ can also be represented by **Eq.** 3-**Eq.** 6. Note that while $f_{gnn}^{(1)}(\cdot), f_{gnn}^{(2)}(\cdot)$ have the same network width and number of layers, the dimensionality of weight matrices $\boldsymbol{\Theta}_{agg}^{(1)} \in \mathbb{R}^{nd \times m}, \boldsymbol{\Theta}_{agg}^{(2)} \in \mathbb{R}^{np \times m}$ is different. Similarly, the exploration GNN model will be trained with GD. With the quadratic loss function, we aim to minimize the difference between estimated potential gains $\{f_{gnn}^{(2)}(\nabla[f_{gnn}^{(1)}]_\tau, \mathcal{G}_\tau^{(2)}; \boldsymbol{\Theta}_{gnn}^{(2)})\}_{\tau \in [t]}$ and the actual ones $\{r_\tau - f_{gnn}^{(1)}(\boldsymbol{x}_\tau, \mathcal{G}_\tau^{(1)}; \boldsymbol{\Theta}_{gnn}^{(1)})\}_{\tau \in [t]}$.

Instead of calculating non-negative UCB intervals (upward exploration only) as in existing works (e.g., Gentile et al. (2014); Ban et al. (2022a)), the exploration GNN model $f_{gnn}^{(2)}(\cdot)$ leverages both gradient information from the exploitation GNN model $f_{gnn}^{(1)}(\cdot)$ and the user exploration correlations (i.e., $\mathcal{G}_{i,t}^{(2)}$) to achieve adaptive exploration (downward and upward exploration).

**Remark 3.1** (Reducing Input Complexity). The input of $f_{gnn}^{(2)}(\cdot)$ is the gradient $\nabla_{\boldsymbol{\Theta}} f_{gnn}^{(1)}(\boldsymbol{x})$ given the arm $\boldsymbol{x}$, and its dimensionality is naturally $p = (nd \times m) + (L-1) \times m^2 + m$, which can be large when increasing the network width $m$ and depth $L$. Inspired by Convolutional Neural Networks (CNNs), e.g., Radenović et al. (2018), we apply the average pooling to calculate the approximation for the original gradient vector in practice. In this way, we can save the running time for large matrix multiplications, and reduce the space complexity at the same time. Note this approach is also compatible with user networks in Subsection 3.1. To prove its effectiveness, we will apply this method on GNB for all the experiments in Section 5.

**Remark 3.2** (Working with Large Systems). When facing a large number of users, to deal with potentially high computational cost, we can apply approximated user neighborhoods to reduce the running time of GNB. Given user graphs $\mathcal{G}_{i,t}^{(1)}, \mathcal{G}_{i,t}^{(2)}$ in terms of arm $\boldsymbol{x}_{i,t}$, we derive approximated user neighborhoods $\tilde{\mathcal{N}}^{(1)}(u_t), \tilde{\mathcal{N}}^{(2)}(u_t) \subset \mathcal{U}$ for target user $u_t$, with the size $|\tilde{\mathcal{N}}^{(1)}(u_t)| = |\tilde{\mathcal{N}}^{(2)}(u_t)| = \tilde{n}$, where $\tilde{n} << n$. For instance, we can choose a subset of $\tilde{n}$ representative users (e.g., users who always post high quality reviews in e-commerce platforms) to form $\tilde{\mathcal{N}}^{(1)}(u_t), \tilde{\mathcal{N}}^{(2)}(u_t)$ for the downstream GNN models, which can significantly reduce the computation cost. Related experiments are provided in Subsection 5.3 and Appendix Section B.

---

**ALGORITHM 1:** Graph Neural Bandits (GNB)

---

1 **Input:** Number of rounds $T$, network width $m$, information propagation hops $k$. Functions for edge weight estimation $\Psi^{(1)}(\cdot, \cdot), \Psi^{(2)}(\cdot, \cdot) : \mathbb{R} \times \mathbb{R} \mapsto \mathbb{R}$.

2 **Output:** Arm recommendation $\boldsymbol{x}_t$ for each time step $t$.

3 **Initialization:** Initialize parameter $\boldsymbol{\Theta}_0$ for all models.

4 **for** $t = 1, 2, ..., T$ **do**

5      Receive a user $u_t$ and a set of arm contexts $\mathcal{X}_t = \{\boldsymbol{x}_{i,t}\}_{i \in [a]}$.

6      Construct two kinds of user graphs $\{\mathcal{G}_{i,t}^{(1)}\}_{i \in [a]}, \{\mathcal{G}_{i,t}^{(2)}\}_{i \in [a]}$ for arm set $\mathcal{X}_t$ with **Algorithm** 2.

7      **for** *each arm $\boldsymbol{x}_{i,t} \in \mathcal{X}_t$* **do**

8          Compute reward estimation $\hat{r}_{i,t} = f_{gnn}^{(1)}(\boldsymbol{x}_{i,t}, \ \mathcal{G}_{i,t}^{(1)}; [\boldsymbol{\Theta}_{gnn}^{(1)}]_{t-1})$, and the potential gain

         $\hat{b}_{i,t} = f_{gnn}^{(2)}(\nabla_{\boldsymbol{\Theta}_{gnn}^{(1)}} f_{gnn}^{(1)}(\boldsymbol{x}_{i,t}, \ \mathcal{G}_{i,t}^{(1)}; [\boldsymbol{\Theta}_{gnn}^{(1)}]_{t-1}), \ \mathcal{G}_{i,t}^{(2)}; [\boldsymbol{\Theta}_{gnn}^{(2)}]_{t-1})$.

9      **end**

10      Play arm $\boldsymbol{x}_t = \arg\max_{\boldsymbol{x}_{i,t} \in \mathcal{X}_t} (\hat{r}_{i,t} + \hat{b}_{i,t})$, and observe its true reward $r_t$.

11      Train the user networks $f_u^{(1)}(\cdot; \boldsymbol{\Theta}_u^{(1)}), f_u^{(2)}(\cdot; \boldsymbol{\Theta}_u^{(2)})$ and GNN models $f_{gnn}^{(1)}(\cdot; \boldsymbol{\Theta}_{gnn}^{(1)})$,

     $f_{gnn}^{(2)}(\cdot; \boldsymbol{\Theta}_{gnn}^{(2)})$ with gradient descent, according to **Algorithm** 3.

12 **end**

---

**Weight Matrices Initialization.** For both GNN models $\boldsymbol{\Theta}_{gnn}^{(1)}$ and $\boldsymbol{\Theta}_{gnn}^{(2)}$, the matrix entries of the aggregation weight matrix $\boldsymbol{\Theta}_{agg}$ and the first $L - 1$ FC layers $\{\boldsymbol{\Theta}_1, \dots \boldsymbol{\Theta}_{L-1}\}$ are drawn from the Gaussian distribution $N(0, 2/m)$. Then, for the last layer $\boldsymbol{\Theta}_L$, we draw its entries from $N(0, 1/m)$.

### 3.2.2 ARM SELECTION MECHANISM AND MODEL TRAINING

In round $t$, with the current parameters $[\Theta_{gnn}^{(1)}]_{t-1}, [\Theta_{gnn}^{(2)}]_{t-1}$ for GNN models before training, the selected arm is chosen as $\boldsymbol{x}_t = \arg\max_{\boldsymbol{x}_{i,t} \in \mathcal{X}_t} \left( f_{gnn}^{(1)}(\boldsymbol{x}_{i,t}, \ \mathcal{G}_{i,t}^{(1)}; [\Theta_{gnn}^{(1)}]_{t-1}) + f_{gnn}^{(2)}(\nabla_{\Theta_{gnn}^{(1)}} f_{gnn}^{(1)}(\boldsymbol{x}_{i,t}, \ \mathcal{G}_{i,t}^{(1)}; [\Theta_{gnn}^{(1)}]_{t-1}), \ \mathcal{G}_{i,t}^{(2)}; [\Theta_{gnn}^{(2)}]_{t-1}) \right)$ based on the estimated reward and potential gain (line 10, **Alg.** 1). After receiving the true reward $r_t$, we proceed to update the user networks and GNN models based on GD and quadratic loss function (line 11, **Alg.** 1). Pseudo-code summarizing the training procedure is shown in **Alg.** 3 (Appendix Section D).

## 4 THEORETICAL ANALYSIS

In this section, we present the theoretical analysis for the proposed GNB. Here, we consider each user $u \in \mathcal{U}$ to be evenly served $T/n$ rounds up to time step $T$, i.e., $|\mathcal{T}_{u,t}| = T_{u,t} = T/n$, which is standard in closely related works (e.g., Gentile et al. (2014); Ban & He (2021)). To ensure the neural models are able to efficiently learn the underlying reward mapping, we have the following mild assumption on arm separateness.

**Assumption 4.1** ($\rho$-Separateness of Arms). *After a total of $T$ rounds, for every pair $\boldsymbol{x}_{i,t}, \boldsymbol{x}_{i',t'}$ with $t, t' \in [T]$ and $i, i' \in [a]$, if $(t, i) \neq (t', i')$, we have $\|\boldsymbol{x}_{i,t} - \boldsymbol{x}_{i',t'}\|_2 \geq \rho$ where $0 < \rho \leq \mathcal{O}(\frac{1}{L})$.*

Note that the above assumption is mild, and it has been repeatedly applied in previous works on neural bandits (Ban et al., 2022b) and over-parameterized neural networks (Allen-Zhu et al., 2019). Meanwhile, Assumption 4.2 in Zhou et al. (2020) and Assumption 3.4 from Zhang et al. (2021) also imply that no two arms are the same, and they measure the arm separateness in terms of the minimum eigenvalue $\lambda_0$ (with $\lambda_0 > 0$) of the Neural Tangent Kernel (NTK) (Jacot et al., 2018) matrix, which is comparable with our Euclidean separateness $\rho$. Note that since $L$ can be manually set (e.g., $L = 2$), we can easily satisfy the condition $0 < \rho \leq \mathcal{O}(\frac{1}{L})$ as long as no two arms are identical.

Based on **Definition** 1 and **Definition** 3, given an arm $\boldsymbol{x}_{i,t} \in \mathcal{X}_t$, we have the adjacency matrices $\boldsymbol{A}_{i,t}^{(1),*}$ and $\boldsymbol{A}_{i,t}^{(2),*}$ for the true arm graphs $\mathcal{G}_{i,t}^{(1),*}, \mathcal{G}_{i,t}^{(2),*}$. For the sake of analysis, given any adjacency matrix $\boldsymbol{A}$, we derive the normalized adjacency matrix $\boldsymbol{S}$ by scaling the elements of $\boldsymbol{A}$ with $1/n$. We also set the neighborhood parameter $k = 1$, and define the mapping functions $\Psi^{(1)}(a, b), \Psi^{(2)}(a, b) := \exp(-|a - b|)$ given the inputs $a, b \in \mathbb{R}$. Note that our results can be readily generalized to other mapping functions with the Lipschitz-continuity properties.

We proceed to derive the regret bound for $T$ time steps, denoted as $R(T)$. Here, the following **Theorem** 4.2 offers the cumulative regret bound covering both types of error: (1) the estimation error of user graphs; and (2) the approximation error of neural models. Let $\eta_1, J_1$ be the learning rate and iterations for user networks, and $\eta_2, J_2$ denote the learning rate and iterations for GNN models.

**Theorem 4.2.** *Define $\delta \in (0, 1)$, $0 < \xi_1, \xi_2 \leq \mathcal{O}(1/T)$ and $0 < \rho \leq \mathcal{O}(1/L)$. With the user networks defined in **Eq.** 7 and the GNN models defined in **Eq.** $3-5$ with $L$ FC-layers, let their width $m \geq \Omega\left( Poly(T, L, a, \frac{1}{\rho}) \cdot \log(1/\delta) \right)$. With training process in **Algorithm** 3, set parameters*

$$\eta_1 = \Theta\left( \frac{\rho}{m \cdot Poly(T, n, a, L)} \right), \quad \eta_2 = \Theta\left( \frac{\rho}{m \cdot Poly(T, a, L)} \right),$$
$$J_1 = \Theta\left( \frac{Poly(T, n, a, L)}{\rho \cdot \delta^2} \cdot \log(\frac{1}{\xi_1}) \right), \quad J_2 = \Theta\left( \frac{Poly(T, a, L)}{\rho \cdot \delta^2} \cdot \log(\frac{1}{\xi_2}) \right).$$

*Then, following **Algorithm** 1, **Algorithm** 2 for arm pulling and user group update, with probability at least $1 - \delta$, the $T$-round pseudo-regret $R(T)$ of GNB could be bounded by*

$$R(T) \leq \sqrt{T} \cdot \mathcal{O}(L) + \sqrt{T} \cdot \mathcal{O}(L^3) + \sqrt{T} \cdot \mathcal{O}(L^2) \cdot \sqrt{\log(\frac{Tn \cdot a}{\delta})} + \mathcal{O}(L^2) + \mathcal{O}(1).$$

The proof of **Theorem** 4.2 and the full regret bound are presented in the Appendix.

**Remark 4.3** (Reducing $\sqrt{n}$ to $\sqrt{\log(n)}$)**.** While our $\mathcal{O}(\sqrt{T \log(T)})$ bound matches theoretical bound of state-of-the-art EE-Net (Ban et al., 2022b), EE-Net only considers the single-bandit setting with no collaboration among users. Compared with Meta-Ban (Ban et al., 2022a), we provide the theoretical analysis from a new perspective regarding the fine-grained user collaborative effect and GNNs. In particular, compared with existing user clustering works (e.g., Ban et al. (2022a); Gentile et al. (2014); Li et al. (2019); Ban & He (2021)) imposing the additional $\sqrt{n}$ (where $n$ is the number of users) factor to incorporate user collaborative effects, our GNB only end up with the $\sqrt{\log(n)}$ term by adopting GNN models for user collaboration, which is sharper than existing works.

**Remark 4.4** (Removing i.i.d. Assumption)**.** Compared with existing clustering of bandits algorithms (e.g., Gentile et al. (2014); Li et al. (2019); Gentile et al. (2017); Ban et al. (2022a)) and the single-bandit algorithm EE-Net (Ban et al., 2022b), we avoid making the i.i.d. assumption for the arms by applying the martingale-based analysis. For real-world applications, their i.i.d. assumption can be strong since the candidate arm pool is always conditioned on the received records, and candidate arms for a specific round can also come from different distributions.

## 5 EXPERIMENTS

In this section, we evaluate the proposed GNB framework on multiple real data sets against nine state-of-the-art algorithms, including: **CLUB** (Gentile et al., 2014), **SCLUB** (Li et al., 2019), **LOCB** (Ban & He, 2021), **DynUCB** (Nguyen & Lauw, 2014), **COFIBA** (Li et al., 2016), **Neural-UCB-Pool (Neural-Pool)** (Zhou et al., 2020), **Neural-UCB-Ind (Neural-Ind)** (Zhou et al., 2020), **EE-Net** (Ban et al., 2022b), and **Meta-Ban** (Ban et al., 2022a). We will include the descriptions for the baselines in the Appendix Section B.

### 5.1 REAL DATA SETS

**Recommendation Data Sets.** "MovieLens rating dataset" (https://www.grouplens.org/datasets/movielens/20m/) includes reviews from $1.6 \times 10^5$ users towards $6 \times 10^4$ movies. Since the genome-scores of user-specified tags are provided for each movie, we select 10 tags with the highest score variance to generate the movie features $\boldsymbol{v}_i \in \mathbb{R}^d, d = 10$. Here, the user features $\boldsymbol{v}_u \in \mathbb{R}^d, u \in \mathcal{U}$ are obtained through singular value decomposition (SVD) of the rating matrix. Here, we use K-means to divide users into 50 groups based on $\boldsymbol{v}_u$, and the group information is unknown to models. In each round $t$, a user $u_t$ is drawn from a randomly selected group. For the arm pool $\mathcal{X}_t$ of 10 arms, we randomly choose one bad movie (with two stars or less, out of five) rated by $u_t$ with reward 1, and randomly pick the other 9 good movies with reward 0. For "Yelp" data set (https://www.yelp.com/dataset), we extract ratings and build the rating matrix w.r.t. the top $2,000$ users and top $10,000$ arms with the most reviews. Then, we use SVD to extract a normalized 10-dimensional feature vector for each user and restaurant. Given the rating for a specific

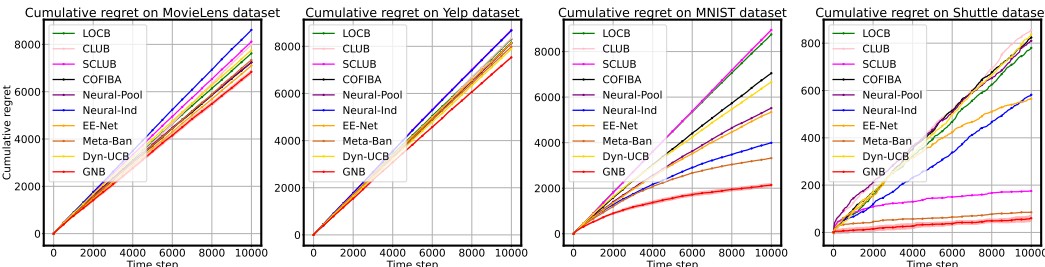

Figure 2: Cumulative regrets on the recommendation and classification data sets.

user-item pair, if the user's rating is greater than three stars (out of five stars), the reward is set to $1$; otherwise, the reward is $0$. Similarly, we apply K-means clustering to divide users into 50 groups based on user features. In each round $t$, a target $u_t$, is sampled from a randomly selected group. For the arm pool $\mathcal{X}_t$, we randomly choose one good restaurant rated by $u_t$ with reward $1$ and randomly pick the other $9$ bad restaurants with reward $0$.

**Classification Data Sets.** In addition to the two recommendation data sets above, we also perform experiments on two real classification data sets under the recommendation settings, which are "MNIST" (http://yann.lecun.com/exdb/mnist/) and "Shuttle" (https://archive.ics.uci.edu/ml/datasets/Statlog+(Shuttle)) data sets. Similar to previous works (Zhou et al., 2020; Ban et al., 2022a), given a sample $\boldsymbol{x} \in \mathbb{R}^d$, we transform it into $\mathcal{C}$ different arms, as $\boldsymbol{x}_1 = (\boldsymbol{x}, 0, \ldots, 0), \boldsymbol{x}_2 = (0, \boldsymbol{x}, \ldots, 0), \ldots, \boldsymbol{x}_{|\mathcal{C}|} = (0, 0, \ldots, \boldsymbol{x}) \in \mathbb{R}^{d+\mathcal{C}-1}$ where we add $\mathcal{C} - 1$ zero digits as the padding. The received reward $r_t$ will be 1 if we select the arm of the correct class, else the reward will be 0.

## 5.2 EXPERIMENT RESULTS

Figure 2 illustrates the cumulative regret results on the four data sets, our proposed GNB manages to achieve the best performance against all these strong benchmarks. First, since the MovieLens data set involves real arm features unlike the Yelp data set that includes high inherent noise, the performance of different algorithms on the MovieLens data set tends to have larger divergence. Among those regret results, the algorithms with neural architectures (Neural-Pool, EE-Net, Meta-Ban) generally perform better than linear algorithms due to the approximation power of neural networks. However, as Neural-Ind considers no collaboration among users, it performs the worst among all baselines on these two data sets. EE-Net outperforms Neural-Pool thanks to its adaptive exploration strategy. For classification data sets, Meta-Ban performs better than the other baselines by modeling user correlations under the non-linear setting. Different from recommendation data sets, the classification data sets involve more complicated reward mapping functions, and this might lead to the poor performances of linear algorithms. Our proposed GNB consistently outperforms all baselines by modeling fine-grained user correlations and utilizing the adaptive exploration strategy simultaneously. In addition, GNB only takes at most $75\%$ of Meta-Ban's running time to finish the experiments (**Table** 1), since Meta-Ban trains the framework individually for each arm before making predictions.

## 5.3 SUPPLEMENTARY EXPERIMENTS

Due to the page limit, we present additional supplementary experiments in the Appendix Section B, including: (1) experiments on additional data sets; (2) with increasing number of users, experiments demonstrating the effectiveness of applying approximated user neighborhoods (Remark 3.2); (3) experiments showing the potential performance impact on GNB when there exist underlying user clusters; (4) the parameter sensitivity study showing that our adaptive exploration strategy can indeed improve the performance of GNB, and the effects of different hops $k$ for information propagation.

## 6 CONCLUSION

In this paper, we propose a novel framework named GNB to model the fine-grained user collaborative effects. Instead of modeling user correlations through the estimation of rigid user groups, we estimate the user graphs to preserve the pair-wise user correlations for exploitation and exploration separately, and utilize individual GNN-based models to achieve the adaptive exploration. Moreover, under standard assumptions, we also demonstrate the improvement of regret bounds over existing methods from a new perspective of "fine-grained" user collaborative effects and GNNs. Extensive experiments are conducted to show the effectiveness of our proposed framework against strong baselines.

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

## A   RELATED WORKS

In this section, we briefly review the existing works related to our proposed GNB framework. Assuming the reward mapping to be linear, linear upper confidence bound (UCB) algorithms (Chu et al., 2011; Li et al., 2010; Auer et al., 2002; Abbasi-Yadkori et al., 2011) were first proposed to solve the exploitation-exploration dilemma. After kernel-based methods (Valko et al., 2013; Deshmukh et al., 2017) were used to address the non-linear setting where the reward mapping is the kernel-based function, neural algorithms (Zhou et al., 2020; Zhang et al., 2021; Ban et al., 2021) have been proposed to utilize neural networks to estimate the reward function and confidence bound. Meanwhile, AGG-UCB (Qi et al., 2022) adopts GNN to model the arm group correlations. GCN-UCB (Upadhyay et al., 2020) manages to apply the GNN model to embed arm contexts for downstream linear regression, and GNN-PE (Kassraie et al., 2022) utilizes the UCB based on information gains to achieve exploration for classification tasks on graphs. Note that the above neural algorithms with UCB-based exploration strategy all suffer from the space complexity $\mathcal{O}(p^2)$ to store their gigantic gradient matrix, where $p$ is the number of model parameters. This space cost is especially enormous when you increase the quantity of model parameters by adding more network width $m$ and depth $L$. Instead of UCB, EE-Net (Ban et al., 2022b) achieves adaptive exploration by using neural models for estimating prediction uncertainty. Assuming a finite number of arms, (Maillard & Mannor, 2014; Hong et al., 2020) discuss the latent bandits where there exist latent states that affect the reward generation. Nonetheless, all of these works fail to consider the collaboration effects among users under the real-world application scenarios.

In order to model the user correlations, (Wu et al., 2016; Cesa-Bianchi et al., 2013) assume the user social graph is known, and apply an ensemble of linear estimators. Without the prior knowledge of user correlations, CLUB (Gentile et al., 2014) introduces the user clustering problem in contextual bandits with the graph connected components, SCLUB (Li et al., 2019) adopts dynamic user sets and applies set operations to update user clusters, and DynUCB (Nguyen & Lauw, 2014) assigns users to their nearest estimated clusters. Then, CAB (Gentile et al., 2017) studies the arm-specific user clustering, and LOCB (Ban & He, 2021) estimates soft-margin user groups through a random-seed based approach. COFIBA (Li et al., 2016) utilizes user and arm clustering for collaborative filtering. Apart from these linear algorithms, we note a concurrent work Meta-Ban (Ban et al., 2022a), which applies a neural meta-model to adapt to different user groups. However, all algorithms mentioned in this paragraph consider rigid user groups, where users from the same group are treated equally with no internal differentiation.

GNNs (Welling & Kipf, 2017; Chen et al., 2018; Wu et al., 2019; Gasteiger et al., 2019; He et al., 2020; Satorras & Estrach, 2018) are a kind of neural models operating on the graph data, and have been proved effective for various tasks, e.g., community detection (You et al., 2019) and recommender systems (Ying et al., 2018). In this work, we leverage GNNs to learn from user correlations and arm contexts simultaneously.

## B   EXPERIMENT SETTINGS AND SUPPLEMENTARY EXPERIMENTS

### B.1   BASELINES AND EXPERIMENT SETTINGS

The descriptions for our nine baseline methods are:

- **CLUB** (Gentile et al., 2014) regards connected components as user groups out of the estimated user graph, and adopts a UCB-type exploration strategy;
- **SCLUB** (Li et al., 2019) estimates dynamic user sets as user groups, and allows set operations for group updates;
- **LOCB** (Ban & He, 2021) applies soft-clustering among users with random seeds and choose the best user group for reward and confidence bound estimations;
- **DynUCB** (Nguyen & Lauw, 2014) dynamically assigns users to its nearest estimated cluster.
- **COFIBA** (Li et al., 2016) estimates user clustering and arm clustering simultaneously, and ensembles linear estimators for reward and confidence bound estimations;
- **Neural-Pool** adopts one single Neural-UCB (Zhou et al., 2020) model for all the users with UCB-type exploration strategy;

- **Neural-Ind** assigns each user with their own separate Neural-UCB (Zhou et al., 2020) model;

- **EE-Net** (Ban et al., 2022b) achieves adaptive exploration by applying additional neural models for the exploration and decision making;

- **Meta-Ban** (Ban et al., 2022a) utilizes individual neural models for each user's behavior, and applies a meta-model to adapt to estimated user groups.

**Baseline Settings.** For all the UCB-based baselines, we choose theirs exploration parameter with grid search in the range $\{0.01, 0.1, 1\}$ individually. And we set the $L = 2$ for all the deep learning models, and set the network width $m = 100$. The learning rate of all neural algorithms are selected by grid search in range $\{0.0001, 0.001, 0.01\}$. For EE-Net, we follow the default setting in their paper by using a hybrid decision maker, where the estimation is $f_1 + f_2$ for the first 500 time steps, and then we apply an additional neural network for decision making afterwards. For Meta-Ban, we follow the settings in their paper by turning the clustering parameter $\gamma$ through grid search $\{0.1, 0.2, 0.3, 0.4\}$. For GNB, we choose the $k$-hop user neighborhood $k \in \{1, 2, 3\}$ with grid search. Reported results are the average of 5 runs.

## B.2 EXPERIMENTS ON ADDITIONAL DATA SETS

Due to the page limit in the main body and to better compare our GNB with the benchmarks, here, we include the experiments on two additional classification data sets in this subsection. They are: (1) the "Letter" data set with $\mathcal{C} = 26$ different classes (https://archive.ics.uci.edu/ml/datasets/letter+recognition), and (2) the "Pendigits" data set with $\mathcal{C} = 10$ classes (https://archive.ics.uci.edu/ml/datasets/Pen-Based+Recognition+of+Handwritten+Digits), under the recommendation settings. Analogous to settings of the "MNIST" and the "Shuttle" data set, we consider each class to be a user. Given a sample $\boldsymbol{x} \in \mathbb{R}^d$, we transform it into $\mathcal{C}$ arms for different classes similar to previous works (Zhou et al., 2020; Ban et al., 2022a), namely $\boldsymbol{x}_1 = (\boldsymbol{x}, 0, \ldots, 0), \boldsymbol{x}_2 = (0, \boldsymbol{x}, \ldots, 0), \ldots, \boldsymbol{x}_{\mathcal{C}} = (0, 0, \ldots, \boldsymbol{x}) \in \mathbb{R}^{d+\mathcal{C}-1}$ where additional $\mathcal{C} - 1$ zero digits are added as the padding. The reward will be $r = 1$ if we choose the correct arm that represents the sample's true class; otherwise, the reward will be 0.

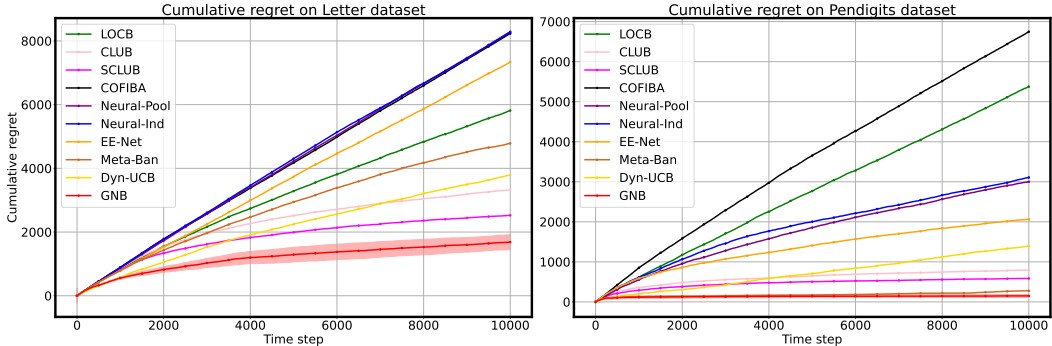

Figure 3: Cumulative regrets on the two additional classification data sets.

The experiment results for these two additional data sets are presented in Figure 3. It is worthwhile to note that EE-Net continues to outperform the two Neural-UCB baselines, which is also another evidence of the effectiveness of the adaptive exploration strategy. On the other hand, our exploration strategy inspired by EE-Net further incorporates the user exploration graphs to exploit the encoded "fine-grained" user collaborative effects. Therefore, analogous to the experiment results in the main body (Figure 2), our proposed GNB framework consistently outperforms the other benchmarks by exploiting and adaptively exploring the "fine-grained" correlations among different classes at the same time.

### B.3 Effects of the Adaptive Exploration and Effects of Information Propagation Hops

In order to demonstrate the necessity of the adaptive exploration strategy, we consider an alternative arm selection approach (different from line 10, **Alg.** 1) at each time step $t$, with the following form:

$$\boldsymbol{x}_t = \arg \max_{\boldsymbol{x}_{i,t} \in \mathcal{X}_t} \Bigg( f_{gnn}^{(1)} \big( \boldsymbol{x}_{i,t}, \, \mathcal{G}_{i,t}^{(1)}; [\Theta_{gnn}^{(1)}]_{t-1} \big)$$

$$+ \alpha \cdot f_{gnn}^{(2)} \big( \nabla_{\Theta_{gnn}^{(1)}} f_{gnn}^{(1)} (\boldsymbol{x}_{i,t}, \, \mathcal{G}_{i,t}^{(1)}; [\Theta_{gnn}^{(1)}]_{t-1}), \, \mathcal{G}_{i,t}^{(2)}; [\Theta_{gnn}^{(2)}]_{t-1} \big) \Bigg)$$

given the candidate arm set $\mathcal{X}_t = \{\boldsymbol{x}_{i,t}\}_{i \in [a]}$ and the model parameters $[\Theta_{gnn}^{(1)}]_{t-1}, [\Theta_{gnn}^{(2)}]_{t-1}$. Here, we introduce an additional parameter $\alpha \in [0, 1]$ as the exploration coefficient to control the levels of exploration (larger the $\alpha$ values will lead to higher levels of exploration). And we will show the experiment results with $\alpha \in \{0, 0.1, 0.3, 0.7, 1.0\}$ on the "MNIST" and the "Yelp" data sets.

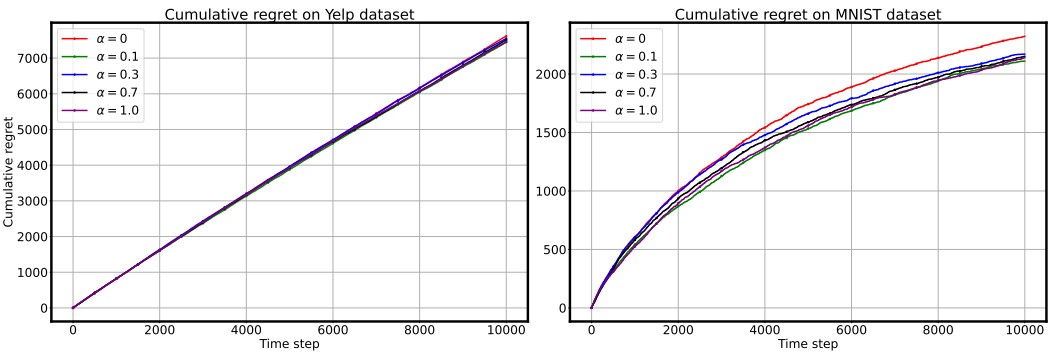

Figure 4: Cumulative regrets for different exploration coefficients $\alpha$.

In Figure 4, we illustrate the effects of different exploration coefficients. Regarding the results in the left figure ("Yelp" data set), the adaptive exploration indeed helps to improve the performance GNB, but the performances of GNB do not differ dramatically with different $\alpha$ values. As the "Yelp" data set contains inherent noise, the curve of cumulative regrets (including cumulative regrets of the other benchmark algorithms) tends to follow a near-linear growing rate. However, our carefully designed adaptive exploration strategy based on user exploration graphs is still helpful to improve the overall performance, and this is validated by the fact that setting $\alpha = 1$ will lead to better performance compared with the situation when no exploration strategy is involved ($\alpha = 0$). On the other hand, based on the figure on the right hand side ("MNIST" data set), different $\alpha$ values tend to have relatively divergent results. The reason can be that in the "MNIST" data set, the mapping from arm contexts to the rewards is more complicated compared with that of the "Yelp" data set. Thus, the adaptive exploration strategy is able to prominently improve the performance of GNB by flexibly estimating potential gains of different classes with the estimated "fine-grained" user (class) correlations.

Recall that there exists a parameter $k$ for the GNB framework in **Eq.** 3, which controls the user neighborhood hops that the two GNN models learn from. In this subsection, we will present the experiment results with $k \in \{1, 2, 3\}$ on the "MNIST" data set and the "Yelp" data set, which are presented in Figure 5.

Based on the results on the two data sets, we can observe that setting $k = 1$, namely making the GNB learn directly from the 1-hop neighborhood, tends to yield the best result. This might be due to the fact that since our user graphs are staying as connected graphs while the user correlations are encoded by the edge weights, learning directly from the neighbor would be good enough. And the pair-wise user correlations between the target user and every other user have already been taken into consideration. Meantime, with larger $k$ values ($k = 2, 3$), raising the matrix to the power of $k$ would lead to more even entry values across the adjacency matrix, which can be related to the over-smoothing problem (Xu et al., 2018; 2021). The figure on the right hand side ("MNIST" data set) may support this claim. Since it has already been shown in the Figure 2 that applying one single

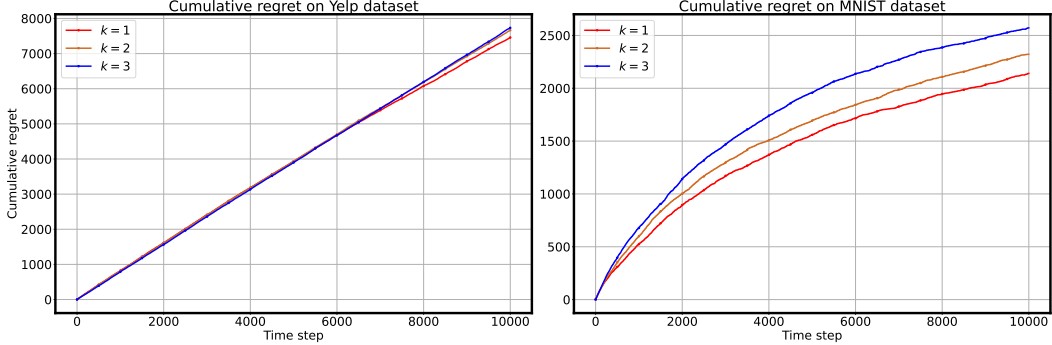

Figure 5: Cumulative regrets for different neighborhood hops $k$.

estimator across all users (classes), i.e., Neural-Pool and EE-Net, will lead to poor performances, the "MNIST" data set tend to have complex correlations among different classes. In this case, when we increase $k$, different user pairs tend to have similar correlations because entries of the adjacency matrix become more close to each other, which may lead to extra estimation error.

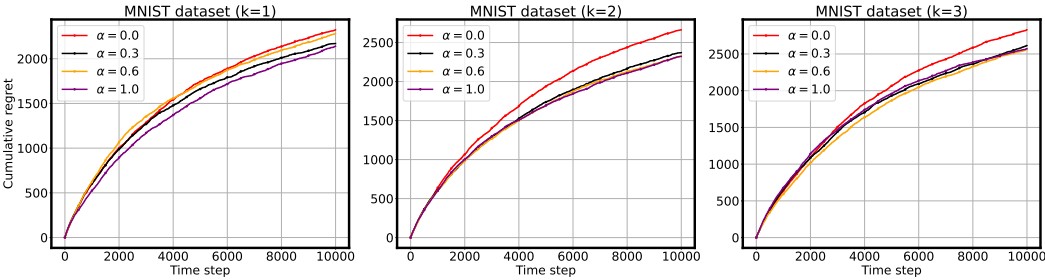

Figure 6: Cumulative regrets for different neighborhood hops $k$ and exploration parameter $\alpha$ for the MNIST data set.

Moreover, we also conduct the experiments on the MNIST data set with different sets of $\alpha$ and $k$ parameters jointly, as shown in Figure 6. Following our conclusion above, setting $k = 1$ generally leads to better results and the adaptive exploration strategy offers considerable help to improve the GNB's performance. One phenomenon to note is that when we increase the value of parameter $k$, the performance difference of GNB with different $\alpha$ values will shrink. One reason for this situation is that when we increase the $k$ value, the propagated adjacency matrix of the user graph will become more "smooth", which makes the users closer to each other in terms of similarity. In this case, the effect of the adaptive exploration strategy can be affected as the user correlations estimated are less divergent.

## B.4  EXPERIMENTS WITH DIFFERENT NUMBER OF UNDERLYING USER GROUPS

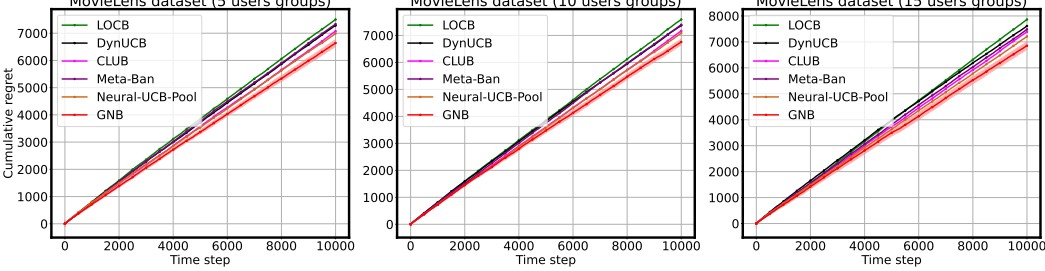

Figure 7: Cumulative regrets for different number of underlying user groups (MovieLens data set).

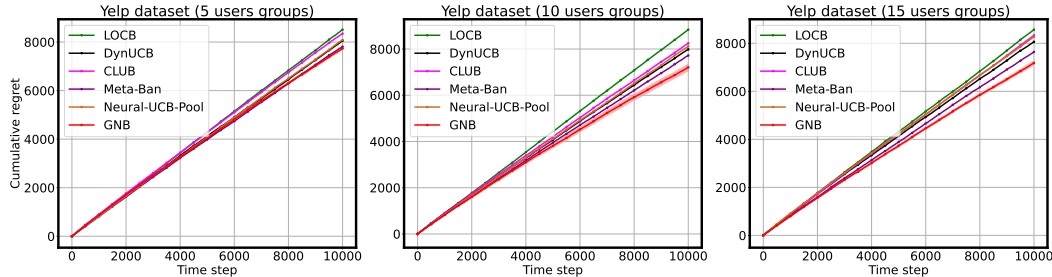

Figure 8: Cumulative regrets for different number of underlying user groups (Yelp data set).

To better understand the influence of potential underlying user clusters, we conduct the experiments on the MovieLens and the Yelp data sets, with controlled number of underlying user groups. The underlying user groups are derived by using hierarchical clustering on the user features, and we maintain approximately a total of 50 users. Here, we apply four representative baselines with relatively good performances, which are DynUCB (Nguyen & Lauw, 2014) [fixed number of user clusters], LOCB (Ban & He, 2021) [fixed number of user clusters], CLUB (Gentile et al., 2014) [distance-based user clustering], Neural-UCB-Pool (Zhou et al., 2020) [neural single-bandit algorithm], and Meta-Ban (Ban et al., 2022a) [neural user clustering bandits]. In particular, DynUCB and LOCB are provided with the **true cluster number** as the prior knowledge to determine the quantity of initial user clusters / random seeds. The experiment results are shown in Fig. 7 and Fig. 8.

As we can see from the results, our proposed GNB consistently outperforms other baselines across different data sets and number of user groups. In particular, with more underlying user groups, the performance improvement of GNB over the baselines will slightly increase, due to the increasingly complicated user correlations. The modeling of fine-grained user correlations and the representation power of our GNN-based architecture can help explain GNB's good performance, and the ability of utilizing user correlations.

## B.5 EXPERIMENTS WITH APPROXIMATED USER NEIGHBORHOOD

In this subsection, we conduct experiments to support our claim that applying approximated user neighborhoods is a feasible solution for increasing number of users (Remark 3.2). Then, we consider three scenarios where the number of users $n \in \{200, 300, 500\}$. Meanwhile, we let the size of the approximated user neighborhood $\tilde{\mathcal{N}}^{(1)}(u_t), \tilde{\mathcal{N}}^{(2)}(u_t)$ fix to $\tilde{n} = |\tilde{\mathcal{N}}^{(1)}(u_t)| = |\tilde{\mathcal{N}}^{(2)}(u_t)| = 50$ for all these three experiment settings, and the neighborhood users are sampled from the user pool $\mathcal{U}$ for experiments.

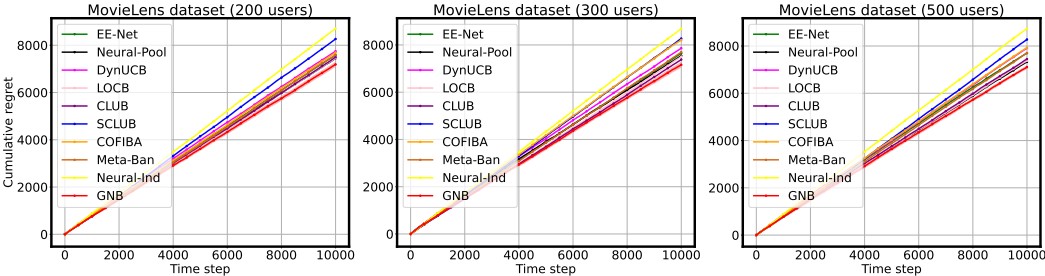

Figure 9: Cumulative regrets for different number of users with approximated user neighborhood (MovieLens data set).

The experiment results are shown in Figure 9. Here, we see that the proposed GNB still outperforms the baselines with increasing number of users. In particular, given a total of 500 users, the approximated neighborhood is only $1/10$ (50 users) of the overall user pool. These results can serve as a clear support that applying approximated user neighborhoods (Remark 3.2) is a practical way to scale-up GNB in real-world application scenarios.

## B.6 RUNNING TIME ON REAL DATA SETS

| Methods | Data sets ($10^4$ rounds) | | | |
|---|---|---|---|---|
| | MovieLens | Yelp | MNIST | Shuttle |
| CLUB | 230 | 36 | 359 | 4 |
| SCLUB | 274 | 82 | 363 | 4 |
| LOCB | 215 | 31 | 223 | 3 |
| DynUCB | 214 | 29 | 357 | 3 |
| Neural-Pool | 509 | 321 | 289 | 226 |
| Neural-Ind | 472 | 281 | 265 | 169 |
| EE-Net | 2435 | 2149 | 2903 | 2052 |
| COFIBA | 321 | 135 | 11874 | 13 |
| Meta-Ban | 20170 | 19825 | 18172 | 18101 |
| **GNB** (Ours) | 14121 [4295] | 12506 [4082] | 4299 [1072] | 1606 [185] |

Table 1: Average running time results (seconds) on real data sets. The running time **in the brackets "[]"** is the actual time consumption for recommendation w/o the time consumption for training.

From **Table** 1, we see that compared with the most closely related work, Meta-Ban, our proposed GNB is generally faster, since GNB does not required to re-train the model for each candidate arm.

Although the other baselines, especially the linear baselines tend to run much faster compared with our proposed GNB, their experiment performances (Section 5) are also not comparable with our proposed GNB as their linear assumption is too strong for most application scenarios. In particular, for the data set with large arm context dimension $d$, the mapping from the arm context to the reward will be much more complicated. In this case, as shown by the experiments on the MNIST data set ($d = 784$) in Figure 2, the neural algorithms manage to achieve an undoubtedly huge improvement over the linear algorithms, and have the reasonable running time.

Here, the numbers in the brackets "[]" are the time consumption for the actual recommendation process. We have the following remarks: (1) Based on the running time in the brackets, we see that for the two recommendation tasks, GNB takes approximately $\sim 0.4$ second / per round to make the arm recommendation for the received user, which is reasonable in real-world cases; (2) In all the experiments, we train the GNB framework per 100 rounds after $T > 1000$ and still manage to achieve good performance. Thus, the running time of GNB in a long run could be further significantly improved by reducing the training frequency since we have already have enough data and an accurate framework; (3) Moreover, since we are actually predicting the rewards and potential gain for all the nodes within the user graph (or the "approximated" user graph), GNB is able to handle multiple users in each round simultaneously without running the recommendation procedure multiple times, which is efficient in real-world cases.

## C    USER NETWORKS ARCHITECTURE.

Here, we can choose different architectures for $f_u^{(1)}(\cdot), f_u^{(2)}(\cdot)$ to deal with various application scenarios (e.g., Convolutional Neural Networks [CNNs] for recommendation tasks of visual contents). In this paper, for the theoretical analysis and experiments, we apply separate $L$-layer fully-connected (FC) networks for user exploitation models and exploration models, as

$$f_u(\boldsymbol{\chi}; \boldsymbol{\Theta}_u) = \boldsymbol{\Theta}_L \sigma(\boldsymbol{\Theta}_{L-1}\sigma(\boldsymbol{\Theta}_{L-2}\ldots\sigma(\boldsymbol{\Theta}_1\boldsymbol{\chi}))) \tag{7}$$

with $\boldsymbol{\Theta}_u = [\text{vec}(\boldsymbol{\Theta}_1)^\intercal, \ldots, \text{vec}(\boldsymbol{\Theta}_L)^\intercal]^\intercal$ being the trainable parameters, and $\sigma$ being the ReLU activation. Here, since $f_u^{(1)}(\cdot), f_u^{(2)}(\cdot)$ are both $L$-layer networks shown in **Eq.**7, the input $\boldsymbol{\chi}$ can be either the arm $\boldsymbol{x}$ or the network gradient $\nabla_{\boldsymbol{\Theta}_u^{(1)}} f_u^{(1)}(\cdot; \boldsymbol{\Theta}_u^{(1)})$.

**Initialization.** Then, the weight matrix of the input layer is different for two user networks where $\boldsymbol{\Theta}_1^{(1)} \in \mathbb{R}^{m \times d}$ and $\boldsymbol{\Theta}_1^{(2)} \in \mathbb{R}^{m \times p}$. The rest of the layers will be the same comparing the two kinds of user networks, which are $\boldsymbol{\Theta}_l \in \mathbb{R}^{m \times m}, l \in [2, \cdots, L-1]$, and $\boldsymbol{\Theta}_L \in \mathbb{R}^{1 \times m}$.

## D    PSEUDO-CODE FOR ESTIMATING USER GRAPHS AND TRAINING THE GNB FRAMEWORK

---
**ALGORITHM 2:** Estimating Arm-Specific User Graphs

---
1  **Input:** Model parameters $\boldsymbol{\Theta}_{t-1}$. Functions for edge weight estimation
$\Psi^{(1)}(\cdot, \cdot), \Psi^{(2)}(\cdot, \cdot) : \mathbb{R} \times \mathbb{R} \mapsto \mathbb{R}$.
2  **Output:** Updated user graphs $\{\mathcal{G}_{i,t}^{(1)}\}_{i\in[a]}, \{\mathcal{G}_{i,t}^{(2)}\}_{i\in[a]}$.
3  Initialize $\{\mathcal{G}_{i,t}^{(1)}\}_{i\in[a]}, \{\mathcal{G}_{i,t}^{(2)}\}_{i\in[a]}$.
4  **for** *each user $u \in \mathcal{U}$* **do**
5      **for** *each arm $\boldsymbol{x}_{i,t} \in \mathcal{X}_t, i \in [a]$* **do**
6          Compute $\hat{r}_{u,i} = f_u^{(1)}(\boldsymbol{x}_{i,t}; [\boldsymbol{\Theta}_{u_t}^{(1)}]_{t-1})$, and
        $\hat{b}_{u,i} = f_u^{(2)}(\nabla_{\boldsymbol{\Theta}_{u_t}^{(1)}} f_u^{(1)}(\boldsymbol{x}_{i,t}; [\boldsymbol{\Theta}_{u_t}^{(1)}]_{t-1}); [\boldsymbol{\Theta}_{u_t}^{(2)}]_{t-1})$.
7      **end**
8  **end**
9  **for** *each arm $\boldsymbol{x}_{i,t} \in \mathcal{X}_t$* **do**
10      **for** *each user pair $(u, u') \in \mathcal{U} \times \mathcal{U}$* **do**
11          For edge weight $w_{i,t}^{(1)}(u, u') \in W_{i,t}^{(1)}$, update $w_{i,t}^{(1)}(u, u') = \Psi^{(1)}(\hat{r}_{u,i}, \hat{r}_{u',i})$.
12          For edge weight $w_{i,t}^{(2)}(u, u') \in W_{i,t}^{(2)}$, update $w_{i,t}^{(2)}(u, u') = \Psi^{(2)}(\hat{b}_{u,i}, \hat{b}_{u',i})$.
13      **end**
14  **end**
15  Return user graphs $\{\mathcal{G}_{i,t}^{(1)}\}_{i\in[a]}, \{\mathcal{G}_{i,t}^{(2)}\}_{i\in[a]}$.

---

---

**ALGORITHM 3:** Model Training

---

1 **Input:** Initial parameter $\boldsymbol{\Theta_0}$, step size $\eta_1, \eta_2$, training steps $J_1, J_2$, network width $m$. Updated user graphs $\mathcal{G}_t^{(1)}, \mathcal{G}_t^{(2)}$. Served user $u_t$.

2 **Output:** Updated model parameters $[\boldsymbol{\Theta}_{u_t}^{(1)}]_t, [\boldsymbol{\Theta}_{u_t}^{(2)}]_t, [\boldsymbol{\Theta}_{gnn}^{(1)}]_t$ and $[\boldsymbol{\Theta}_{gnn}^{(2)}]_t$.

3 $[\boldsymbol{\Theta}_{u_t}^{(1)}]_t, [\boldsymbol{\Theta}_{u_t}^{(2)}]_t = $ User-Model-Training $\left(u_t, [\boldsymbol{\Theta}_{u_t}^{(1)}]_0, [\boldsymbol{\Theta}_{u_t}^{(2)}]_0\right)$.

4 **for** $\forall u' \in \mathcal{U}, u' \neq u_t$ **do**

5 $\quad [\boldsymbol{\Theta}_{u'}^{(1)}]_t \leftarrow [\boldsymbol{\Theta}_{u'}^{(1)}]_{t-1}, \quad [\boldsymbol{\Theta}_{u'}^{(2)}]_t \leftarrow [\boldsymbol{\Theta}_{u'}^{(2)}]_{t-1}$

6 **end**

7 $[\boldsymbol{\Theta}_{gnn}^{(1)}]_t, [\boldsymbol{\Theta}_{gnn}^{(2)}]_t = $ GNN-Model-Training $\left([\boldsymbol{\Theta}_{gnn}^{(1)}]_0, [\boldsymbol{\Theta}_{gnn}^{(2)}]_0\right)$.

8 Return $[\boldsymbol{\Theta}_{u_t}^{(1)}]_t, [\boldsymbol{\Theta}_{u_t}^{(2)}]_t, [\boldsymbol{\Theta}_{gnn}^{(1)}]_t, [\boldsymbol{\Theta}_{gnn}^{(2)}]_t$.

9 **Procedure** *User-Model-Training* $\left(u_t, [\boldsymbol{\Theta}_{u_t}^{(1)}]_0, [\boldsymbol{\Theta}_{u_t}^{(2)}]_0\right)$

10 $\quad [\boldsymbol{\Theta}_{u_t}^{(1)}]^0 \leftarrow [\boldsymbol{\Theta}_{u_t}^{(1)}]_0, \quad [\boldsymbol{\Theta}_{u_t}^{(2)}]^0 \leftarrow [\boldsymbol{\Theta}_{u_t}^{(2)}]_0$.

11 $\quad$ # Training of $f_u^{(1)}(\cdot)$

12 $\quad$ Let $\mathcal{L}(\Theta_{u_t}^{(1)}) := \sum_{\tau \in \mathcal{T}_{u_t,t}} |f_u^{(1)}(\boldsymbol{x}_\tau; \boldsymbol{\Theta}_{u_t}^{(1)}) - r_\tau|^2$

13 $\quad$ **for** $j = 1, 2, \ldots, J_1$ **do**

14 $\quad\quad [\boldsymbol{\Theta}_{u_t}^{(1)}]^j = [\boldsymbol{\Theta}_{u_t}^{(1)}]^{j-1} - \eta_1 \cdot \nabla_{\boldsymbol{\Theta}} \mathcal{L}([\boldsymbol{\Theta}_{u_t}^{(1)}]^{j-1})$

15 $\quad$ **end**

16 $\quad$ # Training of $f_u^{(2)}(\cdot)$

17 $\quad$ Let $\mathcal{L}(\Theta_{u_t}^{(2)}) :=$
$\quad\quad \sum_{\tau \in \mathcal{T}_{u_t,t}} |f_u^{(2)}(\nabla_{\boldsymbol{\Theta}_{u_t}^{(1)}} f_u^{(1)}(\boldsymbol{x}_\tau; [\boldsymbol{\Theta}_{u_t}^{(1)}]_{\tau-1}); \boldsymbol{\Theta}_{u_t}^{(2)}) - \left(r_\tau - f_u^{(1)}(\boldsymbol{x}_\tau; [\boldsymbol{\Theta}_{u_t}^{(1)}]_{\tau-1})\right)|^2$

18 $\quad$ **for** $j = 1, 2, \ldots, J_1$ **do**

19 $\quad\quad [\boldsymbol{\Theta}_{u_t}^{(2)}]^j = [\boldsymbol{\Theta}_{u_t}^{(2)}]^{j-1} - \eta_1 \cdot \nabla_{\boldsymbol{\Theta}} \mathcal{L}([\boldsymbol{\Theta}_{u_t}^{(2)}]^{j-1})$

20 $\quad$ **end**

21 $\quad$ Let $[\widehat{\boldsymbol{\Theta}}_{u_t}^{(1)}]_t \leftarrow [\boldsymbol{\Theta}_{u_t}^{(1)}]^{J_1}, [\widehat{\boldsymbol{\Theta}}_{u_t}^{(2)}]_t \leftarrow [\boldsymbol{\Theta}_{u_t}^{(2)}]^{J_1}$

22 $\quad$ Sample and return new parameters $([\boldsymbol{\Theta}_{u_t}^{(1)}]_t, [\boldsymbol{\Theta}_{u_t}^{(2)}]_t) \sim \{([\widehat{\boldsymbol{\Theta}}_{u_t}^{(1)}]_\tau, [\widehat{\boldsymbol{\Theta}}_{u_t}^{(2)}]_\tau)\}_{\tau \in [t]}$.

23 **end**

24 **Procedure** *GNN-Model-Training* $\left([\boldsymbol{\Theta}_{gnn}^{(1)}]_0, [\boldsymbol{\Theta}_{gnn}^{(2)}]_0\right)$

25 $\quad [\boldsymbol{\Theta}_{gnn}^{(1)}]^0 \leftarrow [\boldsymbol{\Theta}_{gnn}^{(1)}]_0, \quad [\boldsymbol{\Theta}_{gnn}^{(2)}]^0 \leftarrow [\boldsymbol{\Theta}_{gnn}^{(2)}]_0$.

26 $\quad$ # Training of $f_{gnn}^{(1)}(\cdot)$

27 $\quad$ Let $\mathcal{L}(\Theta_{gnn}^{(1)}) := \sum_{\tau \in [t]} |f_{gnn}^{(1)}(\boldsymbol{x}_\tau, \mathcal{G}_\tau^{(1)}; \boldsymbol{\Theta}_{gnn}^{(1)}) - r_\tau|^2$

28 $\quad$ **for** $j = 1, 2, \ldots, J_2$ **do**

29 $\quad\quad [\boldsymbol{\Theta}_{gnn}^{(1)}]^j = [\boldsymbol{\Theta}_{gnn}^{(1)}]^{j-1} - \eta_2 \cdot \nabla_{\boldsymbol{\Theta}} \mathcal{L}([\boldsymbol{\Theta}_{gnn}^{(1)}]^{j-1})$

30 $\quad$ **end**

31 $\quad$ # Training of $f_{gnn}^{(2)}(\cdot)$

32 $\quad$ Apply $f_{gnn}^{(1)}(\boldsymbol{x}_\tau)$ to denote $f_{gnn}^{(1)}(\boldsymbol{x}_\tau, \mathcal{G}_\tau^{(1)}; [\boldsymbol{\Theta}_{gnn}^{(1)}]_{\tau-1})$.

33 $\quad$ Let $\mathcal{L}(\Theta_{gnn}^{(2)}) := \sum_{\tau \in [t]} |f_{gnn}^{(2)}(\nabla_{\boldsymbol{\Theta}_{gnn}^{(1)}} f_{gnn}^{(1)}(\boldsymbol{x}_\tau), \mathcal{G}_\tau^{(2)}; \boldsymbol{\Theta}_{gnn}^{(2)}) - \left(r_\tau - f_{gnn}^{(1)}(\boldsymbol{x}_\tau, \mathcal{G}_\tau^{(1)})\right)|^2$

34 $\quad$ **for** $j = 1, 2, \ldots, J_2$ **do**

35 $\quad\quad [\boldsymbol{\Theta}_{gnn}^{(2)}]^j = [\boldsymbol{\Theta}_{gnn}^{(2)}]^{j-1} - \eta_2 \cdot \nabla_{\boldsymbol{\Theta}} \mathcal{L}([\boldsymbol{\Theta}_{gnn}^{(2)}]^{j-1})$

36 $\quad$ **end**

37 $\quad$ Let $[\widehat{\boldsymbol{\Theta}}_{gnn}^{(1)}]_t \leftarrow [\boldsymbol{\Theta}_{gnn}^{(1)}]^{J_2}, [\widehat{\boldsymbol{\Theta}}_{gnn}^{(2)}]_t \leftarrow [\boldsymbol{\Theta}_{gnn}^{(2)}]^{J_2}$

38 $\quad$ Sample and return new parameters $([\boldsymbol{\Theta}_{gnn}^{(1)}]_t, [\boldsymbol{\Theta}_{gnn}^{(2)}]_t) \sim \{([\widehat{\boldsymbol{\Theta}}_{gnn}^{(1)}]_\tau, [\widehat{\boldsymbol{\Theta}}_{gnn}^{(2)}]_\tau)\}_{\tau \in [t]}$.

39 **end**

---

# E    PROOF OF **THEOREM** 4.2

Before presenting the regret bound after $T$ rounds, we proceed to bound the regret at a single time step $t \in [T]$. Recall that there are two kinds of user graphs $\{\mathcal{G}_{i,t}^{(1)}\}_{i \in [a]}$, $\{\mathcal{G}_{i,t}^{(2)}\}_{i \in [a]}$ at each time step $t$, while we can also build true user exploitation graph $\{\mathcal{G}_{i,t}^{(1),*}\}_{i \in [a]}$ and true user exploration graph $\{\mathcal{G}_{i,t}^{(2),*}\}_{i \in [a]}$ based on the **Definition** 1 and **Definition** 3 respectively. Comparably, the true normalized adjacency matrices of $\mathcal{G}_{i,t}^{(1),*}, i \in [a]$ are represented as $\boldsymbol{S}_{i,t}^{(1),*}$.

With $r_t, r_t^*$ separately being rewards for actual selected arm $\boldsymbol{x}_t \in \mathcal{X}_t$ and the optimal arm $\boldsymbol{x}_t^* \in \mathcal{X}_t$, we formulate the pseudo-regret for a single round $t$ as $R_t = \mathbb{E}[r_t^*|u_t, \mathcal{X}_t] - \mathbb{E}[r_t|u_t, \mathcal{X}_t]$ based on the candidate arms $\mathcal{X}_t$ and received user $u_t$ for the current round $t$. Here, regarding our arm pulling mechanism in **Algorithm** 1, we have $f_{gnn}(\boldsymbol{x}_t) = f_{gnn}^{(1)}(\boldsymbol{x}_t, \mathcal{G}_t^{(1)}; [\boldsymbol{\Theta}_{gnn}^{(1)}]_{t-1}) + f_{gnn}^{(2)}(\nabla f_t^{(1)}(\boldsymbol{x}_t), \mathcal{G}_t^{(2)}; [\boldsymbol{\Theta}_{gnn}^{(2)}]_{t-1})$ given the selected arm $\boldsymbol{x}_t$ with the input gradient $\nabla f_t^{(1)}(\boldsymbol{x}_t) = \frac{\nabla_{\boldsymbol{\Theta}_{gnn}^{(1)}} f_{gnn}^{(1)}(\boldsymbol{x}_t, \mathcal{G}_t^{(1)};[\boldsymbol{\Theta}_{gnn}^{(1)}]_{t-1})}{c_g L}$ ($c_g > 0$ as the normalization factor, such that $\|\nabla f_t^{(1)}(\boldsymbol{x}_t)\|_2 \leq 1$), and the estimated user graphs $\mathcal{G}_t^{(1)}, \mathcal{G}_t^{(2)}$ related to $\boldsymbol{x}_t$. Analogously, we also have estimated user graphs $\mathcal{G}_{t,*}^{(1)}, \mathcal{G}_{t,*}^{(2)}$ for the optimal arm $\boldsymbol{x}_t^*$.

Then, in round $t \in [T]$, the single-round regret $R_t$ can be bounded as

$$
\begin{aligned}
R_t &= \mathbb{E}[r_t^*|u_t, \mathcal{X}_t] - \mathbb{E}[r_t|u_t, \mathcal{X}_t] \\
&= \mathbb{E}[r_t^*|u_t, \mathcal{X}_t] - f_{gnn}(\boldsymbol{x}_t) + f_{gnn}(\boldsymbol{x}_t) - \mathbb{E}[r_t|u_t, \mathcal{X}_t] \\
&\underset{(i)}{\leq} \mathbb{E}[r_t^*|u_t, \mathcal{X}_t] - f_{gnn}(\boldsymbol{x}_t^*) + f_{gnn}(\boldsymbol{x}_t) - \mathbb{E}[r_t|u_t, \mathcal{X}_t] \\
&\leq \mathbb{E}\big[|r_t^* - f_{gnn}(\boldsymbol{x}_t^*)|\big|u_t, \mathcal{X}_t\big] + \mathbb{E}\big[|r_t - f_{gnn}(\boldsymbol{x}_t)|\big|u_t, \mathcal{X}_t\big] \\
&= \mathbb{E}\bigg[|f_{gnn}^{(2)}(\nabla f_t^{(1)}(\boldsymbol{x}_t^*), \mathcal{G}_{t,*}^{(2)}; [\boldsymbol{\Theta}_{gnn}^{(2)}]_{t-1}) - (r_t^* - f_{gnn}^{(1)}(\boldsymbol{x}_t^*, \mathcal{G}_{t,*}^{(1)}; [\boldsymbol{\Theta}_{gnn}^{(1)}]_{t-1}))|\bigg|u_t, \mathcal{X}_t\bigg] + \\
&\quad \mathbb{E}\bigg[|f_{gnn}^{(2)}(\nabla f_t^{(1)}(\boldsymbol{x}_t), \mathcal{G}_t^{(2)}; [\boldsymbol{\Theta}_{gnn}^{(2)}]_{t-1}) - (r_t - f_{gnn}^{(1)}(\boldsymbol{x}_t, \mathcal{G}_t^{(1)}; [\boldsymbol{\Theta}_{gnn}^{(1)}]_{t-1}))|\bigg|u_t, \mathcal{X}_t\bigg] \\
&= \mathsf{CB}_t(\boldsymbol{x}_t) + \mathsf{CB}_t(\boldsymbol{x}_t^*)
\end{aligned}
$$

where inequality (i) is due to the arm pulling mechanism, i.e., $f_{gnn}(\boldsymbol{x}_t) \geq f_{gnn}(\boldsymbol{x}_t^*)$, and $\mathsf{CB}_t(\cdot)$ is the regret bound function at round $t$ formulated by the last equation. Then, given an arbitrary candidate arm $\boldsymbol{x} \in \mathcal{X}_t$ with reward $r$, and its estimated user graphs $\mathcal{G}^{(1)}, \mathcal{G}^{(2)}$, we have

$$
\begin{aligned}
\mathsf{CB}_t(\boldsymbol{x}) &= \mathbb{E}\bigg[|f_{gnn}^{(2)}(\nabla f_t^{(1)}(\boldsymbol{x}), \mathcal{G}^{(2)}; [\boldsymbol{\Theta}_{gnn}^{(2)}]_{t-1}) - (r - f_{gnn}^{(1)}(\boldsymbol{x}, \mathcal{G}^{(1)}; [\boldsymbol{\Theta}_{gnn}^{(1)}]_{t-1}))|\bigg|u_t, \mathcal{X}_t\bigg] \\
&\leq \underbrace{\mathbb{E}\bigg[|f_{gnn}^{(2)}(\nabla f_t^{(1),*}(\boldsymbol{x}), \mathcal{G}^{(2),*}; [\boldsymbol{\Theta}_{gnn}^{(2)}]_{t-1}) - (r - f_{gnn}^{(1)}(\boldsymbol{x}, \mathcal{G}^{(1),*}; [\boldsymbol{\Theta}_{gnn}^{(1)}]_{t-1}))|\bigg|u_t, \mathcal{X}_t\bigg]}_{I_1} + \\
&\quad \underbrace{\mathbb{E}\bigg[|f_{gnn}^{(1)}(\boldsymbol{x}, \mathcal{G}^{(1),*}; [\boldsymbol{\Theta}_{gnn}^{(1)}]_{t-1}) - f_{gnn}^{(1)}(\boldsymbol{x}, \mathcal{G}^{(1)}; [\boldsymbol{\Theta}_{gnn}^{(1)}]_{t-1})|\bigg|u_t, \mathcal{X}_t\bigg]}_{I_2} + \\
&\quad \underbrace{\mathbb{E}\bigg[|f_{gnn}^{(2)}(\nabla f_t^{(1),*}(\boldsymbol{x}), \mathcal{G}^{(2),*}; [\boldsymbol{\Theta}_{gnn}^{(2)}]_{t-1}) - f_{gnn}^{(2)}(\nabla f_t^{(1),*}(\boldsymbol{x}), \mathcal{G}^{(2)}; [\boldsymbol{\Theta}_{gnn}^{(2)}]_{t-1})|\bigg|u_t, \mathcal{X}_t\bigg]}_{I_3} + \\
&\quad \underbrace{\mathbb{E}\bigg[|f_{gnn}^{(2)}(\nabla f_t^{(1),*}(\boldsymbol{x}), \mathcal{G}^{(2)}; [\boldsymbol{\Theta}_{gnn}^{(2)}]_{t-1}) - f_{gnn}^{(2)}(\nabla f_t^{(1)}(\boldsymbol{x}), \mathcal{G}^{(2)}; [\boldsymbol{\Theta}_{gnn}^{(2)}]_{t-1})|\bigg|u_t, \mathcal{X}_t\bigg]}_{I_4} .
\end{aligned}
$$
(8)

Here, we have the term $I_1$ representing the estimation error induced by the GNN model parameters $\{[\boldsymbol{\Theta}_{gnn}^{(1)}]_{t-1}, [\boldsymbol{\Theta}_{gnn}^{(2)}]_{t-1}\}$, the term $I_2$ denoting the error caused by the estimation of user exploitation

graph. Then, error term $I_3$ is caused by the estimation of user exploitation graph, and term $I_4$ is the output difference given input gradients $\nabla f_t^{(1),*}(\boldsymbol{x})$ and $\nabla f_t^{(1)}(\boldsymbol{x})$, which are individually generated by true user exploitation graph $\mathcal{G}^{(1),*}$ and the estimated exploitation graph $\mathcal{G}^{(1)}$.

These four terms $I_1, I_2, I_3, I_4$ are respectively bounded by **Lemma** G.2 (**Corollary** G.3 and the bounds in Subsection G.1), **Lemma** G.4, **Lemma** G.5, and **Lemma** G.7 in the appendix. Then, with the notation from **Theorem** 4.2, the pseudo regret after $T$ rounds, namely $R(T)$, can be bounded by

$$R(t) = \sum_{t \in [T]} R_t$$

$$\leq 2 \cdot \left( \sqrt{t} \cdot \left( \sqrt{2\xi_2} + \frac{3L}{\sqrt{2}} + (1 + \gamma_2) \sqrt{2 \log(\frac{Tn \cdot a}{\delta})} \right) \right.$$

$$+ \left( 1 + \mathcal{O}(\frac{tL^3 \log^{5/6}(m)}{\rho^{1/3} m^{1/6}}) \right) \cdot \mathcal{O}(\frac{t^3 L}{\rho \sqrt{m}} \log(m)) + \mathcal{O}(\frac{t^4 L^2 \log^{11/6}(m)}{\rho^{4/3} m^{1/6}})$$

$$+ 2 \cdot \mathcal{O}(L) \cdot \sqrt{8t} \cdot \left( \sqrt{2\xi_1} + \frac{3L}{\sqrt{2}} + (1 + \gamma_1) \sqrt{2 \log(\frac{Tn \cdot a}{\delta})} \right)$$

$$\left. + \mathcal{O}(\frac{tL^5 \log^{5/6}(m)}{\rho^{1/3} m^{1/6}}) + \mathcal{O}(L^2) \cdot \sqrt{8t} \cdot \left( \sqrt{2\xi_1} + \frac{3L}{\sqrt{2}} + (1 + \gamma_1) \sqrt{2 \log(\frac{Tn \cdot a}{\delta})} \right) + 4\Gamma_t \right) \implies$$

$$R(T) \leq 2 \cdot \sqrt{T} \left( \sqrt{2\xi_2} + \frac{3L}{\sqrt{2}} + (1 + \gamma_2) \sqrt{2 \log(\frac{Tn \cdot a}{\delta})} \right)$$

$$+ \sqrt{T} \cdot \mathcal{O}(L^2) \cdot \left( \sqrt{2\xi_1} + \frac{3L}{\sqrt{2}} + (1 + \gamma_1) \sqrt{2 \log(\frac{Tn \cdot a}{\delta})} \right) + \mathcal{O}(1)$$

$$\leq \sqrt{T} \cdot \left( (\sqrt{8\xi_2} + \mathcal{O}(L^2) \sqrt{2\xi_1}) + \mathcal{O}(L^3) + \mathcal{O}(L^2) \cdot \sqrt{2 \log(\frac{Tn \cdot a}{\delta})} \right) + \sqrt{T} \cdot \mathcal{O}(L) + \mathcal{O}(1)$$

where the second inequality is because we have sufficient large network width $m \geq \Omega\left( \mathrm{Poly}(T, L, a, \frac{1}{\rho}) \cdot \log(1/\delta) \right)$ as indicated in **Theorem** 4.2. Here, since $m \geq \Omega(\mathrm{Poly}(T))$, terms $\gamma_1, \gamma_2$ can also be bounded by $\mathcal{O}(1)$. Therefore,

$$R(T) \leq \sqrt{T} \cdot \left( (\sqrt{8\xi_2} + \mathcal{O}(L^2) \sqrt{2\xi_1}) + \mathcal{O}(L^3) + \mathcal{O}(L^2) \cdot \sqrt{2 \log(\frac{Tn \cdot a}{\delta})} \right) + \sqrt{T} \cdot \mathcal{O}(L) + \mathcal{O}(1)$$

$$\leq \sqrt{T} \cdot \left( \mathcal{O}(L^3) + \mathcal{O}(L^2) \cdot \sqrt{2 \log(\frac{Tn \cdot a}{\delta})} \right) + \sqrt{T} \cdot \mathcal{O}(L) + \mathcal{O}(L^2) + \mathcal{O}(1)$$

$$= \sqrt{T} \cdot \mathcal{O}(L) + \sqrt{T} \cdot \mathcal{O}(L^3) + \sqrt{T} \cdot \mathcal{O}(L^2) \cdot \sqrt{\log(\frac{Tn \cdot a}{\delta})} + \mathcal{O}(L^2) + \mathcal{O}(1)$$

when we have $\xi_1, \xi_2 \leq \mathcal{O}(\frac{1}{T})$. The proof is then completed.

Apart from the two remarks in the main body (Remark 4.4, 4.3), we also want to mention another improvement over existing works with the **Remark** E.1 below.

**Remark E.1** (Removing $d, \tilde{d}$ Terms). Existing neural single-bandit (i.e., with no user collaboration) algorithms (Zhou et al., 2020; Zhang et al., 2021) derive the bound $\mathcal{O}(\tilde{d}\sqrt{T} \log(T))$ based on neural gradient mappings and ridge regression, and they involve the effective dimension term $\tilde{d}$ of the NTK matrix, which can grow along with the scale of network parameters and number of rounds $T$. The linear user clustering algorithms (e.g., Li et al. (2019); Ban & He (2021); Gentile et al. (2017)) have the bound $\mathcal{O}(d\sqrt{T} \log(T))$ with the term of arm dimension $d$, which can be large given arm contexts in the high-dimensional space. Here, we improve their bounds by a multiplicative factor of $\sqrt{\log(T)}$ and remove the dimension terms $d, \tilde{d}$. We apply the generalization bound for over-parameterized neural networks (Allen-Zhu et al., 2019; Cao & Gu, 2019) instead of regression-based analysis to remove the $\sqrt{\log(T)}$ term, and the generalization error is also unrelated to $d$ or $\tilde{d}$ for over-parameterized neural networks.

## F    GENERALIZATION OF USER NETWORKS AFTER GD

In this section, we present the generalization results of user networks $f_u^{(1)}(\cdot; \mathbf{\Theta}_u^{(1)}), f_u^{(2)}(\cdot; \mathbf{\Theta}_u^{(2)}), u \in \mathcal{U}$. Up to a certain time step $t$ and for a given user $u \in \mathcal{U}$, we have all its past arm-reward pairs $\mathcal{P}_{u,t-1} = \{(\boldsymbol{x}_\tau, r_\tau)\}_{\tau \in \mathcal{T}_{u,t}}$. Before presenting the bounds, with two vectors $\tilde{\boldsymbol{x}}, \boldsymbol{x}$ as the input such that $\|\tilde{\boldsymbol{x}}\|_2 \leq 1, \|\boldsymbol{x}\|_2 = 1$, inspired by (Allen-Zhu et al., 2019), we first define the the following operator

$$\phi(\tilde{\boldsymbol{x}}, \boldsymbol{x}) = (\frac{\tilde{\boldsymbol{x}}}{\sqrt{2}}, \frac{\boldsymbol{x}}{2}, c) \tag{9}$$

as the concatenation of the two vectors $\frac{\tilde{\boldsymbol{x}}}{\sqrt{2}}, \frac{\boldsymbol{x}}{2}$ and one constant $c$, where $c = \sqrt{\frac{3}{4} - (\frac{\|\tilde{\boldsymbol{x}}\|_2}{\sqrt{2}})^2} \geq \frac{1}{2}$. And this operator makes the transformed vector $\|\phi(\tilde{\boldsymbol{x}}, \boldsymbol{x})\|_2 = 1$. The idea of this operator is to make the gradients $\nabla_{\mathbf{\Theta}_u^{(1)}} f_u^{(1)}(\cdot; \mathbf{\Theta}_u^{(1)})$ of the user exploitation model, which is the input of the user exploration model $f_u^{(2)}(\cdot)$, comply with the normalization requirement and the separateness assumption (**Assumption** 4.1). For the sake of analysis, we will adopt this operation in the following proof. Note that this operator is just one possible solution, and our results could be easily generalized to other forms of input gradients under the unit-length and separateness assumption. Similar ideas are also applied in previous works (Ban et al., 2022b).

### F.1    USER EXPLOITATION MODEL

With the convergence result presented in **Lemma** F.6, we could bound the output of the user exploitation model $f_u^{(1)}(\cdot)$ after GD with the following lemma.

**Lemma F.1.** *For the constants $\rho \in (0, \mathcal{O}(\frac{1}{L}))$ and $\xi_1 \in (0, 1)$, given user $u \in \mathcal{U}$ and its past records $\mathcal{P}_{u,t-1}$ up to time step $t$, we suppose $m, \eta_1, J_1$ satisfy the conditions in **Theorem** 4.2. Then, with probability at least $1 - \delta$, given an arm-reward pair $(\boldsymbol{x}, r)$, we have*

$$|f_u^{(1)}(\boldsymbol{x}; [\widehat{\mathbf{\Theta}}_u^{(1)}]_t)| \leq \gamma_1$$

*where*

$$\gamma_1 = 2 + \mathcal{O}\left(\frac{t^3 L}{n^3 \rho \sqrt{m}} \log m\right) + \mathcal{O}\left(\frac{L^2 t^4}{n^4 \rho^{4/3} m^{1/6}} \log^{11/6}(m)\right).$$

**Proof.** For brevity, we use $\widehat{\mathbf{\Theta}}_u^{(1)}$ to denote $[\widehat{\mathbf{\Theta}}_u^{(1)}]_t$. The LHS of the inequality could be written as

$$|f_u^{(1)}(\boldsymbol{x}; \widehat{\mathbf{\Theta}}_u^{(1)})| \leq |f_u^{(1)}(\boldsymbol{x}; \widehat{\mathbf{\Theta}}_u^{(1)}) - f_u^{(1)}(\boldsymbol{x}; [\widehat{\mathbf{\Theta}}_u^{(1)}]_0) - \langle \nabla_{[\widehat{\mathbf{\Theta}}_u^{(1)}]_0} f_u^{(1)}(\boldsymbol{x}; [\widehat{\mathbf{\Theta}}_u^{(1)}]_0), \widehat{\mathbf{\Theta}}_u^{(1)} - [\widehat{\mathbf{\Theta}}_u^{(1)}]_0 \rangle|$$

$$+ |f_u^{(1)}(\boldsymbol{x}; [\widehat{\mathbf{\Theta}}_u^{(1)}]_0) + \langle \nabla_{[\widehat{\mathbf{\Theta}}_u^{(1)}]_0} f_u^{(1)}(\boldsymbol{x}; [\widehat{\mathbf{\Theta}}_u^{(1)}]_0), \widehat{\mathbf{\Theta}}_u^{(1)} - [\widehat{\mathbf{\Theta}}_u^{(1)}]_0 \rangle|.$$

Here, we could bound the first term on the RHS with **Lemma** F.7. Applying **Lemma** F.8 on the second term, and recalling $\|\widehat{\mathbf{\Theta}}_u^{(1)} - [\mathbf{\Theta}_u^{(1)}]_0\|_2 \leq \omega$, would give

$$|f_u^{(1)}(\boldsymbol{x}; \widehat{\mathbf{\Theta}}_u^{(1)})| \leq 2 + \|\nabla_{[\widehat{\mathbf{\Theta}}_u^{(1)}]_0} f_u^{(1)}(\boldsymbol{x}; [\widehat{\mathbf{\Theta}}_u^{(1)}]_0)\|_2 \|\widehat{\mathbf{\Theta}}_u^{(1)} - [\mathbf{\Theta}_u^{(1)}]_0\|_2 +$$

$$\mathcal{O}(\omega^{1/3} L^2 \sqrt{m \log(m)}) \cdot \|\widehat{\mathbf{\Theta}}_u^{(1)} - [\mathbf{\Theta}_u^{(1)}]_0\|_2$$

$$\leq 2 + \mathcal{O}(L) \cdot \|\widehat{\mathbf{\Theta}}_u^{(1)} - [\mathbf{\Theta}_u^{(1)}]_0\|_2 + \mathcal{O}(L^2 \sqrt{m \log(m)})(\|\widehat{\mathbf{\Theta}}_u^{(1)} - [\mathbf{\Theta}_u^{(1)}]_0\|_2)^{\frac{4}{3}}.$$

Then, with $T_{u,t} = \frac{t}{n}$, applying the conclusion of **Lemma** F.6 would lead to

$$|f_u^{(1)}(\boldsymbol{x}; \widehat{\mathbf{\Theta}}_u^{(1)})| \leq 2 + \mathcal{O}(L) \cdot \mathcal{O}\left(\frac{(T_{u,t})^3}{\rho \sqrt{m}} \log m\right) + \mathcal{O}(L^2 \sqrt{m \log(m)}) \left(\mathcal{O}(\frac{(T_{u,t})^3}{\rho \sqrt{m}} \log m)\right)^{\frac{4}{3}}$$

$$= 2 + \mathcal{O}\left(\frac{t^3 L}{n^3 \rho \sqrt{m}} \log m\right) + \mathcal{O}\left(\frac{L^2 t^4}{n^4 \rho^{4/3} m^{1/6}} \log^{11/6}(m)\right) = \gamma_1.$$

$\square$

Then, under the assumption of arm separateness (**Assumption** 4.1), we proceed to bound the reward estimation error of the user exploitation network $f_u^{(1)}(\cdot; [\mathbf{\Theta}_u^{(1)}]_t)$ in the current round $t$.

**Lemma F.2.** *For the constants $\rho \in (0, \mathcal{O}(\frac{1}{L}))$ and $\xi_1 \in (0, 1)$, given user $u \in \mathcal{U}$ and its past records $\mathcal{P}_{u,t-1}$, we suppose $m, \eta_1, J_1$ satisfy the conditions in **Theorem** 4.2, and randomly draw the parameter $[\boldsymbol{\Theta}_u^{(1)}]_t \sim \{[\widehat{\boldsymbol{\Theta}}_u^{(1)}]_\tau\}_{\tau \in \mathcal{T}_{u,t}}$. Consider the past records $\mathcal{P}_{u,t}$ up to round $t$ are generated by a fixed policy when witness the candidate arms $\{\mathcal{X}_\tau\}_{\tau \in \mathcal{T}_{u,t}}$. Then, with probability at least $1 - \delta$ given an arm-reward pair $(\boldsymbol{x}_t, r_t)$, we have*

$$\sum_{\tau \in \mathcal{T}_{u,t} \cup \{t\}} \mathbb{E}\big[|f_u^{(1)}(\boldsymbol{x}_\tau; [\boldsymbol{\Theta}_u^{(1)}]_\tau) - r_\tau| \, | \, \mathcal{X}_\tau\big] \leq \sqrt{\frac{t}{n}} \cdot \left(\sqrt{2\xi_1} + \frac{3L}{\sqrt{2}} + (1 + \gamma_1)\sqrt{2 \log(\frac{tn \cdot a}{\delta})}\right)$$

*where $r_\tau$ is the corresponding reward generated by the reward mapping function given an arm $\boldsymbol{x}_\tau$.*

**Proof.** We proof this Lemma following a similar approach as in Lemma C.1 from (Ban et al., 2022b) and Lemma D.1 from (Ban et al., 2022a). First, for the LHS and with $\tau \in \mathcal{T}_{u,t} \cup \{t\}$, we have

$$|f_u^{(1)}(\boldsymbol{x}_t; [\widehat{\boldsymbol{\Theta}}_u^{(1)}]_\tau) - r_t| \leq |f_u^{(1)}(\boldsymbol{x}_t; [\widehat{\boldsymbol{\Theta}}_u^{(1)}]_\tau)| + |r_t| \leq 1 + \gamma_1$$

based on the conclusion from **Lemma** F.1. Then, for user $u$, we define the following martingale difference sequence with regard to the previous records $\mathcal{P}_{u,\tau}$ up to round $\tau$ as

$$V_\tau^{(1)} = \mathbb{E}[|f_u^{(1)}(\boldsymbol{x}_\tau; [\widehat{\boldsymbol{\Theta}}_u^{(1)}]_\tau) - r_\tau|] - |f_u^{(1)}(\boldsymbol{x}_\tau; [\widehat{\boldsymbol{\Theta}}_u^{(1)}]_\tau) - r_\tau|.$$

Since the records in set $\mathcal{P}_{u,\tau}$ are sharing the same reward mapping function, we have the expectation

$$\mathbb{E}[V_\tau^{(1)} | F_{u,\tau}] = \mathbb{E}[|f_u^{(1)}(\boldsymbol{x}_\tau; [\widehat{\boldsymbol{\Theta}}_u^{(1)}]_\tau) - r_\tau|] - \mathbb{E}[|f_u^{(1)}(\boldsymbol{x}_\tau; [\widehat{\boldsymbol{\Theta}}_u^{(1)}]_\tau) - r_\tau| | F_{u,\tau}] = 0.$$

where $F_{u,\tau}$ denotes the filtration given the past records $\mathcal{P}_{u,\tau}$. And we have the mean value of $V_\tau^{(1)}$ across different time steps as

$$\frac{1}{T_{u,t}} \sum_{\tau \in \mathcal{T}_{u,t}} [V_\tau^{(1)}] = \frac{1}{T_{u,t}} \sum_{\tau \in \mathcal{T}_{u,t}} \mathbb{E}[|f_u^{(1)}(\boldsymbol{x}_\tau; [\widehat{\boldsymbol{\Theta}}_u^{(1)}]_\tau) - r_\tau|] - \frac{1}{T_{u,t}} \sum_{\tau \in \mathcal{T}_{u,t}} |f_u^{(1)}(\boldsymbol{x}_\tau; [\widehat{\boldsymbol{\Theta}}_u^{(1)}]_\tau) - r_\tau|.$$

with the expectation of zero. Then, we proceed to bound the expected estimation error of the exploitation model with the estimation error from existing samples following the Proposition 1 from (Cesa-Bianchi et al., 2004). Applying the Azuma-Hoeffding inequality, with a constant $\delta \in (0, 1)$, it leads to

$$\mathbb{P}\bigg[\frac{1}{T_{u,t}} \sum_{\tau \in \mathcal{T}_{u,t}} \mathbb{E}[|f_u^{(1)}(\boldsymbol{x}_\tau; [\widehat{\boldsymbol{\Theta}}_u^{(1)}]_\tau) - r_t|] - \frac{1}{T_{u,t}} \sum_{\tau \in \mathcal{T}_{u,t}} |f_u^{(1)}(\boldsymbol{x}_\tau; [\widehat{\boldsymbol{\Theta}}_u^{(1)}]_\tau) - r_\tau|$$

$$\geq (1 + \gamma_1) \cdot \sqrt{\frac{2}{T_{u,t}} \ln(\frac{1}{\delta})}\bigg] \leq \delta.$$

As we have the parameter $[\boldsymbol{\Theta}_u^{(1)}]_t \sim \{[\widehat{\boldsymbol{\Theta}}_u^{(1)}]_\tau\}_{\tau \in \mathcal{T}_{u,t}}$, with the probability at least $1 - \delta$, the expected loss on $[\boldsymbol{\Theta}_u^{(1)}]_t$ could be bounded as

$$\frac{1}{T_{u,t}} \sum_{\tau \in \mathcal{T}_{u,t}} \mathbb{E}[|f_u^{(1)}(\boldsymbol{x}_t; [\widehat{\boldsymbol{\Theta}}_u^{(1)}]_\tau) - r_t|] \leq (1 + \gamma_1) \cdot \sqrt{\frac{2}{T_{u,t}} \ln(\frac{1}{\delta})} + \left(\frac{1}{T_{u,t}} \sum_{\tau \in \mathcal{T}_{u,t}} |f_u^{(1)}(\boldsymbol{x}_\tau; [\widehat{\boldsymbol{\Theta}}_u^{(1)}]_\tau) - r_\tau|\right)$$

where for the second term on the RHS, we have

$$\frac{1}{T_{u,t}} \sum_{\tau \in \mathcal{T}_{u,t}} |f_u^{(1)}(\boldsymbol{x}_\tau; [\widehat{\boldsymbol{\Theta}}_u^{(1)}]_\tau) - r_\tau| \leq \frac{1}{T_{u,t}} \sum_{\tau \in \mathcal{T}_{u,t}} |f_u^{(1)}(\boldsymbol{x}_\tau; [\widehat{\boldsymbol{\Theta}}_u^{(1)}]_t) - r_\tau| + \frac{3L\sqrt{2T_{u,t}}}{2} \cdot \frac{1}{T_{u,t}}$$

$$\leq \frac{1}{T_{u,t}} \sqrt{T_{u,t} \cdot \sum_{\tau \in \mathcal{T}_{u,t}} |f_u^{(1)}(\boldsymbol{x}_\tau; [\widehat{\boldsymbol{\Theta}}_u^{(1)}]_t) - r_\tau|^2} + \frac{3L}{\sqrt{2T_{u,t}}}$$

$$\leq \sqrt{\frac{\xi_1}{T_{u,t}}} + \frac{3L}{\sqrt{2T_{u,t}}}$$

where the first inequality is the application of **Lemma** F.10, and the last inequality is due to **Lemma** F.6. Summing up all the components and applying the union bound for all $a$ arms, all $n$ users and $t$ time steps would complete the proof. $\qquad \square$

Then, we also have the following Corollary for the rest of the candidate arms $\boldsymbol{x}_{i,t} \in (\mathcal{X}_t \setminus \{\boldsymbol{x}_t\})$.

**Corollary F.3.** *For the constants $\rho \in (0, \mathcal{O}(\frac{1}{L}))$ and $\xi_1 \in (0,1)$, given user $u \in \mathcal{U}$ and its past records $\mathcal{P}_{u,t-1}$, we suppose $m, \eta_1, J_1$ satisfy the conditions in **Theorem** 4.2, and randomly draw the parameter $[\boldsymbol{\Theta}_u^{(1)}]_t \sim \{[\widehat{\boldsymbol{\Theta}}_u^{(1)}]_\tau\}_{\tau \in \mathcal{T}_{u,t}}$. For an arm $\boldsymbol{x}_{i,t} \in \mathcal{X}_t$, consider its union set with the the collection of arms $\widetilde{\mathcal{P}}_{u,t} \cup \{\boldsymbol{x}_{i,t}, r_{i,t}\}$ are generated by a fixed policy when witness the candidate arms $\{\mathcal{X}_\tau\}_{\tau \in \mathcal{T}_{u,t}}$, with $\widetilde{\mathcal{P}}_{u,t} = \{\boldsymbol{x}_{i_\tau,\tau}, r_{i_\tau,\tau}\}_\tau$ being the collection of arms chosen by this policy. Then, with probability at least $1 - \delta$, we have*

$$\sum_{\tau \in \mathcal{T}_{u,t} \cup \{t\}} \mathbb{E}\big[|f_u^{(1)}(\boldsymbol{x}_{i_\tau,\tau}; [\boldsymbol{\Theta}_u^{(1)}]_t) - r_{i_\tau,\tau}|\,\big|\,\mathcal{X}_\tau\big] \le \sqrt{T_{u,t}} \cdot \left(\sqrt{2\xi_1} + \frac{3L}{\sqrt{2}} + (1+\gamma_1)\sqrt{2\log(\frac{tn \cdot a}{\delta})}\right) + \Gamma_t$$

*where $r_{i,\tau}$ is the corresponding reward generated by the mapping function given an arm $\boldsymbol{x}_{i,\tau}$, and*

$$\Gamma_t \le \left(1 + \mathcal{O}(\frac{tL^3 \log^{5/6}(m)}{n\rho^{1/3}m^{1/6}})\right) \cdot \mathcal{O}(\frac{t^4 L}{n^4 \rho\sqrt{m}}\log(m)) + \mathcal{O}\left(\frac{t^5 L^2 \log^{11/6}(m)}{n^5 \rho^{4/3} m^{1/6}}\right).$$

**Proof.** The proof of this Corollary follows an analogous approach as in Lemma F.2. First, suppose a shadow model $f_u^{(1)}(\cdot; [\widetilde{\boldsymbol{\Theta}}_u^{(1)}]_t)$, which is trained on the alternative trajectory $\widetilde{\mathcal{P}}_{u,t}$. Analogous to the proof of Lemma F.2, for user $u$, we can define the following martingale difference sequence with regard to the previous records $\widetilde{\mathcal{P}}_{u,\tau}$ up to round $\tau \in [t]$ as

$$\tilde{V}_\tau^{(1)} = \mathbb{E}[|f_u^{(1)}(\boldsymbol{x}_{i_\tau,\tau}; [\widetilde{\boldsymbol{\Theta}}_u^{(1)}]_\tau) - r_\tau|] - |f_u^{(1)}(\boldsymbol{x}_{i_\tau,\tau}; [\widetilde{\boldsymbol{\Theta}}_u^{(1)}]_\tau) - r_{i_\tau,\tau}|.$$

Since the records in set $\widetilde{\mathcal{P}}_{u,\tau}$ are sharing the same reward mapping function, we have the expectation

$$\mathbb{E}[\tilde{V}_\tau^{(1)}|\tilde{F}_{u,\tau}] = \mathbb{E}[|f_u^{(1)}(\boldsymbol{x}_{i_\tau,\tau}; [\widetilde{\boldsymbol{\Theta}}_u^{(1)}]_\tau) - r_{i_\tau,\tau}|] - \mathbb{E}[|f_u^{(1)}(\boldsymbol{x}_{i_\tau,\tau}; [\widetilde{\boldsymbol{\Theta}}_u^{(1)}]_\tau) - r_{i_\tau,\tau}|\,|\tilde{F}_{u,\tau}] = 0.$$

where $\tilde{F}_{u,\tau}$ denotes the filtration given the past records $\widetilde{\mathcal{P}}_{u,\tau}$. The mean value of $\tilde{V}_\tau^{(1)}$ across different time steps will be

$$\frac{1}{T_{u,t}}\sum_{\tau \in \mathcal{T}_{u,t}}[\tilde{V}_\tau^{(1)}] = \frac{1}{T_{u,t}}\sum_{\tau \in \mathcal{T}_{u,t}}\mathbb{E}[|f_u^{(1)}(\boldsymbol{x}_{i_\tau,\tau}; [\widetilde{\boldsymbol{\Theta}}_u^{(1)}]_\tau) - r_{i_\tau,\tau}|] - \frac{1}{T_{u,t}}\sum_{\tau \in \mathcal{T}_{u,t}}|f_u^{(1)}(\boldsymbol{x}_{i_\tau,\tau}; [\widetilde{\boldsymbol{\Theta}}_u^{(1)}]_\tau) - r_{i_\tau,\tau}|.$$

with the expectation of zero. Afterwards, applying the Azuma-Hoeffding inequality, with a constant $\delta \in (0,1)$, it leads to

$$\mathbb{P}\Bigg[\frac{1}{T_{u,t}}\sum_{\tau \in \mathcal{T}_{u,t}}\mathbb{E}[|f_u^{(1)}(\boldsymbol{x}_{i_\tau,\tau}; [\widetilde{\boldsymbol{\Theta}}_u^{(1)}]_\tau) - r_{i_\tau,\tau}|] - \frac{1}{T_{u,t}}\sum_{\tau \in \mathcal{T}_{u,t}}|f_u^{(1)}(\boldsymbol{x}_{i_\tau,\tau}; [\widetilde{\boldsymbol{\Theta}}_u^{(1)}]_\tau) - r_{i_\tau,\tau}|$$
$$\ge (1+\gamma_1) \cdot \sqrt{\frac{2}{T_{u,t}}\ln(\frac{1}{\delta})}\Bigg] \le \delta.$$

To bound the output difference between the shadow model $f_u^{(1)}(\cdot; [\widetilde{\boldsymbol{\Theta}}_u^{(1)}]_t)$ and the model we trained based on received records $f_u^{(1)}(\cdot; [\widehat{\boldsymbol{\Theta}}_u^{(1)}]_t)$, we apply the conclusion from **Lemma** G.14, which leads to that given the same input $\boldsymbol{x}$, we have

$$|f_u^{(1)}(\boldsymbol{x}; [\widetilde{\boldsymbol{\Theta}}_u^{(1)}]_t) - f_u^{(1)}(\boldsymbol{x}; [\widehat{\boldsymbol{\Theta}}_u^{(1)}]_t)| \le$$
$$\left(1 + \mathcal{O}(\frac{tL^3 \log^{5/6}(m)}{\rho^{1/3}m^{1/6}})\right) \cdot \mathcal{O}(\frac{t^3 L}{\rho\sqrt{m}}\log(m)) + \mathcal{O}\left(\frac{t^4 L^2 \log^{11/6}(m)}{\rho^{4/3}m^{1/6}}\right).$$

Finally, assembling all the components together will finish the proof.

$\square$

## F.2 USER EXPLORATION MODEL

To ensure the unit length of $f_u^{(2)}(\cdot)$'s input, we normalize the gradient $\frac{\nabla_{[\boldsymbol{\Theta}_u^{(1)}]_t} f_u^{(1)}(\boldsymbol{x};[\boldsymbol{\Theta}_u^{(1)}]_t)}{c'_g L}$ with **Lemma** F.8, **Lemma** F.9 and a normalization constant $c'_g > 0$. Then, to satisfy the separateness

(**Assumption** 4.1) assumption, we adopt the operation mentioned in **Eq.** 9 to derive the transformation $\phi(\frac{\nabla_{[\boldsymbol{\Theta}_u^{(1)}]_t} f_u^{(1)}(\boldsymbol{x};[\boldsymbol{\Theta}_u^{(1)}]_t)}{c_g' L}, \boldsymbol{x})$ to make sure the transformed input gradient is of the norm of 1, and the separateness of at least $\frac{\rho}{\sqrt{2}}$.

Analogous to the user exploitation model, regarding the convergence result for FC networks in **Lemma** F.6, we proceed to present the generalization result of the user exploration model $f_u^{(2)}(\cdot)$ after GD with the following lemma.

**Lemma F.4.** *For the constants $c_g' > 0$, $\rho \in (0, \mathcal{O}(\frac{1}{L}))$ and $\xi_1 \in (0,1)$, given user $u \in \mathcal{U}$ and its past records $\mathcal{P}_{u,t-1}$, we suppose $m, \eta_1, J_1$ satisfy the conditions in **Theorem** 4.2, and randomly draw the parameter $[\boldsymbol{\Theta}_u^{(2)}]_t \sim \{[\widehat{\boldsymbol{\Theta}}_u^{(2)}]_\tau\}_{\tau \in \mathcal{T}_{u,t}}$. Consider the past records $\mathcal{P}_{u,t}$ up to round $t$ are generated by a fixed policy when witness the candidate arms $\{\mathcal{X}_\tau\}_{\tau \in \mathcal{T}_{u,t}}$. Then, with probability at least $1 - \delta$ given an arm-reward pair $(\boldsymbol{x}_t, r_t)$, we have*

$$\sum_{\tau \in \mathcal{T}_{u,t} \cup \{t\}} \mathbb{E}\left[|f_u^{(2)}\left(\phi(\frac{\nabla_{[\boldsymbol{\Theta}_u^{(1)}]_\tau} f_u^{(1)}(\boldsymbol{x}_\tau;[\boldsymbol{\Theta}_u^{(1)}]_\tau)}{c_g' L}, \boldsymbol{x}_\tau);[\boldsymbol{\Theta}_u^{(2)}]_\tau\right) - \left(r_\tau - f_u^{(1)}(\boldsymbol{x}_\tau;[\boldsymbol{\Theta}_u^{(1)}]_\tau)\right)|\,|\mathcal{X}_\tau\right]$$

$$\leq \sqrt{T_{u,t}} \cdot \left(\sqrt{2\xi_1} + \frac{3L}{\sqrt{2}} + (1 + 2\gamma_1)\sqrt{2\log(\frac{tn \cdot a}{\delta})}\right)$$

**Proof.** The proof of this lemma is inspired by Lemma C.1 from (Ban et al., 2022b). Following the same procedure as in the proof of **Lemma** F.2, we bound

$$\left|f_u^{(2)}\left(\phi(\frac{\nabla_{[\boldsymbol{\Theta}_u^{(1)}]_t} f_u^{(1)}(\boldsymbol{x}_t;[\boldsymbol{\Theta}_u^{(1)}]_t)}{c_g' L}, \boldsymbol{x}_t);[\boldsymbol{\Theta}_u^{(2)}]_t\right) - \left(r_t - f_u^{(1)}(\boldsymbol{x}_t;[\boldsymbol{\Theta}_u^{(1)}]_t)\right)\right|$$

$$\leq \left|f_u^{(2)}\left(\phi(\frac{\nabla_{[\boldsymbol{\Theta}_u^{(1)}]_t} f_u^{(1)}(\boldsymbol{x}_t;[\boldsymbol{\Theta}_u^{(1)}]_t)}{c_g' L}, \boldsymbol{x}_t);[\boldsymbol{\Theta}_u^{(2)}]_t\right)\right| + \left|f_u^{(1)}(\boldsymbol{x}_t;[\boldsymbol{\Theta}_u^{(1)}]_t)\right| + 1$$

$$\leq 1 + 2\gamma_1$$

by triangle inequality and applying the generalization result of FC networks (**Lemma** F.1) on $f_u^{(1)}(\cdot;\boldsymbol{\Theta}_u^{(1)}), f_u^{(2)}(\cdot;\boldsymbol{\Theta}_u^{(2)})$.

For brevity, we use $\nabla f_{u,\tau}^{(1)}(\boldsymbol{x}_t)$ to denote $\phi(\frac{\nabla_{[\boldsymbol{\Theta}_u^{(1)}]_\tau} f_u^{(1)}(\boldsymbol{x}_t;[\boldsymbol{\Theta}_u^{(1)}]_\tau)}{c_g' L}, \boldsymbol{x}_t)$ for the following proof. Define the difference sequence as

$$V_\tau^{(2)} = \mathbb{E}\left[\left|f_u^{(2)}\left(\nabla f_{u,\tau}^{(1)}(\boldsymbol{x}_\tau);[\boldsymbol{\Theta}_u^{(2)}]_\tau\right) - \left(r_\tau - f_u^{(1)}(\boldsymbol{x}_\tau;[\boldsymbol{\Theta}_u^{(1)}]_\tau)\right)\right|\right]$$

$$- \left|f_u^{(2)}\left(\nabla f_{u,\tau}^{(1)}(\boldsymbol{x}_\tau);[\boldsymbol{\Theta}_u^{(2)}]_\tau\right) - \left(r_\tau - f_u^{(1)}(\boldsymbol{x}_\tau;[\boldsymbol{\Theta}_u^{(1)}]_\tau)\right)\right|.$$

Since the reward mapping is fixed given the specific user $u$, which means that the past rewards and the received arm-reward pairs $(\boldsymbol{x}_\tau, r_\tau)$ are generated by the same reward mapping function, we have the expectation

$$\mathbb{E}[V_\tau^{(2)}|F_{u,\tau}] = \mathbb{E}\left[\left|f_u^{(2)}\left(\nabla f_{u,\tau}^{(1)}(\boldsymbol{x}_\tau);[\boldsymbol{\Theta}_u^{(2)}]_\tau\right) - \left(r_\tau - f_u^{(1)}(\boldsymbol{x}_\tau;[\boldsymbol{\Theta}_u^{(1)}]_\tau)\right)\right|\right]$$

$$- \mathbb{E}\left[\left|f_u^{(2)}\left(\nabla f_{u,\tau}^{(1)}(\boldsymbol{x}_\tau);[\boldsymbol{\Theta}_u^{(2)}]_\tau\right) - \left(r_\tau - f_u^{(1)}(\boldsymbol{x}_\tau;[\boldsymbol{\Theta}_u^{(1)}]_\tau)\right)\right|\,\Big|F_{u,\tau}\right] = 0.$$

where $F_{u,\tau}$ denotes the filtration given the past records $\mathcal{P}_{u,\tau}$, up to round $\tau \in [t]$. This also gives the fact that $V_\tau^{(2)}$ is a martingale difference sequence. Then, after applying the martingale difference sequence over $\mathcal{T}_{u,t}$, we have

$$\frac{1}{T_{u,t}}\sum_{\tau \in \mathcal{T}_{u,t}} V_\tau^{(2)} = \frac{1}{T_{u,t}}\sum_{\tau \in \mathcal{T}_{u,t}} \mathbb{E}\left[\left|f_u^{(2)}\left(\nabla f_{u,\tau}^{(1)}(\boldsymbol{x}_\tau);[\boldsymbol{\Theta}_u^{(2)}]_\tau\right) - \left(r_\tau - f_u^{(1)}(\boldsymbol{x}_\tau;[\boldsymbol{\Theta}_u^{(1)}]_\tau)\right)\right|\right]$$

$$- \frac{1}{T_{u,t}}\sum_{\tau \in \mathcal{T}_{u,t}} \left|f_u^{(2)}\left(\nabla f_{u,\tau}^{(1)}(\boldsymbol{x}_\tau);[\boldsymbol{\Theta}_u^{(2)}]_\tau\right) - \left(r_\tau - f_u^{(1)}(\boldsymbol{x}_\tau;[\boldsymbol{\Theta}_u^{(1)}]_\tau)\right)\right|.$$

Analogous to the proof of **Lemma** F.2, by applying the Azuma-Hoeffding inequality, it leads to

$$\mathbb{P}\left[\frac{1}{T_{u,t}}\sum_{\tau\in\mathcal{T}_{u,t}}V_\tau^{(2)}-\frac{1}{t}\sum_{\tau\in\mathcal{T}_{u,t}}\mathbb{E}[V_\tau^{(2)}]\geq(1+2\gamma_1)\sqrt{\frac{2\log(1/\delta)}{T_{u,t}}}\right]\leq\delta$$

Since the expectation of $V_\tau^{(2)}$ is zero, with the probability at least $1-\delta$ and an existing set of parameters $\widetilde{\boldsymbol{\Theta}}_u^{(2)}$ s.t. $\|\widetilde{\boldsymbol{\Theta}}_u^{(2)}-[\boldsymbol{\Theta}_u^{(2)}]_\tau\|\leq\mathcal{O}\left(\frac{t^3}{n^3\rho\sqrt{m}}\log m\right)$, the above inequality implies

$$\frac{1}{T_{u,t}}\sum_{\tau\in\mathcal{T}_{u,t}}V_\tau^{(2)}\leq(1+2\gamma_1)\sqrt{\frac{2\log(1/\delta)}{T_{u,t}}}\implies$$

$$\frac{1}{T_{u,t}}\sum_{\tau\in\mathcal{T}_{u,t}}\mathbb{E}\left[\left|f_u^{(2)}\left(\nabla f_{u,\tau}^{(1)}(\boldsymbol{x}_t);[\boldsymbol{\Theta}_u^{(2)}]_\tau\right)-\left(r_t-f_u^{(1)}(\boldsymbol{x}_t;[\boldsymbol{\Theta}_u^{(1)}]_\tau)\right)\right|\right]$$

$$\leq\frac{1}{T_{u,t}}\sum_{\tau\in\mathcal{T}_{u,t}}\left|f_u^{(2)}\left(\nabla f_{u,\tau}^{(1)}(\boldsymbol{x}_\tau);[\boldsymbol{\Theta}_u^{(2)}]_\tau\right)-\left(r_\tau-f_u^{(1)}(\boldsymbol{x}_\tau;[\boldsymbol{\Theta}_u^{(1)}]_\tau)\right)\right|+(1+2\gamma_1)\sqrt{\frac{2\log(1/\delta)}{T_{u,t}}}$$

$$\underset{\text{(i)}}{\leq}\frac{1}{T_{u,t}}\sum_{\tau\in\mathcal{T}_{u,t}}\left|f_u^{(2)}\left(\nabla f_{u,\tau}^{(1)}(\boldsymbol{x}_\tau);\widetilde{\boldsymbol{\Theta}}_u^{(2)}\right)-\left(r_\tau-f_u^{(1)}(\boldsymbol{x}_\tau;[\boldsymbol{\Theta}_u^{(1)}]_\tau)\right)\right|+(1+2\gamma_1)\sqrt{\frac{2\log(1/\delta)}{T_{u,t}}}$$

$$\leq\frac{1}{\sqrt{T_{u,t}}}\sqrt{\sum_{\tau\in\mathcal{T}_{u,t}}\left|f_u^{(2)}\left(\nabla f_{u,\tau}^{(1)}(\boldsymbol{x}_\tau);\widetilde{\boldsymbol{\Theta}}_u^{(2)}\right)-\left(r_\tau-f_u^{(1)}(\boldsymbol{x}_\tau;[\boldsymbol{\Theta}_u^{(1)}]_\tau)\right)\right|^2}+(1+2\gamma_1)\sqrt{\frac{2\log(1/\delta)}{T_{u,t}}}$$

$$\underset{\text{(ii)}}{\leq}\sqrt{\frac{2\xi_1}{T_{u,t}}}+(1+2\gamma_1)\sqrt{\frac{2\log(1/\delta)}{T_{u,t}}}.$$

Here, the upper bound (i) is derived by applying the conclusions of **Lemma** F.6 and **Lemma** F.10, and the inequality (ii) is derived by adopting **Lemma** F.6 while defining the empirical loss to be $\frac{1}{2}\sum_{\tau\in\mathcal{T}_{u,t}}\left|f_u^{(2)}\left(\nabla f_{u,\tau}^{(1)}(\boldsymbol{x}_\tau);\widetilde{\boldsymbol{\Theta}}_u^{(2)}\right)-\left(r_\tau-f_u^{(1)}(\boldsymbol{x}_\tau;[\boldsymbol{\Theta}_u^{(1)}]_\tau)\right)\right|^2$. Finally, applying the union bound would give the aforementioned results.

$\square$

**Corollary F.5.** *For the constants $\rho\in(0,\mathcal{O}(\frac{1}{L}))$ and $\xi_1\in(0,1)$, given user $u\in\mathcal{U}$ and its past records $\mathcal{P}_{u,t-1}$, we suppose $m,\eta_1,J_1$ satisfy the conditions in **Theorem** 4.2, and randomly draw the parameter $[\boldsymbol{\Theta}_u^{(1)}]_t\sim\{[\widehat{\boldsymbol{\Theta}}_u^{(1)}]_\tau\}_{\tau\in\mathcal{T}_{u,t}}$. For an arm $\boldsymbol{x}_{i,t}\in\mathcal{X}_t$, consider its union set with the the collection of arms $\widetilde{\mathcal{P}}_{u,t}\cup\{\boldsymbol{x}_{i,t},r_{i,t}\}$ are generated by a fixed policy when witness the candidate arms $\{\mathcal{X}_\tau\}_{\tau\in\mathcal{T}_{u,t}}$, with $\widehat{\mathcal{P}}_{u,t}=\{\boldsymbol{x}_{i_\tau,\tau},r_{i_\tau,\tau}\}_\tau$ being the collection of arms chosen by this policy. Then, with probability at least $1-\delta$, we have*

$$\sum_{\tau\in\mathcal{T}_{u,t}\cup\{t\}}\mathbb{E}\left[|f_u^{(2)}\left(\phi(\frac{\nabla_{[\boldsymbol{\Theta}_u^{(1)}]_\tau}f_u^{(1)}(\boldsymbol{x}_{i_\tau,\tau};[\boldsymbol{\Theta}_u^{(1)}]_\tau)}{c_g'L},\boldsymbol{x}_{i_\tau,\tau});[\boldsymbol{\Theta}_u^{(2)}]_\tau\right)-\left(r_{i_\tau,\tau}-f_u^{(1)}(\boldsymbol{x}_{i_\tau,\tau};[\boldsymbol{\Theta}_u^{(1)}]_\tau)\right)|\,|\,\mathcal{X}_\tau\right]$$

$$\leq\sqrt{T_{u,t}}\cdot\left(\sqrt{2\xi_1}+\frac{3L}{\sqrt{2}}+(1+\gamma_1)\sqrt{2\log(\frac{tn\cdot a}{\delta})}\right)+\Gamma_t$$

*where $r_{i_\tau,\tau}$ is the corresponding reward generated by the mapping function given an arm $\boldsymbol{x}_{i_\tau,\tau}$, and*

$$\Gamma_t=\left(1+\mathcal{O}(\frac{tL^3\log^{5/6}(m)}{n\rho^{1/3}m^{1/6}})\right)\cdot\mathcal{O}(\frac{t^4L}{n^4\rho\sqrt{m}}\log(m))+\mathcal{O}\left(\frac{t^5L^2\log^{11/6}(m)}{n^5\rho^{4/3}m^{1/6}}\right).$$

This corollary is the direct application of Lemma F.4, and the proof is analogous to that of Corollary F.3.

### F.3    LEMMAS FOR OVER-PARAMETERIZED USER NETWORKS

Applying $\mathcal{P}_{u,t-1}$ as the training data, we have the following convergence result for the user exploitation network $f_u^{(1)}(\cdot; \boldsymbol{\Theta}_u^{(1)})$ after GD.

**Lemma F.6** (Theorem 1 from (Allen-Zhu et al., 2019)). *For any $0 < \xi_1 \le 1$, $0 < \rho \le \mathcal{O}(\frac{1}{L})$. Given user $u \in \mathcal{U}$ and its past records $\mathcal{P}_{u,t-1}$, suppose $m, \eta_1, J_1$ satisfy the conditions in **Theorem 4.2**, then with probability at least $1 - \delta$, we could have*

1. *$\mathcal{L}(\boldsymbol{\Theta}_u^{(1)}) \le \xi_1$ after $J_1$ iterations of GD.*

2. *For any $j \in [J_1]$, $\|[\boldsymbol{\Theta}_u^{(1)}]^j - [\boldsymbol{\Theta}_u^{(1)}]^0\| \le \mathcal{O}\left(\frac{(T_{u,t})^3}{\rho\sqrt{m}} \log m\right) = \mathcal{O}\left(\frac{t^3}{n^3\rho\sqrt{m}} \log m\right)$.*

In particular, **Lemma** F.6 above provides the convergence guarantee for $f_u^{(1)}(\cdot; \boldsymbol{\Theta}_u^{(1)})$ after certain rounds of GD training on the past records $\mathcal{P}_{u,t-1}$.

**Lemma F.7** (Lemma 4.1 in (Cao & Gu, 2019)). *Assume a constant $\omega$ such that $\mathcal{O}(m^{-3/2}L^{-3/2}[\log(TnL^2/\delta)]^{3/2}) \le \omega \le \mathcal{O}(L^{-6}[\log m]^{-3/2})$ and $n$ training samples. With randomly initialized $[\boldsymbol{\Theta}_u^{(1)}]_0$, for parameters $\boldsymbol{\Theta}, \boldsymbol{\Theta}'$ satisfying $\|\boldsymbol{\Theta} - [\boldsymbol{\Theta}_u^{(1)}]_0\|, \|\boldsymbol{\Theta} - [\boldsymbol{\Theta}_u^{(1)}]_0\| \le \omega$, we have*

$$|f_u^{(1)}(\boldsymbol{x};\boldsymbol{\Theta}) - f_u^{(1)}(\boldsymbol{x};\boldsymbol{\Theta}') - \langle\nabla_{\boldsymbol{\Theta}'} f_u^{(1)}(\boldsymbol{x};\boldsymbol{\Theta}'), \boldsymbol{\Theta} - \boldsymbol{\Theta}'\rangle| \le \mathcal{O}(\omega^{1/3}L^2\sqrt{m\log(m)})\|\boldsymbol{\Theta} - \boldsymbol{\Theta}'\|$$

*with the probability at least $1 - \delta$.*

**Lemma F.8.** *Assume $m, \eta_1, J_1$ satisfy the conditions in **Theorem** 4.2 and $[\boldsymbol{\Theta}_u^{(1)}]_0$ being randomly initialized. Then, with probability at least $1 - \delta$ and given an arm $\|\boldsymbol{x}\|_2 = 1$, we have*

1. *$|f_u^{(1)}(\boldsymbol{x}; [\boldsymbol{\Theta}_u^{(1)}]_0)| \le 2$,*

2. *$\|\nabla_{[\boldsymbol{\Theta}_u^{(1)}]_0} f_u^{(1)}(\boldsymbol{x}; [\boldsymbol{\Theta}_u^{(1)}]_0)\|_2 \le \mathcal{O}(L)$.*

**Proof.** The conclusion (1) is a direct application of Lemma 7.1 in (Allen-Zhu et al., 2019). For conclusion (2), applying Lemma 7.3 in (Allen-Zhu et al., 2019), for each layer $\boldsymbol{\Theta}_l \in \{\boldsymbol{\Theta}_1, \dots, \boldsymbol{\Theta}_L\}$, we have

$$\|\nabla_{\boldsymbol{\Theta}_l} f_u^{(1)}(\boldsymbol{x}; [\boldsymbol{\Theta}_u^{(1)}]_0)\|_2 = \|(\boldsymbol{\Theta}_L \boldsymbol{D}_{L-1} \cdots \boldsymbol{D}_{l+1}\boldsymbol{\Theta}_{l+1}) \cdot (\boldsymbol{D}_{l+1}\boldsymbol{\Theta}_{l+1} \cdots \boldsymbol{D}_1\boldsymbol{\Theta}_1) \cdot \boldsymbol{x}^{\intercal}\|_2 = \mathcal{O}(\sqrt{L}).$$

Then, we could have the conclusion that

$$\|\nabla_{[\boldsymbol{\Theta}_u^{(1)}]_0} f_u^{(1)}(\boldsymbol{x}; [\boldsymbol{\Theta}_u^{(1)}]_0)\|_2 = \sqrt{\sum_{l \in [L]} \|\nabla_{\boldsymbol{\Theta}_l} f_u^{(1)}(\boldsymbol{x}; [\boldsymbol{\Theta}_u^{(1)}]_0)\|_2^2} = \mathcal{O}(L).$$

$\square$

**Lemma F.9** (Theorem 5 in (Allen-Zhu et al., 2019)). *Assume $m, \eta_1, J_1$ satisfy the conditions in **Theorem** 4.2 and $[\boldsymbol{\Theta}_u^{(1)}]_0$ being randomly initialized. Then, with probability at least $1 - \delta$, and for all parameter $\boldsymbol{\Theta}_u^{(1)}$ such that $\|\boldsymbol{\Theta}_u^{(1)} - [\boldsymbol{\Theta}_u^{(1)}]_0\|_2 \le \omega$, we have*

$$\|\nabla_{\boldsymbol{\Theta}_u^{(1)}} f_u^{(1)}(\boldsymbol{x}; \boldsymbol{\Theta}_u^{(1)}) - \nabla_{[\boldsymbol{\Theta}_u^{(1)}]_0} f_u^{(1)}(\boldsymbol{x}; [\boldsymbol{\Theta}_u^{(1)}]_0)\|_2 \le \mathcal{O}(\omega^{1/3}L^3\sqrt{\log(m)})$$

**Lemma F.10.** *Assume $m, \eta_1$ satisfy the condition in **Theorem** 4.2. With the probability at least $1 - \delta$, we have*

$$\sum_{\tau \in \mathcal{T}_{u,t}} |f_u^{(1)}(\boldsymbol{x}_\tau; [\widehat{\boldsymbol{\Theta}}_u^{(1)}]_\tau) - r_\tau| \le \sum_{\tau \in \mathcal{T}_{u,t}} |f_u^{(1)}(\boldsymbol{x}_\tau; [\widehat{\boldsymbol{\Theta}}_u^{(1)}]_t) - r_\tau| + \frac{3L\sqrt{2T_{u,t}}}{2}$$

**Proof.** With the notation from Lemma 4.3 in (Cao & Gu, 2019), set $R = \frac{T_{u,t}^3 \log(m)}{\delta}$, $\nu = R^2$, and $\epsilon = \frac{LR}{\sqrt{2\nu T_{u,t}}}$. Then, considering the loss function to be $\mathcal{L}(\boldsymbol{\Theta}_u^{(1)}) := \sum_{\tau \in \mathcal{T}_{u,t}} |f_u^{(1)}(\boldsymbol{x}_\tau; \boldsymbol{\Theta}_u^{(1)}) - r_\tau|$ would complete the proof. $\square$

## G    PROOF OF THE REGRET BOUND

In this section, we present the generalization results of GNN models $f_{gnn}^{(1)}(\cdot; \boldsymbol{\Theta}_{gnn}^{(1)}), f_{gnn}^{(2)}(\cdot; \boldsymbol{\Theta}_{gnn}^{(2)})$. Recall that up to round $t$, we have all the past arm-reward pairs $\mathcal{P}_t = \{(\boldsymbol{x}_\tau, r_\tau)\}_{\tau \in [t-1]}$ for the previous $t-1$ time steps. Analogous to the generalization analysis of user models in Section F, we adopt the the operation in **Eq.** 9 on the gradients $\nabla_{\boldsymbol{\Theta}_{gnn}^{(1)}} f_{gnn}^{(1)}(\cdot; \boldsymbol{\Theta}_{gnn}^{(1)})$ to comply with the assumptions of unit-length and separateness, and the transformed gradient input is denoted as $\nabla f^{(1)}(\boldsymbol{x})$ given the arm $\boldsymbol{x}$.

### G.1    BOUNDING THE PARAMETER ESTIMATION ERROR

Regarding **Eq.**8, given an arbitrary candidate arm $\boldsymbol{x} \in \mathcal{X}_t$ with its reward $r$, and its user graphs $\mathcal{G}^{(1)}, \mathcal{G}^{(2)}$, we have the bound for the estimation error as

$$\mathsf{CB}_t(\boldsymbol{x}) = \mathbb{E}\left[|f_{gnn}^{(2)}(\nabla f_t^{(1)}(\boldsymbol{x}), \mathcal{G}^{(2)}; [\boldsymbol{\Theta}_{gnn}^{(2)}]_{t-1}) - (r_t - f_{gnn}^{(1)}(\boldsymbol{x}, \mathcal{G}^{(1)}; [\boldsymbol{\Theta}_{gnn}^{(1)}]_{t-1}))|\Big| u_t, \mathcal{X}_t\right]$$

$$\leq \underbrace{\mathbb{E}\left[|f_{gnn}^{(2)}(\nabla f_t^{(1),*}(\boldsymbol{x}), \mathcal{G}^{(2),*}; [\boldsymbol{\Theta}_{gnn}^{(2)}]_{t-1}) - (r_t - f_{gnn}^{(1)}(\boldsymbol{x}, \mathcal{G}^{(1),*}; [\boldsymbol{\Theta}_{gnn}^{(1)}]_{t-1}))|\Big| u_t, \mathcal{X}_t\right]}_{I_1}$$

$$+ I_2 + I_3 + I_4$$

where we have the term $I_1$ representing the estimation error induced by the GNN model parameters $\{[\boldsymbol{\Theta}_{gnn}^{(1)}]_{t-1}, [\boldsymbol{\Theta}_{gnn}^{(2)}]_{t-1}\}$. Based on our arm selected strategy given in **Algorithm** 1, we have the selected arms and their rewards $\{\boldsymbol{x}_\tau, r_\tau\}_{\tau \in [t-1]}$ up to round $t$. And we first proceed to bound term $I_1$ w.r.t. the selected arm $\boldsymbol{x}_t$, i.e., $\mathsf{CB}_t(\boldsymbol{x}_t)$.

Analogous to the user-specific models, we also have bounded outputs for the GNN models shown in the following lemma.

**Lemma G.1.** *For the constants $\rho \in (0, \mathcal{O}(\frac{1}{L}))$ and $\xi_2 \in (0, 1)$, the past records $\mathcal{P}_t$ up to time step $t$, we suppose $m, \eta_1, \eta_2, J_1, J_2$ satisfy the conditions in **Theorem** 4.2. Then, with probability at least $1 - \delta$ and given an arm-reward pair $(\boldsymbol{x}, r)$, we have*

$$|f_{gnn}^{(1)}(\boldsymbol{x}; [\widehat{\boldsymbol{\Theta}}_{gnn}^{(1)}]_t)| \leq \gamma_2$$

*where*

$$\gamma_2 = 2 + \mathcal{O}\left(\frac{t^3 L}{\rho\sqrt{m}} \log m\right) + \mathcal{O}\left(\frac{L^2 t^4}{\rho^{4/3} m^{1/6}} \log^{11/6}(m)\right).$$

**Proof.** The proof of this lemma follows an analogous approach as in **Lemma** F.1 where we have proved the conclusion for the FC networks.

Given an arm $\boldsymbol{x}$, we denote the adjacency matrix of its estimated user graph $\mathcal{G}^{(1)}$ as $\boldsymbol{A}^{(1)}$, and we have the normalized adjacency matrices as $\boldsymbol{S}^{(1)} = \boldsymbol{A}^{(1)}/n$. For the received user $u_t \in \mathcal{U}$, we could deem the corresponding row of the matrix multiplication $\boldsymbol{S} \cdot \boldsymbol{X}$, represented by $\boldsymbol{h}_{u_t} = [\boldsymbol{S} \cdot \boldsymbol{X}]_{i:}$, as the aggregated input for the network for the user-arm pair $(\boldsymbol{x}, u_t)$. Note that in this way, the rest of the network could be regarded as a $L+1$-layer FC network (one layer GNN + $L$-layer FC network), where the weight matrix of the first layer is $\boldsymbol{\Theta}_{agg}^{(1)}$. Then, to make sure each aggregated input has the norm of 1, we apply an additional transformation mentioned in **Eq.** 9 as $\tilde{\boldsymbol{h}}_{u_t} = \phi(\boldsymbol{h}_{u_t}, \boldsymbol{x}) = (\frac{\boldsymbol{h}_{u_t}}{\sqrt{2}}, \frac{\boldsymbol{x}}{2}, c_{u_t})$ where $c_{u_t} = \sqrt{\frac{3}{4} - \frac{1}{2}\|\boldsymbol{h}_{u_t}\|_2^2}$. This transformation ensures $\|\tilde{\boldsymbol{h}}_{u_t}\|_2 = 1$ while preserving the original information w.r.t. the user-arm pair $(\boldsymbol{x}, u_t)$, as it does not change the original aggregated hidden representation. Meantime, this transformation also ensures the separateness of the transformed contexts to be at least $\frac{\rho}{2}$, which would fit the original data separateness assumption (**Assumption** 4.1). Finally, following a similar approach as in the FC networks (**Lemma** F.1), on the transformed aggregated hidden representations would complete the proof.

$\square$

Regarding the definition for the true reward mapping function in Section 2, we have the following lemma for term $I_1$ given the arm-reward pair $(\boldsymbol{x}_t, r_t)$.

**Lemma G.2.** *For the constants $\rho \in (0, \mathcal{O}(\frac{1}{L}))$ and $\xi_2 \in (0, 1)$, given user $u \in \mathcal{U}$ and its past records $\mathcal{P}_{u,t}$, we suppose $m, \eta_1, \eta_2, J_1, J_2$ satisfy the conditions in **Theorem** 4.2, and randomly draw the parameters $[\boldsymbol{\Theta}_{gnn}^{(1)}]_t \sim \{[\widehat{\boldsymbol{\Theta}}_{gnn}^{(1)}]_\tau\}_{\tau \in [t]}$, $[\boldsymbol{\Theta}_{gnn}^{(2)}]_t \sim \{[\widehat{\boldsymbol{\Theta}}_{gnn}^{(2)}]_\tau\}_{\tau \in [t]}$. Then, with probability at least $1 - \delta$ given a sampled arm-reward pair $(\boldsymbol{x}, r)$, we have*

$$\sum_{\tau \in [t]} \mathbb{E}\left[\left|f_{gnn}^{(2)}\left(\nabla f_t^{(1),*}(\boldsymbol{x}_t), \mathcal{G}_t^{(2),*}; [\boldsymbol{\Theta}_{gnn}^{(2)}]_{t-1}\right) - \left(r_t - f_{gnn}^{(1)}(\boldsymbol{x}_t, \mathcal{G}_t^{(1),*}; [\boldsymbol{\Theta}_{gnn}^{(1)}]_{t-1})\right)\right| \Big| u_t, \mathcal{X}_t\right]$$

$$\leq \sqrt{t} \cdot \left(\sqrt{2\xi_2} + \frac{3L}{\sqrt{2}} + (1 + \gamma_2)\sqrt{2\log(\frac{tn \cdot a}{\delta})}\right)$$

*where*

$$\gamma_2 = 2 + \mathcal{O}\left(\frac{t^3 L}{\rho\sqrt{m}}\log m\right) + \mathcal{O}\left(\frac{L^2 t^4}{\rho^{4/3} m^{1/6}}\log^{11/6}(m)\right).$$

**Proof.** Based on the conclusion of **Lemma** G.1, we have the upper bound as

$$\left|f_{gnn}^{(2)}\left(\nabla f_t^{(1),*}(\boldsymbol{x}_t), \mathcal{G}_t^{(2),*}; [\boldsymbol{\Theta}_{gnn}^{(2)}]_{t-1}\right) - \left(r_t - f_{gnn}^{(1)}(\boldsymbol{x}_t, \mathcal{G}_t^{(1),*}; [\boldsymbol{\Theta}_{gnn}^{(1)}]_{t-1})\right)\right| \leq 1 + 2\gamma_2$$

by simply using the triangular inequality. Then we proceed to define the sequence $V_\tau, \tau \in [t]$ as

$$V_\tau = \mathbb{E}_{\mathcal{X}_\tau}\left[\left|f_{gnn}^{(2)}(\nabla f_\tau^{(1),*}(\boldsymbol{x}_\tau), \mathcal{G}_\tau^{(2),*}; [\boldsymbol{\Theta}_{gnn}^{(2)}]_{\tau-1}) - (r_\tau - f_{gnn}^{(1)}(\boldsymbol{x}_\tau, \mathcal{G}_\tau^{(1),*}; [\boldsymbol{\Theta}_{gnn}^{(1)}]_{\tau-1}))\right|\right]$$
$$- \left|f_{gnn}^{(2)}(\nabla f_\tau^{(1),*}(\boldsymbol{x}_\tau), \mathcal{G}_\tau^{(2),*}; [\boldsymbol{\Theta}_{gnn}^{(2)}]_{\tau-1}) - (r_\tau - f_{gnn}^{(1)}(\boldsymbol{x}_\tau, \mathcal{G}_\tau^{(1),*}; [\boldsymbol{\Theta}_{gnn}^{(1)}]_{\tau-1}))\right|.$$

And since the candidate arms and the corresponding rewards are associated with the same reward mapping function $h(\cdot)$, the sequence $V_\tau$ is a martingale difference sequence with the expectation

$$\mathbb{E}[V_\tau | F_\tau] = \mathbb{E}_{\mathcal{X}_\tau}\left[\left|f_{gnn}^{(2)}(\nabla f_\tau^{(1),*}(\boldsymbol{x}_\tau), \mathcal{G}_\tau^{(2),*}; [\boldsymbol{\Theta}_{gnn}^{(2)}]_{\tau-1}) - (r_\tau - f_{gnn}^{(1)}(\boldsymbol{x}_\tau, \mathcal{G}_\tau^{(1),*}; [\boldsymbol{\Theta}_{gnn}^{(1)}]_{\tau-1}))\right|\right]$$
$$- \mathbb{E}_{\mathcal{X}_\tau}\left[\left|f_{gnn}^{(2)}(\nabla f_\tau^{(1),*}(\boldsymbol{x}_\tau), \mathcal{G}_\tau^{(2),*}; [\boldsymbol{\Theta}_{gnn}^{(2)}]_{\tau-1}) - (r_\tau - f_{gnn}^{(1)}(\boldsymbol{x}_\tau, \mathcal{G}_\tau^{(1),*}; [\boldsymbol{\Theta}_{gnn}^{(1)}]_{\tau-1}))\right|\right] = 0.$$

where $F_\tau$ denotes the filtration of all the past records $\mathcal{P}_\tau$ up to time step $\tau$. Then, we will have the mean value for this sequence as

$$\frac{1}{t}\sum_{\tau \in [t]} V_\tau =$$
$$\frac{1}{t}\sum_{\tau \in [t]} \mathbb{E}_{\mathcal{X}_\tau}\left[\left|f_{gnn}^{(2)}(\nabla f_\tau^{(1),*}(\boldsymbol{x}_\tau), \mathcal{G}_\tau^{(2),*}; [\boldsymbol{\Theta}_{gnn}^{(2)}]_{\tau-1}) - (r_\tau - f_{gnn}^{(1)}(\boldsymbol{x}_\tau, \mathcal{G}_\tau^{(1),*}; [\boldsymbol{\Theta}_{gnn}^{(1)}]_{\tau-1}))\right|\right]$$
$$- \frac{1}{t}\sum_{\tau \in [t]}\left|f_{gnn}^{(2)}(\nabla f_\tau^{(1),*}(\boldsymbol{x}_\tau), \mathcal{G}_\tau^{(2),*}; [\boldsymbol{\Theta}_{gnn}^{(2)}]_{\tau-1}) - (r_\tau - f_{gnn}^{(1)}(\boldsymbol{x}_\tau, \mathcal{G}_\tau^{(1),*}; [\boldsymbol{\Theta}_{gnn}^{(1)}]_{\tau-1}))\right|.$$

As it has shown that the sequence is a martingale difference sequence, by directly applying the Azuma-Hoeffding inequality, we could bound the difference between the mean and its expectation as

$$\mathbb{P}\left[\frac{1}{t}\sum_{\tau \in [t]} V_\tau - \frac{1}{t}\sum_{\tau \in [t]} \mathbb{E}[V_\tau] \geq (1 + 2\gamma_2)\sqrt{\frac{2\log(1/\delta)}{t}}\right] \leq \delta$$

with the probability at least $1 - 2\delta$. Since it has shown that the $V_\tau$ is of zero expectation, we have the second term on the LHS of the inequality to be zero. Then, the inequality above is equivalent to

$$\frac{1}{t} \sum_{\tau \in [t]} V_\tau \le (1 + 2\gamma_2)\sqrt{\frac{2\log(1/\delta)}{t}} \implies$$

$$\frac{1}{t} \sum_{\tau \in [t]} \mathbb{E}_{\mathcal{X}_\tau}\left[\left|f_{gnn}^{(2)}(\nabla f_\tau^{(1),*}(\boldsymbol{x}_\tau), \mathcal{G}_\tau^{(2),*}; [\boldsymbol{\Theta}_{gnn}^{(2)}]_{\tau-1}) - (r_\tau - f_{gnn}^{(1)}(\boldsymbol{x}_\tau, \mathcal{G}_\tau^{(1),*}; [\boldsymbol{\Theta}_{gnn}^{(1)}]_{\tau-1}))\right|\right]$$

$$\le \frac{1}{t} \sum_{\tau \in [t]} \left|f_{gnn}^{(2)}(\nabla f_\tau^{(1),*}(\boldsymbol{x}_\tau), \mathcal{G}_\tau^{(2),*}; [\boldsymbol{\Theta}_{gnn}^{(2)}]_{\tau-1}) - (r_\tau - f_{gnn}^{(1)}(\boldsymbol{x}_\tau, \mathcal{G}_\tau^{(1),*}; [\boldsymbol{\Theta}_{gnn}^{(1)}]_{\tau-1}))\right|$$

$$+ (1 + 2\gamma_2)\sqrt{\frac{2\log(1/\delta)}{t}}$$

with the probability at least $1 - 2\delta$. Then, for the RHS of the above inequality, by further applying **Lemma** G.8 and **Lemma** G.12, we have

$$\frac{1}{t} \sum_{\tau \in [t]} \left|f_{gnn}^{(2)}(\nabla f_\tau^{(1),*}(\boldsymbol{x}_\tau), \mathcal{G}_\tau^{(2),*}; [\boldsymbol{\Theta}_{gnn}^{(2)}]_{\tau-1}) - (r_\tau - f_{gnn}^{(1)}(\boldsymbol{x}_\tau, \mathcal{G}_\tau^{(1),*}; [\boldsymbol{\Theta}_{gnn}^{(1)}]_{\tau-1}))\right|$$

$$\le \frac{1}{t} \sum_{\tau \in [t]} \left|f_{gnn}^{(2)}(\nabla f_\tau^{(1),*}(\boldsymbol{x}_\tau), \mathcal{G}_\tau^{(2),*}; \widetilde{\boldsymbol{\Theta}}_{gnn}^{(2)}) - (r_\tau - f_{gnn}^{(1)}(\boldsymbol{x}_\tau, \mathcal{G}_\tau^{(1),*}; [\boldsymbol{\Theta}_{gnn}^{(1)}]_{\tau-1}))\right|$$

$$+ \frac{3L\sqrt{2t}}{2}$$

with regard to the parameter $\widetilde{\boldsymbol{\Theta}}_{gnn}^{(2)}$ s.t. $\|\widetilde{\boldsymbol{\Theta}}_{gnn}^{(2)} - [\boldsymbol{\Theta}_{gnn}^{(2)}]_0\|_2 \le \mathcal{O}\left(\frac{t^3}{\rho\sqrt{m}}\log m\right)$. Therefore, by applying the conclusion from **Lemma** G.8, we could bound the empirical loss w.r.t. $\widetilde{\boldsymbol{\Theta}}_{gnn}^{(2)}$ as

$$\frac{1}{t} \sum_{\tau \in [t]} \left|f_{gnn}^{(2)}(\nabla f_\tau^{(1),*}(\boldsymbol{x}_\tau), \mathcal{G}_\tau^{(2),*}; \widetilde{\boldsymbol{\Theta}}_{gnn}^{(2)}) - (r_\tau - f_{gnn}^{(1)}(\boldsymbol{x}_\tau, \mathcal{G}_\tau^{(1),*}; [\boldsymbol{\Theta}_{gnn}^{(1)}]_{\tau-1}))\right|$$

$$\le \frac{1}{\sqrt{t}} \sqrt{\sum_{\tau \in [t]} \left|f_{gnn}^{(2)}(\nabla f_\tau^{(1),*}(\boldsymbol{x}_\tau), \mathcal{G}_\tau^{(2),*}; \widetilde{\boldsymbol{\Theta}}_{gnn}^{(2)}) - (r_\tau - f_{gnn}^{(1)}(\boldsymbol{x}_\tau, \mathcal{G}_\tau^{(1),*}; [\boldsymbol{\Theta}_{gnn}^{(1)}]_{\tau-1}))\right|^2}$$

$$\le \sqrt{\frac{2\xi_2}{t}}.$$

Finally, assembling all the components and applying the union bound would complete the proof.

$\square$

Analogous to the **Lemma** G.1, we could also have the following corollary of the generalization results for the optimal arms and their rewards $\{\boldsymbol{x}_\tau^*, r_\tau^*\}_{\tau \in [t]}$ up to round $t$. Then, let $[\widehat{\boldsymbol{\Theta}}_{gnn}^{(1),*}]_t$ be the parameter that is trained on $\{\boldsymbol{x}_\tau^*, r_\tau^*\}_{\tau \in [t]}$, and denote $[\widehat{\boldsymbol{\Theta}}_{gnn}^{(2),*}]_t$ as the parameter of $f_{gnn}^{(2)}(\cdot)$ trained on corresponding gradients and residuals.

**Corollary G.3.** *For the constants $\rho \in (0, \mathcal{O}(\frac{1}{L}))$ and $\xi_2 \in (0, 1)$, given user $u \in \mathcal{U}$ and its past records $\mathcal{P}_{u,t}$, we suppose $m, \eta_1, \eta_2, J_1, J_2$ satisfy the conditions in **Theorem** 4.2, and randomly draw the parameter $[\boldsymbol{\Theta}_{gnn}^{(1),*}]_t \sim \{[\widehat{\boldsymbol{\Theta}}_{gnn}^{(1),*}]_\tau\}_{\tau \in [t]}$, $[\boldsymbol{\Theta}_{gnn}^{(2),*}]_t \sim \{[\widehat{\boldsymbol{\Theta}}_{gnn}^{(2),*}]_\tau\}_{\tau \in [t]}$. Then, with probability at least $1 - \delta$ given a sampled arm-reward pair $(\boldsymbol{x}, r)$, we have*

$$\sum_{\tau \in [t]} \mathbb{E}\left[\left|f_{gnn}^{(2)}\left(\nabla f_t^{(1),*}(\boldsymbol{x}_t^*), \mathcal{G}_{t,*}^{(2)}; [\boldsymbol{\Theta}_{gnn}^{(2)}]_{t-1}\right) - \left(r_t^* - f_{gnn}^{(1)}(\boldsymbol{x}_t, \mathcal{G}_{t,*}^{(1),*}; [\boldsymbol{\Theta}_{gnn}^{(1)}]_{t-1})\right)\right| \Big| u_t, \mathcal{X}_t\right]$$

$$\le \sqrt{t} \cdot \left(\sqrt{2\xi_2} + \frac{3L}{\sqrt{2}} + (1 + \gamma_2)\sqrt{2\log(\frac{tn \cdot a}{\delta})}\right) + \Gamma_t$$

*where*

$$\gamma_2 = 2 + \mathcal{O}\left(\frac{t^3 L}{\rho\sqrt{m}}\log m\right) + \mathcal{O}\left(\frac{L^2 t^4}{\rho^{4/3} m^{1/6}}\log^{11/6}(m)\right).$$

**Proof.** The proof of this corollary is comparable to the proof of **Lemma** G.2. At each time step $t$, regarding the definition of the optimal arm, we have $\boldsymbol{x}_t^* = \max_{\boldsymbol{x}_{i,t} \in \mathcal{X}_t} \mathbb{E}[r_{i,t}|u_t, \boldsymbol{x}_{i,t}]$. Then, analogously, we could define the difference sequence as

$$
\begin{aligned}
V_\tau^* =& \mathbb{E}_{\mathcal{X}_\tau}\left[\left| f_{gnn}^{(2)}(\nabla f_\tau^{(1),*}(\boldsymbol{x}_\tau^*),\ \mathcal{G}_\tau^{(2),*};[\boldsymbol{\Theta}_{gnn}^{(2),*}]_{\tau-1}) - (r_\tau - f_{gnn}^{(1)}(\boldsymbol{x}_\tau^*,\ \mathcal{G}_\tau^{(1),*};[\boldsymbol{\Theta}_{gnn}^{(1),*}]_{\tau-1})) \right|\right] \\
& - \left| f_{gnn}^{(2)}(\nabla f_\tau^{(1),*}(\boldsymbol{x}_\tau^*),\ \mathcal{G}_\tau^{(2),*};[\boldsymbol{\Theta}_{gnn}^{(2),*}]_{\tau-1}) - (r_\tau - f_{gnn}^{(1)}(\boldsymbol{x}_\tau^*,\ \mathcal{G}_\tau^{(1),*};[\boldsymbol{\Theta}_{gnn}^{(1),*}]_{\tau-1})) \right|
\end{aligned}
$$

where by reusing the notation, we denote $\mathcal{G}_\tau^{(1),*}, \mathcal{G}_\tau^{(2),*}$ to be the true user graphs w.r.t. the optimal arm $\boldsymbol{x}_\tau^*$ here. Then, similar to the proof of **Lemma** G.2, we have the sequence to be the martingale difference sequence as

$$
\begin{aligned}
\mathbb{E}[V_\tau^*|F_\tau^*] =& \mathbb{E}_{\mathcal{X}_\tau}\left[\left| f_{gnn}^{(2)}(\nabla f_\tau^{(1),*}(\boldsymbol{x}_\tau^*),\ \mathcal{G}_\tau^{(2),*};[\boldsymbol{\Theta}_{gnn}^{(2),*}]_{\tau-1}) - (r_\tau - f_{gnn}^{(1)}(\boldsymbol{x}_\tau^*,\ \mathcal{G}_\tau^{(1),*};[\boldsymbol{\Theta}_{gnn}^{(1),*}]_{\tau-1})) \right|\right] \\
& - \mathbb{E}_{\mathcal{X}_\tau}\left[\left| f_{gnn}^{(2),*}(\nabla f_\tau^{(1),*}(\boldsymbol{x}_\tau^*),\ \mathcal{G}_\tau^{(2),*};[\boldsymbol{\Theta}_{gnn}^{(2),*}]_{\tau-1}) - (r_\tau - f_{gnn}^{(1)}(\boldsymbol{x}_\tau^*,\ \mathcal{G}_\tau^{(1),*};[\boldsymbol{\Theta}_{gnn}^{(1),*}]_{\tau-1})) \right|\right] = 0
\end{aligned}
$$

with $F_\tau^*$ being the filtration of past optimal arms up to round $\tau$. Then, we could also applying the Azuma-Hoeffding inequality to bound the difference between the mean $\frac{1}{t}\sum_{\tau \in [t]} V_\tau^*$ and its expectation $\frac{1}{t}\sum_{\tau \in [t]} \mathbb{E}[V_\tau^*]$. Finally, like in the proof of **Lemma** G.2, applying the conclusion from **Lemma** G.8 and **Lemma** G.12 would complete the proof.

$\square$

Then, recall the definition of of the confidence bound function $\mathsf{CB}_t(\boldsymbol{x}_t^*)$ w.r.t. the optimal arm $\boldsymbol{x}_t^*$, we the corresponding term $I_1$ as

$$
I_1 = \mathbb{E}\left[ |f_{gnn}^{(2)}(\nabla f_t^{(1),*}(\boldsymbol{x}_t^*),\ \mathcal{G}_t^{(2),*};[\boldsymbol{\Theta}_{gnn}^{(2)}]_{t-1}) - (r_t - f_{gnn}^{(1)}(\boldsymbol{x}_t^*,\ \mathcal{G}_t^{(1),*};[\boldsymbol{\Theta}_{gnn}^{(1)}]_{t-1}))| \Big| u_t, \mathcal{X}_t \right].
$$

And it can be further decomposed as

$$
\begin{aligned}
& |f_{gnn}^{(2)}(\nabla f_t^{(1),*}(\boldsymbol{x}_t^*),\ \mathcal{G}_t^{(2),*};[\boldsymbol{\Theta}_{gnn}^{(2)}]_{t-1}) - (r_t - f_{gnn}^{(1)}(\boldsymbol{x}_t^*,\ \mathcal{G}_t^{(1),*};[\boldsymbol{\Theta}_{gnn}^{(1)}]_{t-1}))| \\
& \leq |f_{gnn}^{(2)}(\nabla f_t^{(1),*}(\boldsymbol{x}_t^*),\ \mathcal{G}_t^{(2),*};[\boldsymbol{\Theta}_{gnn}^{(2),*}]_{t-1}) - (r_t - f_{gnn}^{(1)}(\boldsymbol{x}_t^*,\ \mathcal{G}_t^{(1),*};[\boldsymbol{\Theta}_{gnn}^{(1),*}]_{t-1}))| + \\
& \quad + |f_{gnn}^{(1)}(\boldsymbol{x}_t^*,\ \mathcal{G}_t^{(1),*};[\boldsymbol{\Theta}_{gnn}^{(1),*}]_{t-1}) - f_{gnn}^{(1)}(\boldsymbol{x}_t^*,\ \mathcal{G}_t^{(1),*};[\boldsymbol{\Theta}_{gnn}^{(1)}]_{t-1})| \\
& \quad + |f_{gnn}^{(2)}(\nabla f_t^{(1),*}(\boldsymbol{x}_t^*),\ \mathcal{G}_t^{(2),*};[\boldsymbol{\Theta}_{gnn}^{(2),*}]_{t-1}) - f_{gnn}^{(2)}(\nabla f_t^{(1),*}(\boldsymbol{x}_t^*),\ \mathcal{G}_t^{(2),*};[\boldsymbol{\Theta}_{gnn}^{(2)}]_{t-1})|
\end{aligned}
$$

where the first term on the RHS could be bounded by **Corollary** G.3. Then, for the second term, we first denote $\boldsymbol{h}_i^* \in \mathbb{R}^m$ to be the aggregated hidden representation w.r.t. the user-arm pair $(u_i, \boldsymbol{x}_t^*)$ where $u_i$ is the $i$-th user. Here, $\boldsymbol{h}_i^*$ is essentially the row in the aggregated representation matrix $\boldsymbol{H}_{agg}$ corresponding to the user arm pair $(u_t, \boldsymbol{x}_t^*)$. Therefore, for the received user $u_t \in \mathcal{U}$, the reward estimation based on two samples regarding the two sets of parameters would have the same the input $\boldsymbol{h}_t^*$. Then, for the second term, since the outputs w.r.t. two sets of parameters have the same input $\boldsymbol{h}_t^*$, we could apply the conclusion from **Lemma** G.14, which will lead to

$$
\begin{aligned}
& |f_{gnn}^{(1)}(\boldsymbol{x}_t^*,\ \mathcal{G}_t^{(1),*};[\boldsymbol{\Theta}_{gnn}^{(1),*}]_{t-1}) - f_{gnn}^{(1)}(\boldsymbol{x}_t^*,\ \mathcal{G}_t^{(1),*};[\boldsymbol{\Theta}_{gnn}^{(1)}]_{t-1})| \\
& \leq \left(1 + \mathcal{O}(\frac{t L^3 \log^{5/6}(m)}{\rho^{1/3} m^{1/6}})\right) \cdot \mathcal{O}(\frac{t^3 L}{\rho\sqrt{m}}\log(m)) + \mathcal{O}\left(\frac{t^4 L^2 \log^{11/6}(m)}{\rho^{4/3} m^{1/6}}\right).
\end{aligned}
$$

Analogously, we could also have the same bound for the third term on the RHS. Summing up the bounds for three terms on the RHS would finish deriving the upper bound for term $I_1$.

### G.2  BOUNDING THE EXPLOITATION GRAPH ESTIMATION ERROR

Then, we proceed to bound the error induced by the estimation of user exploitation graph, i.e., the error term $I_2$. Recall that the confidence bound function $\mathsf{CB}_t(x)$ for the given arm $x \in \mathcal{X}_t$ is

$$\mathsf{CB}_t(x) = \mathbb{E}\left[|f_{gnn}^{(2)}(\nabla f_t^{(1)}(x),\, \mathcal{G}^{(2)};\, [\Theta_{gnn}^{(2)}]_{t-1}) - (r - f_{gnn}^{(1)}(x,\, \mathcal{G}^{(1)};\, [\Theta_{gnn}^{(1)}]_{t-1}))|\,\Big|\,u_t, \mathcal{X}_t\right]$$

$$\leq \underbrace{\mathbb{E}\left[|f_{gnn}^{(1)}(x,\, \mathcal{G}^{(1),*};\, [\Theta_{gnn}^{(1)}]_{t-1}) - f_{gnn}^{(1)}(x,\, \mathcal{G}^{(1)};\, [\Theta_{gnn}^{(1)}]_{t-1})|\,\Big|\,u_t, \mathcal{X}_t\right]}_{I_2} + I_1 + I_3 + I_4.$$

given an arbitrary arm $x \in \mathcal{X}_t$. For arm $x$, we use the following lemma to bound the error caused by the difference between the estimated exploitation graph $\mathcal{G}^{(1)}$ and the true exploitation graph $\mathcal{G}^{(1),*}$ associated with arm $x$.

Denoting the adjacency matrix of the estimated graph $\mathcal{G}^{(1)}$ as $A^{(1)}$, and the adjacency matrix for the true user exploitation graph $\mathcal{G}^{(1),*}$ as $A^{(1),*}$, we have the normalized adjacency matrices as $S^{(1)} = A^{(1)}/n$ and $S^{(1),*} = A^{(1),*}/n$. For the $i$-th user $u_i \in \mathcal{U}$, we could deem the $i$-th row of the matrix multiplication $S \cdot X$, represented by $h_{0,i} = [S \cdot X]_{i:}$, as the aggregated input for the network for the user-arm pair $(x, u_i)$. Note that in this way, the rest of the network could be regarded as a $L + 1$-layer FC network, where the weight matrix for the first layer is $\Theta_{agg}^{(1)}$. Then, to make sure each aggregated input has the norm of 1, we apply an additional transformation mentioned in **Eq.** 9 as $\tilde{h}_{0,i} = \phi(h_{0,i}, x) = (\frac{h_{0,i}}{\sqrt{2}}, \frac{x}{2}, c_{0,i})$ where $c_{0,i} = \sqrt{\frac{3}{4} - \frac{1}{2}\|h_{0,i}\|_2^2}$. And this transformation ensures $\|\tilde{h}_{0,i}\|_2 = 1$ and $c_{0,i} \geq \frac{1}{2}$. Since this transformation does not alter the original aggregated representation $h_{0,i}$, it will not impair the original information w.r.t. the user-arm pair $(x, u_i)$. Meantime, note that this transformation also ensures the separateness of the transformed contexts to be at least $\frac{\rho}{2}$.

**Lemma G.4.** *For the constants $\rho \in (0, \mathcal{O}(\frac{1}{L}))$ and $\xi_1 \in (0, 1)$, given past records $\mathcal{P}_{t-1}$, we suppose $m, \eta_1, \eta_2, J_1, J_2$ satisfy the conditions in **Theorem** 4.2, and randomly draw the parameter $[\Theta_{gnn}^{(1)}]_t \sim \{[\widehat{\Theta}_{gnn}^{(1)}]_\tau\}_{\tau \in [t]}$. Then, with probability at least $1 - \delta$, given an arm $x \in \mathbb{R}^d$, we have*

$$\sum_{\tau \in [t]} |f_{gnn}^{(1)}(x,\, \mathcal{G}_\tau^{(1),*};\, [\Theta_{gnn}^{(1)}]_{\tau-1}) - f_{gnn}^{(1)}(x,\, \mathcal{G}_\tau^{(1)};\, [\Theta_{gnn}^{(1)}]_{\tau-1})|$$

$$\leq \mathcal{O}(L) \cdot \sqrt{8t} \cdot \left(\sqrt{2\xi_1} + \frac{3L}{\sqrt{2}} + (1 + \gamma_1)\sqrt{2\log(\frac{tn \cdot a}{\delta})}\right) + \Gamma_t.$$

**Proof.** By the conclusion of **Lemma** F.2, at time step $t$, the reward estimation error of the user exploitation model could be bounded as

$$\sum_{\tau \in [t]} \mathbb{E}\left[|f_u^{(1)}(x; [\Theta_u^{(1)}]_t) - r|\,|\mathcal{X}_t\right] \leq \sqrt{\frac{t}{n}} \cdot \left(\sqrt{2\xi_1} + \frac{3L}{\sqrt{2}} + (1 + \gamma_1)\sqrt{2\log(\frac{tn \cdot a}{\delta})}\right) + \Gamma_t.$$

with the probability at least $1 - \delta$. And given two users $u_i, u_j \in \mathcal{U}$ and an arbitrary arm $x \in \mathcal{X}_t$, we denote their individual reward as $r_i, r_j$ separately. We omit the expectation notation below for simplicity. Then, we could bound the absolute difference between the reward estimations as

$$||r_i - r_j| - |f_u^{(1)}(x_i; [\Theta_{u_i}^{(1)}]_t) - f_u^{(1)}(x_j; [\Theta_{u_j}^{(1)}]_t)|| \leq |(r_i - f_u^{(1)}(x_i; [\Theta_{u_i}^{(1)}]_t)) - (r_j - f_u^{(1)}(x_j; [\Theta_{u_j}^{(1)}]_t))|$$

$$\leq |(r_i - f_u^{(1)}(x_i; [\Theta_{u_i}^{(1)}]_t)) - (r_j - f_u^{(1)}(x_j; [\Theta_{u_j}^{(1)}]_t))|$$

$$\leq |(r_i - f_u^{(1)}(x_i; [\Theta_{u_i}^{(1)}]_t))| + |(r_j - f_u^{(1)}(x_j; [\Theta_{u_j}^{(1)}]_t))|.$$

Based on the definition of the mapping function $\Psi_1$, it would naturally be Lipschitz continuous with the coefficient of 1, which is

$$|\exp(-|r_i - r_j|) - \exp(-|f_u^{(1)}(x_i; [\Theta_{u_i}^{(1)}]_t) - f_u^{(1)}(x_j; [\Theta_{u_j}^{(1)}]_t)|)|$$

$$\leq ||r_i - r_j| - |f_u^{(1)}(x_i; [\Theta_{u_i}^{(1)}]_t) - f_u^{(1)}(x_j; [\Theta_{u_j}^{(1)}]_t)||.$$

with the probability at least $1 - \delta$. Finally, applying the union bound for all the $(n^2 - n)/2$ user pairs and re-scaling the $\delta$ would give us the estimation error bound for the reward difference for each pair of users. To achieve the upper bound, we apply the Corollary F.3 by considering the trajectory $\tilde{\mathcal{P}}_{u,t}$ consists of the past arm-reward pairs $\{\boldsymbol{x}_{i_\tau,\tau}, r_{i_\tau,\tau}\}_{\tau \in [t]}$, where arm $\boldsymbol{x}_{i_\tau,\tau}$ leads to the largest estimation error of the estimation model $f_{u_\tau}^{(1)}(\cdot)$ in each round $\tau \in [t]$. Thus, we have the bound for the edge weight difference, where the difference of an arbitrary $i$-th row could be bounded by

$$\sum_{\tau \in [t]} \|[\boldsymbol{A}_\tau^{(1)}]_{i:} - [\boldsymbol{A}_\tau^{(1),*}]_{i:}\|_2 \leq 2n\sqrt{t} \cdot \left( \sqrt{2\xi_1} + \frac{3L}{\sqrt{2}} + (1+\gamma_1)\sqrt{2\log(\frac{tn \cdot a}{\delta})} \right) + \Gamma_t,$$

which implies

$$\sum_{\tau \in [t]} \|[\boldsymbol{S}_\tau^{(1)}]_{i:} - [\boldsymbol{S}_\tau^{(1),*}]_{i:}\|_2 \leq 2\sqrt{t} \cdot \left( \sqrt{2\xi_1} + \frac{3L}{\sqrt{2}} + (1+\gamma_1)\sqrt{2\log(\frac{tn \cdot a}{\delta})} \right) + \Gamma_t.$$

Therefore, applying the conclusions from **Lemma** F.2, it leads to

$$\sum_{\tau \in [t]} ||r_{i,\tau} - r_{j,\tau}| - |f_u^{(1)}(\boldsymbol{x}_{i,\tau}; [\boldsymbol{\Theta}_{u_i}^{(1)}]_t) - f_u^{(1)}(\boldsymbol{x}_{j,\tau}; [\boldsymbol{\Theta}_{u_j}^{(1)}]_t)||$$

$$\leq 2\sqrt{\frac{t}{n}} \cdot \left( \sqrt{2\xi_1} + \frac{3L}{\sqrt{2}} + (1+\gamma_1)\sqrt{2\log(\frac{tn \cdot a}{\delta})} \right) + \Gamma_t$$

Afterwards, recalling the transformation at the beginning of this subsection, and given an user-arm pair $(u_i, \boldsymbol{x})$ for the $i$-th user, we denote $\boldsymbol{h} = [\boldsymbol{S}^{(1)} \cdot \boldsymbol{X}]_{i:}$ and $\boldsymbol{h}^* = [\boldsymbol{S}^{(1),*} \cdot \boldsymbol{X}]_{i:}$. Based the aforementioned transformation in **Eq.** 9, their transformed form could naturally be $\tilde{\boldsymbol{h}} = (\frac{\sqrt{2}}{2}\boldsymbol{h}, \frac{\boldsymbol{x}}{2}, c)$ and $\tilde{\boldsymbol{h}}^* = (\frac{\sqrt{2}}{2}\boldsymbol{h}^*, \frac{\boldsymbol{x}}{2}, c^*)$ with $\|\boldsymbol{x}\|_2 = 1$. Without the loss of generality, we let $c > c^*$. Then, we could have

$$\|\tilde{\boldsymbol{h}} - \tilde{\boldsymbol{h}}^*\|_2 = \sqrt{\|\boldsymbol{h} - \boldsymbol{h}^*\|_2^2 + (c - c^*)^2} \underset{(i)}{\leq} \sqrt{\|\boldsymbol{h} - \boldsymbol{h}^*\|_2^2 + (c^2 - (c^*)^2)^2}$$

$$\underset{(ii)}{=} \sqrt{\|\boldsymbol{h} - \boldsymbol{h}^*\|_2^2 + \frac{1}{4}(\|\boldsymbol{h}^*\|_2^2 - \|\boldsymbol{h}\|_2^2)^2}$$

$$= \sqrt{\|\boldsymbol{h} - \boldsymbol{h}^*\|_2^2 + \frac{1}{4}(\|\boldsymbol{h}^* - \boldsymbol{h}\|_2 \cdot \|\boldsymbol{h}^* + \boldsymbol{h}\|_2)^2}$$

$$\underset{(iii)}{\leq} \sqrt{2} \cdot \|\boldsymbol{h} - \boldsymbol{h}^*\|_2$$

Here, (i) is because $c, c^* \geq \frac{1}{2}$. (ii) is because of $c^2 + \frac{\|\boldsymbol{h}\|_2^2}{2} = (c^*)^2 + \frac{\|\boldsymbol{h}^*\|_2^2}{2} = \frac{3}{4}$, and (iii) is due to $\|\boldsymbol{h}\|_2, \|\boldsymbol{h}^*\|_2 \leq 1$.

Then, we proceed to bound $\|\boldsymbol{h} - \boldsymbol{h}^*\|_2$. Recall the definition from **Eq.** 4. Extending the above conclusion across different rounds $\tau \in [t]$, we will have

$$\sum_{\tau \in [t]} \|\boldsymbol{h}_\tau - \boldsymbol{h}_\tau^*\|_2 \leq \|\boldsymbol{x}_\tau\| \cdot \|[\boldsymbol{S}_\tau^{(1)}]_{i:} - [\boldsymbol{S}_\tau^{(1),*}]_{i:}\|_2 \leq 2\sqrt{t}\left( \sqrt{2\xi_1} + \frac{3L}{\sqrt{2}} + (1+\gamma_1)\sqrt{2\log(\frac{tn \cdot a}{\delta})} \right) + \Gamma_t.$$

Finally, combining the conclusion from **Lemma** G.13, we finally have

$$\sum_{\tau \in [t]} |f_{gnn}^{(1)}(\boldsymbol{x}, \mathcal{G}_\tau^{(1),*}; [\boldsymbol{\Theta}_{gnn}^{(1)}]_{\tau-1}) - f_{gnn}^{(1)}(\boldsymbol{x}, \mathcal{G}_\tau^{(1)}; [\boldsymbol{\Theta}_{gnn}^{(1)}]_{\tau-1})|$$

$$\leq \mathcal{O}(L) \cdot \sqrt{8t} \cdot \left( \sqrt{2\xi_1} + \frac{3L}{\sqrt{2}} + (1+\gamma_1)\sqrt{2\log(\frac{tn \cdot a}{\delta})} \right) + \Gamma_t$$

which concludes the proof.

$$\square$$

### G.3 BOUNDING THE EXPLORATION GRAPH ESTIMATION ERROR

Again, recall the definition of the confidence bound function $\mathsf{CB}_t(\boldsymbol{x})$ which is

$$\mathsf{CB}_t(\boldsymbol{x}) = \mathbb{E}\left[|f_{gnn}^{(2)}(\nabla f_t^{(1)}(\boldsymbol{x}), \mathcal{G}^{(2)}; [\boldsymbol{\Theta}_{gnn}^{(2)}]_{t-1}) - (r - f_{gnn}^{(1)}(\boldsymbol{x}, \mathcal{G}^{(1)}; [\boldsymbol{\Theta}_{gnn}^{(1)}]_{t-1}))|\,\Big|u_t, \mathcal{X}_t\right]$$

$$\leq \underbrace{\mathbb{E}\left[|f_{gnn}^{(2)}(\nabla f_t^{(1),*}(\boldsymbol{x}), \mathcal{G}^{(2),*}; [\boldsymbol{\Theta}_{gnn}^{(2)}]_{t-1}) - f_{gnn}^{(2)}(\nabla f_t^{(1),*}(\boldsymbol{x}), \mathcal{G}^{(2)}; [\boldsymbol{\Theta}_{gnn}^{(2)}]_{t-1})|\,\Big|u_t, \mathcal{X}_t\right]}_{I_3}$$

$$+ I_1 + I_2 + I_4.$$

Analogous to the procedure for the user exploitation graph, we have the following lemma to bound the error induced by user exploitation graph estimation.

**Lemma G.5.** *For the constants $\rho \in (0, \mathcal{O}(\frac{1}{L}))$ and $\xi_1 \in (0,1)$, given past records $\mathcal{P}_{t-1}$, we suppose $m, \eta_1, \eta_2, J_1, J_2$ satisfy the conditions in **Theorem 4.2**, and randomly draw the parameter $[\boldsymbol{\Theta}_{gnn}^{(2)}]_t \sim \{[\widehat{\boldsymbol{\Theta}}_{gnn}^{(2)}]_\tau\}_{\tau \in [t]}$. Then, with probability at least $1 - \delta$, given an arm $\boldsymbol{x} \in \mathbb{R}^d$, we have*

$$\sum_{\tau \in [t]} |f_{gnn}^{(2)}(\nabla f_t^{(1),*}(\boldsymbol{x}), \mathcal{G}^{(2),*}; [\boldsymbol{\Theta}_{gnn}^{(2)}]_{t-1}) - f_{gnn}^{(2)}(\nabla f_t^{(1),*}(\boldsymbol{x}), \mathcal{G}^{(2)}; [\boldsymbol{\Theta}_{gnn}^{(2)}]_{t-1})|$$

$$\leq \mathcal{O}(L) \cdot \sqrt{8t} \cdot \left(\sqrt{2\xi_1} + \frac{3L}{\sqrt{2}} + (1 + \gamma_1)\sqrt{2\log(\frac{tn \cdot a}{\delta})}\right) + \Gamma_t.$$

**Proof.** The proof of this lemma could be derived based on a similar approach as in **Lemma** G.4. Recall that for the exploration GNN model $f_{gnn}^{(2)}(\cdot)$, we have the gradients of the GNN exploitation model $\nabla f_{u,t}^{(1)}(\boldsymbol{x}) = \frac{\nabla_{[\boldsymbol{\Theta}_u^{(1)}]_t} f_u^{(1)}(\boldsymbol{x};[\boldsymbol{\Theta}_u^{(1)}]_t)}{c_g' L}$ as the input given an arm $\boldsymbol{x}$ and user $u \in \mathcal{U}$, whose norm $\|\nabla f_{u,t}^{(1)}(\boldsymbol{x})\|_2 \leq 1$.

Given two users $u_i, u_j \in \mathcal{U}$ and an arbitrary arm $\boldsymbol{x} \in \mathcal{X}_t$, we denote their individual reward as $r_i, r_j$ separately. Then, we could bound the absolute difference between the potential gain estimations as

$$||(r_i - f_u^{(1)}(\boldsymbol{x};[\boldsymbol{\Theta}_{u_i}^{(1)}]_t)) - (r_j - f_u^{(1)}(\boldsymbol{x};[\boldsymbol{\Theta}_{u_j}^{(1)}]_t))| - |f_u^{(2)}(\nabla f_{u_i,t}^{(1)}(\boldsymbol{x});[\boldsymbol{\Theta}_{u_i}^{(2)}]_t) - f_u^{(2)}(\nabla f_{u_j,t}^{(1)}(\boldsymbol{x});[\boldsymbol{\Theta}_{u_j}^{(2)}]_t)||$$

$$\leq |f_u^{(2)}(\nabla f_{u_i,t}^{(1)}(\boldsymbol{x});[\boldsymbol{\Theta}_{u_i}^{(2)}]_t)) - (r_i - f_u^{(1)}(\boldsymbol{x};[\boldsymbol{\Theta}_{u_i}^{(1)}]_t))|$$

$$+ |f_u^{(2)}(\nabla f_{u_j,t}^{(1)}(\boldsymbol{x});[\boldsymbol{\Theta}_{u_j}^{(2)}]_t) - (r_j - f_u^{(1)}(\boldsymbol{x};[\boldsymbol{\Theta}_{u_j}^{(1)}]_t))|.$$

Afterwards, applying the conclusion from **Lemma** F.4 would lead to the result that

$$\sum_{\tau \in [t]} ||(r_i - f_u^{(1)}(\boldsymbol{x};[\boldsymbol{\Theta}_{u_i}^{(1)}]_t)) - (r_j - f_u^{(1)}(\boldsymbol{x};[\boldsymbol{\Theta}_{u_j}^{(1)}]_t))| - |f_u^{(2)}(\nabla f_{u_i,t}^{(1)}(\boldsymbol{x});[\boldsymbol{\Theta}_{u_i}^{(2)}]_t) - f_u^{(2)}(\nabla f_{u_j,t}^{(1)}(\boldsymbol{x});[\boldsymbol{\Theta}_{u_j}^{(2)}]_t)||$$

$$\leq 2\sqrt{\frac{t}{n}} \cdot \left(\sqrt{2\xi_1} + \frac{3L}{\sqrt{2}} + (1 + \gamma_1)\sqrt{2\log(\frac{tn \cdot a}{\delta})}\right) + \Gamma_t.$$

Following a similar approach as in the proof of **Lemma** G.4, we proceed to consider the aggregated hidden representations for the input gradients. Since the entries between $\boldsymbol{A}^{(2)} - \boldsymbol{A}^{(2),*}$ (and also the distance between $\boldsymbol{S}^{(2)} - \boldsymbol{S}^{(2),*}$) are bounded, by adopting the aforementioned transformation in **Eq.** 9 on the aggregated hidden representations for the input gradients and the initial arm contexts $\boldsymbol{x}$, we would end up with the bound for the difference between transformed representations for input gradients. Finally, combining the conclusion from **Lemma** G.13 would give the proof.

$$\square$$

### G.4 BOUNDING THE GRADIENT INPUT ESTIMATION ERROR

For the last term $I_4$ in the confidence bound function $\mathsf{CB}_t(\boldsymbol{x})$, we have

$$I_4 = \mathbb{E}\left[|f_{gnn}^{(2)}(\nabla f_t^{(1),*}(\boldsymbol{x}), \mathcal{G}^{(2)}; [\boldsymbol{\Theta}_{gnn}^{(2)}]_{t-1}) - f_{gnn}^{(2)}(\nabla f_t^{(1)}(\boldsymbol{x}), \mathcal{G}^{(2)}; [\boldsymbol{\Theta}_{gnn}^{(2)}]_{t-1})|\,\Big|u_t, \mathcal{X}_t\right]$$

which represents the estimation error induced by the difference of input gradients. And we first bound the gradient difference with the following lemma

**Lemma G.6.** *For the constants* $\rho \in (0, \mathcal{O}(\frac{1}{L}))$ *and* $\xi_1 \in (0,1)$, *given past records* $\mathcal{P}_{t-1}$, *we suppose* $m, \eta_1, \eta_2, J_1, J_2$ *satisfy the conditions in **Theorem 4.2**, and randomly draw the parameter* $[\mathbf{\Theta}_{gnn}^{(1)}]_t \sim \{[\widehat{\mathbf{\Theta}}_{gnn}^{(1)}]_\tau\}_{\tau \in [t]}$. *Then, with probability at least* $1 - \delta$, *given an arm* $\boldsymbol{x} \in \mathbb{R}^d$, *we have*

$$\sum_{\tau \in [t]} \|\nabla f_\tau^{(1)}(\boldsymbol{x}) - \nabla f_\tau^{(1),*}(\boldsymbol{x})\|_2$$

$$\leq \mathcal{O}(\frac{t^2 L^4 \log^{5/6}(m)}{\rho^{1/3} m^{1/6}}) + \mathcal{O}(L) \cdot \sqrt{8t} \cdot \left( \sqrt{2\xi_1} + \frac{3L}{\sqrt{2}} + (1 + \gamma_1)\sqrt{2 \log(\frac{tn \cdot a}{\delta})} \right) + \Gamma_t$$

*where* $\nabla f_t^{(1)}(\boldsymbol{x}) = \frac{\nabla_{\mathbf{\Theta}_{gnn}^{(1)}} f_{gnn}^{(1)}(\boldsymbol{x}, \mathcal{G}^{(1)};[\mathbf{\Theta}_{gnn}^{(1)}]_{t-1})}{c_g L}$, *and* $\nabla f_t^{(1),*}(\boldsymbol{x}) = \frac{\nabla_{\mathbf{\Theta}_{gnn}^{(1)}} f_{gnn}^{(1)}(\boldsymbol{x}, \mathcal{G}^{(1),*};[\mathbf{\Theta}_{gnn}^{(1)}]_{t-1})}{c_g L}$.

**Proof.** Following the aggregation procedure and transformation procedure shown in section G.2, we have the transformed representations for given an user-arm pair $(u_i, \boldsymbol{x})$ with the $i$-th user, which are $\boldsymbol{h} = [\boldsymbol{S}^{(1)} \cdot \boldsymbol{X}]_{i:}$ and $\boldsymbol{h}^* = [\boldsymbol{S}^{(1),*} \cdot \boldsymbol{X}]_{i:}$. And their transformed form could naturally be $\tilde{\boldsymbol{h}} = (\boldsymbol{h}, c)$ and $\tilde{\boldsymbol{h}}^* = (\boldsymbol{h}^*, c^*)$. From the conclusion of **Lemma G.4**, we have

$$\sum_{\tau \in [t]} \|\tilde{\boldsymbol{h}}_\tau - \tilde{\boldsymbol{h}}_\tau^*\|_2 \leq \sqrt{8t} \cdot \left( \sqrt{2\xi_1} + \frac{3L}{\sqrt{2}} + (1 + \gamma_1)\sqrt{2 \log(\frac{tn \cdot a}{\delta})} \right) + \Gamma_t.$$

Then, applying the conclusion from **Lemma G.13** would complete the proof.

$\square$

Then, we have te following lemma to bound the term $I_4$.

**Lemma G.7.** *For the constants* $\rho \in (0, \mathcal{O}(\frac{1}{L}))$ *and* $\xi_1 \in (0,1)$, *given past records* $\mathcal{P}_{t-1}$, *we suppose* $m, \eta_1, \eta_2, J_1, J_2$ *satisfy the conditions in **Theorem 4.2**, and randomly draw the parameter* $[\mathbf{\Theta}_{gnn}^{(1)}]_t \sim \{[\widehat{\mathbf{\Theta}}_{gnn}^{(1)}]_\tau\}_{\tau \in [t]}$. *Then, with probability at least* $1 - \delta$, *given an arm* $\boldsymbol{x} \in \mathbb{R}^d$, *we have*

$$\sum_{\tau \in [t]} |f_{gnn}^{(2)}(\nabla f_\tau^{(1),*}(\boldsymbol{x}), \mathcal{G}^{(2)};[\mathbf{\Theta}_{gnn}^{(2)}]_{\tau-1}) - f_{gnn}^{(2)}(\nabla f_\tau^{(1)}(\boldsymbol{x}), \mathcal{G}^{(2)};[\mathbf{\Theta}_{gnn}^{(2)}]_{\tau-1})|$$

$$\leq \mathcal{O}(\frac{t^2 L^5 \log^{5/6}(m)}{\rho^{1/3} m^{1/6}}) + \mathcal{O}(L^2) \cdot \sqrt{8t} \cdot \left( \sqrt{2\xi_1} + \frac{3L}{\sqrt{2}} + (1 + \gamma_1)\sqrt{2 \log(\frac{tn \cdot a}{\delta})} \right) + \Gamma_t.$$

**Proof.** We again follow the aggregation procedure and transformation procedure presented in section G.2. Then, the aggregated and transformed input gradient could be denoted as we have the transformed representations for given an user-arm pair $(u_i, \boldsymbol{x})$ with the $i$-th user, which are $\boldsymbol{g} = [\boldsymbol{S}^{(2)} \cdot \boldsymbol{G}]_{i:}$ and $\boldsymbol{g}^* = [\boldsymbol{S}^{(2)} \cdot \boldsymbol{G}^*]_{i:}$, where $\boldsymbol{G}$ denotes the gradient matrix embedded w.r.t. **Eq. 4**. And their transformed form could be $\tilde{\boldsymbol{g}} = (\frac{\sqrt{2}}{2}\boldsymbol{g}, c)$ and $\tilde{\boldsymbol{g}}^* = (\frac{\sqrt{2}}{2}\boldsymbol{g}^*, c^*)$. Then, according to the definition of **Eq. 4**, we could naturally have

$$\|\tilde{\boldsymbol{g}} - \tilde{\boldsymbol{g}}^*\|_2 \leq \|[\boldsymbol{S}^{(2)}]_{i:}\|_2 \cdot \|\nabla f_t^{(1),*}(\boldsymbol{x}) - \nabla f_t^{(1)}(\boldsymbol{x})\|_2 \leq \|\nabla f_t^{(1),*}(\boldsymbol{x}) - \nabla f_t^{(1)}(\boldsymbol{x})\|_2$$

since the normalization of the adjacency matrix ensures its arbitrary row has the norm smaller than 1. Finally, applying the conclusions from **Lemma G.6** and **Lemma G.13**, it will leads to

$$\sum_{\tau \in [t]} |f_{gnn}^{(2)}(\nabla f_\tau^{(1),*}(\boldsymbol{x}), \mathcal{G}^{(2)};[\mathbf{\Theta}_{gnn}^{(2)}]_{\tau-1}) - f_{gnn}^{(2)}(\nabla f_\tau^{(1)}(\boldsymbol{x}), \mathcal{G}^{(2)};[\mathbf{\Theta}_{gnn}^{(2)}]_{\tau-1})|$$

$$\leq \mathcal{O}(\frac{t^2 L^5 \log^{5/6}(m)}{\rho^{1/3} m^{1/6}}) + \mathcal{O}(L^2) \cdot \sqrt{8t} \cdot \left( \sqrt{2\xi_1} + \frac{3L}{\sqrt{2}} + (1 + \gamma_1)\sqrt{2 \log(\frac{tn \cdot a}{\delta})} \right) + \Gamma_t.$$

$\square$

### G.5 Lemmas for Over-parameterized Networks

Applying $\mathcal{P}_{t-1}$ as the training data, we have the following convergence result for the exploitation GNN network $f_u^{(1)}(\cdot; \mathbf{\Theta}_{gnn}^{(1)})$ after GD.

**Lemma G.8** (Theorem 1 from (Allen-Zhu et al., 2019)). *For any $0 < \xi_2 \leq 1$, $0 < \rho \leq \mathcal{O}(\frac{1}{L})$. Given past records $\mathcal{P}_{t-1}$, suppose $m, \eta_1, \eta_2, J_1, J_2$ satisfy the conditions in **Theorem** 4.2, then with probability at least $1 - \delta$, we could have*

1. *$\mathcal{L}(\mathbf{\Theta}_{gnn}^{(1)}) \leq \xi_2$ after $J_2$ iterations of GD.*

2. *For any $j \in [J_2]$, $\|[\mathbf{\Theta}_{gnn}^{(1)}]^j - [\mathbf{\Theta}_{gnn}^{(1)}]^0\| \leq \mathcal{O}\left(\frac{t^3}{\rho\sqrt{m}} \log m\right)$.*

In particular, **Lemma** F.6 above provides the convergence guarantee for $f_u^{(1)}(\cdot; \mathbf{\Theta}_{gnn}^{(1)})$ after certain rounds of GD training on the past records $\mathcal{P}_{t-1}$.

**Lemma G.9** (Lemma 4.1 in (Cao & Gu, 2019)). *Assume a constant $\omega$ such that $\mathcal{O}(m^{-3/2}L^{-3/2}[\log(TnL^2/\delta)]^{3/2}) \leq \omega \leq \mathcal{O}(L^{-6}[\log m]^{-3/2})$ and $n$ training samples. With randomly initialized $[\mathbf{\Theta}_{gnn}^{(1)}]_0$, for parameters $\mathbf{\Theta}, \mathbf{\Theta}'$ satisfying $\|\mathbf{\Theta} - [\mathbf{\Theta}_{gnn}^{(1)}]_0\|, \|\mathbf{\Theta} - [\mathbf{\Theta}_{gnn}^{(1)}]_0\| \leq \omega$, we have*

$$|f^{(1)}(\boldsymbol{x}; \mathbf{\Theta}) - f^{(1)}(\boldsymbol{x}; \mathbf{\Theta}') - \langle \nabla_{\mathbf{\Theta}'} f^{(1)}(\boldsymbol{x}; \mathbf{\Theta}'), \mathbf{\Theta} - \mathbf{\Theta}' \rangle| \leq \mathcal{O}(\omega^{1/3} L^2 \sqrt{m \log(m)}) \|\mathbf{\Theta} - \mathbf{\Theta}'\|$$

*with the probability at least $1 - \delta$.*

**Lemma G.10.** *Assume $m, \eta_1, \eta_2, J_1, J_2$ satisfy the conditions in **Theorem** 4.2 and $[\mathbf{\Theta}_{gnn}^{(1)}]_0$ being randomly initialized. Then, with probability at least $1 - \delta$ and given an arm $\|\boldsymbol{x}\|_2 = 1$, we have*

1. *$|f_u^{(1)}(\boldsymbol{x}; [\mathbf{\Theta}_{gnn}^{(1)}]_0)| \leq 2$,*

2. *$\|\nabla_{[\mathbf{\Theta}_{gnn}^{(1)}]_0} f_u^{(1)}(\boldsymbol{x}; [\mathbf{\Theta}_{gnn}^{(1)}]_0)\|_2 \leq \mathcal{O}(L)$.*

**Proof.** The conclusion (1) is a direct application of Lemma 7.1 in (Allen-Zhu et al., 2019). For conclusion (2), for each weight matrix $\mathbf{\Theta}_l \in \{\mathbf{\Theta}_0^{(1)}, \mathbf{\Theta}_1^{(1)}, \ldots, \mathbf{\Theta}_L^{(1)}\}$ where $\mathbf{\Theta}_0^{(1)} = \mathbf{\Theta}_{agg}^{(1)}$, we have

$$\|\nabla_{\mathbf{\Theta}} f_u^{(1)}(\boldsymbol{x}; [\mathbf{\Theta}_{gnn}^{(1)}]_0)\|_2 = \|(\mathbf{\Theta}_L \boldsymbol{D}_{L-1} \cdots \boldsymbol{D}_{l+1}\mathbf{\Theta}_{l+1}) \cdot (\boldsymbol{D}_{l+1}\mathbf{\Theta}_{l+1} \cdots \boldsymbol{D}_1\mathbf{\Theta}_1\boldsymbol{D}_0\mathbf{\Theta}_0) \cdot \boldsymbol{h}^{\mathsf{T}}\|_2 \leq \mathcal{O}(\sqrt{L})$$

by applying Lemma 7.3 in (Allen-Zhu et al., 2019), and $\boldsymbol{h}$ denotes the aggregated hidden representation for each user-pair, namely the corresponding row in $\boldsymbol{H}_{agg}$. Therefore, by combining the bounds for all the weight matrices, we could have

$$\|\nabla_{[\mathbf{\Theta}_{gnn}^{(1)}]_0} f_u^{(1)}(\boldsymbol{x}; [\mathbf{\Theta}_{gnn}^{(1)}]_0)\|_2 = \sqrt{\sum_{l \in \{0, \ldots, L\}} \|\nabla_{\mathbf{\Theta}} f_u^{(1)}(\boldsymbol{x}; [\mathbf{\Theta}_{gnn}^{(1)}]_0)\|_2^2} = \mathcal{O}(L).$$

which finishes the proof.

$\square$

**Lemma G.11** (Theorem 5 in (Allen-Zhu et al., 2019)). *Assume the training parameters $m, \eta_2, J_2$ satisfy the conditions in **Theorem** 4.2 and $[\mathbf{\Theta}_{gnn}^{(1)}]_0$ being randomly initialized. Then, with probability at least $1 - \delta$, and for all parameter $\mathbf{\Theta}_{gnn}^{(1)}$ such that $\|\mathbf{\Theta}_{gnn}^{(1)} - [\mathbf{\Theta}_{gnn}^{(1)}]_0\|_2 \leq \omega$, we have*

$$\|\nabla_{\mathbf{\Theta}_{gnn}^{(1)}} f_u^{(1)}(\boldsymbol{x}; \mathbf{\Theta}_{gnn}^{(1)}) - \nabla_{[\mathbf{\Theta}_{gnn}^{(1)}]_0} f_u^{(1)}(\boldsymbol{x}; [\mathbf{\Theta}_{gnn}^{(1)}]_0)\|_2 \leq \mathcal{O}(\omega^{1/3} L^3 \sqrt{\log(m)})$$

**Lemma G.12.** *Assume $m, \eta_2$ satisfy the condition in **Theorem** 4.2. With the probability at least $1 - \delta$, we have*

$$\sum_{\tau \in [t]} |f(\boldsymbol{x}_\tau; [\widehat{\mathbf{\Theta}}_{gnn}^{(1)}]_\tau) - r_\tau| \leq \sum_{\tau \in [t]} |f(\boldsymbol{x}_\tau; [\widehat{\mathbf{\Theta}}_{gnn}^{(1)}]_t) - r_\tau| + \frac{3L\sqrt{2t}}{2}$$

**Proof.** With the notation from Lemma 4.3 in (Cao & Gu, 2019), set $R = \frac{t^3 \log(m)}{\delta}$, $\nu = R^2$, and $\epsilon = \frac{LR}{\sqrt{2\nu t}}$. Then, considering the loss function to be $\mathcal{L}(\boldsymbol{\Theta}_{gnn}^{(1)}) := \sum_{\tau \in [t]} |f(\boldsymbol{x}_\tau; \boldsymbol{\Theta}_{gnn}^{(1)}) - r_\tau|$ would complete the proof. $\qquad\square$

**Lemma G.13.** *Consider a L-layer fully-connected network $f(\cdot; \boldsymbol{\Theta}_t)$ initialized w.r.t. Subsection 3.2.1. For any $0 < \xi_2 \le 1$, $0 < \rho \le \mathcal{O}(\frac{1}{L})$. Given the training data set with $t$ samples satisfying the unit-length and the $\rho$-separateness assumption, suppose the training parameters $m, \eta_2, J_2$ satisfy the conditions in **Theorem** 4.2. Then, with probability at least $1 - \delta$, we have*

$$|f(\boldsymbol{x}; \boldsymbol{\Theta}_t) - f(\boldsymbol{x}'; \boldsymbol{\Theta}_t)| \le \mathcal{O}(L) \cdot \|\boldsymbol{x} - \boldsymbol{x}'\|_2$$

$$\|\nabla_{\boldsymbol{\Theta}_t} f(\boldsymbol{x}; \boldsymbol{\Theta}_t) - \nabla_{\boldsymbol{\Theta}_t} f(\boldsymbol{x}'; \boldsymbol{\Theta}_t)\|_2 \le \mathcal{O}(\frac{tL^4 \log^{5/6}(m)}{\rho^{1/3} m^{1/6}}) + \mathcal{O}(L) \cdot \|\boldsymbol{x} - \boldsymbol{x}'\|_2$$

*when given two new samples $\boldsymbol{x}, \boldsymbol{x}'$.*

**Proof.** Denoting $\boldsymbol{D}_l$ to be the diagonal sign matrix of the $l$-th layer such that $\boldsymbol{D}_l[i, i] = \mathbb{I}[(\boldsymbol{\Theta}_l \boldsymbol{h}_{l-1})_i \ge 0], i \in [m]$, we could have

$$|f(\boldsymbol{x}; \boldsymbol{\Theta}_t) - f(\boldsymbol{x}'; \boldsymbol{\Theta}_t)| = |(\boldsymbol{\Theta}_L \boldsymbol{D}_{L-1} \cdots \boldsymbol{D}_1 \boldsymbol{\Theta}_1) \cdot (\boldsymbol{x} - \boldsymbol{x}')^\mathsf{T}|$$
$$\le \|\boldsymbol{\Theta}_L \boldsymbol{D}_{L-1} \cdots \boldsymbol{D}_1 \boldsymbol{\Theta}_1\|_2 \cdot \|\boldsymbol{x} - \boldsymbol{x}'\|_2.$$

Based on Lemma 7.3 from (Allen-Zhu et al., 2019) and Lemma C.4 from (Ban et al., 2022b), we have we have $\|\boldsymbol{\Theta}_L \boldsymbol{D}_{L-1} \cdots \boldsymbol{D}_1 \boldsymbol{\Theta}_1\|_2 = \mathcal{O}(L)$ for the initialized parameters $\boldsymbol{\Theta}_0 = \{[\boldsymbol{\Theta}_1]_0, \dots, [\boldsymbol{\Theta}_L]_0\}$. Meantime, after training the network and ending up with trained parameters $\boldsymbol{\Theta}_t = \{[\boldsymbol{\Theta}_1]_t, \dots, [\boldsymbol{\Theta}_L]_t\}$, according to Lemma 8.6 from (Allen-Zhu et al., 2019), the bound $\|\boldsymbol{\Theta}_L \boldsymbol{D}_{L-1} \cdots \boldsymbol{D}_1 \boldsymbol{\Theta}_1\|_2 = \mathcal{O}(L)$ still holds, which proves this statement.

Then, for the bound on the gradients, we have

$$\|\nabla_{\boldsymbol{\Theta}_t} f(\boldsymbol{x}; \boldsymbol{\Theta}_t) - \nabla_{\boldsymbol{\Theta}_t} f(\boldsymbol{x}'; \boldsymbol{\Theta}_t)\|_2$$
$$= \|\nabla_{\boldsymbol{\Theta}_t} f(\boldsymbol{x}; \boldsymbol{\Theta}_t) - \nabla_{\boldsymbol{\Theta}_0} f(\boldsymbol{x}; \boldsymbol{\Theta}_0) + \nabla_{\boldsymbol{\Theta}_0} f(\boldsymbol{x}; \boldsymbol{\Theta}_0) - \nabla_{\boldsymbol{\Theta}_0} f(\boldsymbol{x}'; \boldsymbol{\Theta}_0) + \nabla_{\boldsymbol{\Theta}_0} f(\boldsymbol{x}'; \boldsymbol{\Theta}_0) - \nabla_{\boldsymbol{\Theta}_t} f(\boldsymbol{x}'; \boldsymbol{\Theta}_t)\|_2$$
$$\le \|\nabla_{\boldsymbol{\Theta}_t} f(\boldsymbol{x}; \boldsymbol{\Theta}_t) - \nabla_{\boldsymbol{\Theta}_0} f(\boldsymbol{x}; \boldsymbol{\Theta}_0)\|_2 + \|\nabla_{\boldsymbol{\Theta}_0} f(\boldsymbol{x}; \boldsymbol{\Theta}_0) - \nabla_{\boldsymbol{\Theta}_0} f(\boldsymbol{x}'; \boldsymbol{\Theta}_0)\|_2 +$$
$$\|\nabla_{\boldsymbol{\Theta}_0} f(\boldsymbol{x}'; \boldsymbol{\Theta}_0) - \nabla_{\boldsymbol{\Theta}_t} f(\boldsymbol{x}'; \boldsymbol{\Theta}_t)\|_2.$$

Firstly, we have

$$\|\nabla_{[\boldsymbol{\Theta}_l]} f(\boldsymbol{x}; \boldsymbol{\Theta}_0)\|_2 = \|(\boldsymbol{\Theta}_L \boldsymbol{D}_{L-1} \cdots \boldsymbol{D}_{l+1} \boldsymbol{\Theta}_{l+1}) \cdot (\boldsymbol{D}_{l+1} \boldsymbol{\Theta}_{l+1} \cdots \boldsymbol{D}_1 \boldsymbol{\Theta}_1) \cdot \boldsymbol{x}^\mathsf{T}\|_2 \le \mathcal{O}(\sqrt{L})$$

based on Lemma 7.3 from (Allen-Zhu et al., 2019), and this leads to $\|\nabla_{\boldsymbol{\Theta}_0} f(\boldsymbol{x}; \boldsymbol{\Theta}_0)\|_2 \le \mathcal{O}(L)$. Analogously, we also derive

$$\|\nabla_{\boldsymbol{\Theta}_0} f(\boldsymbol{x}; \boldsymbol{\Theta}_0) - \nabla_{\boldsymbol{\Theta}_0} f(\boldsymbol{x}'; \boldsymbol{\Theta}_0)\|_2$$
$$= \|(\boldsymbol{\Theta}_L \boldsymbol{D}_{L-1} \cdots \boldsymbol{D}_{l+1} \boldsymbol{\Theta}_{l+1}) \cdot (\boldsymbol{D}_{l+1} \boldsymbol{\Theta}_{l+1} \cdots \boldsymbol{D}_1 \boldsymbol{\Theta}_1) \cdot (\boldsymbol{x} - \boldsymbol{x}')^\mathsf{T}\|_2 \le \mathcal{O}(L) \cdot \|\boldsymbol{x} - \boldsymbol{x}'\|_2.$$

Then, according to Theorem 5 from (Allen-Zhu et al., 2019) and with $\|\boldsymbol{\Theta}_0 - \boldsymbol{\Theta}_t\|_2 \le \omega$, we could have $\|\nabla_{\boldsymbol{\Theta}_0} f(\boldsymbol{x}; \boldsymbol{\Theta}_0) - \nabla_{\boldsymbol{\Theta}_t} f(\boldsymbol{x}; \boldsymbol{\Theta}_t)\|_2 \le \mathcal{O}(\omega^{1/3} L^2 \sqrt{\log(m)}) \cdot \|\nabla_{\boldsymbol{\Theta}_0} f(\boldsymbol{x}; \boldsymbol{\Theta}_0)\|_2$. Substituting the $\omega$ value with the conclusion from **Lemma** G.8, we could have

$$\|\nabla_{\boldsymbol{\Theta}_0} f(\boldsymbol{x}; \boldsymbol{\Theta}_0) - \nabla_{\boldsymbol{\Theta}_t} f(\boldsymbol{x}; \boldsymbol{\Theta}_t)\|_2 \le \mathcal{O}(\omega^{1/3} L^2 \sqrt{\log(m)}) \cdot \|\nabla_{\boldsymbol{\Theta}_0} f(\boldsymbol{x}; \boldsymbol{\Theta}_0)\|_2$$
$$= \mathcal{O}(\frac{tL^4 \log^{5/6}(m)}{\rho^{1/3} m^{1/6}}).$$

Finally, assembling all parts together will lead to the conclusion.

$\qquad\square$

**Lemma G.14.** *Consider a L-layer fully-connected network $f(\cdot; \boldsymbol{\Theta}_t)$ initialized w.r.t. Section 3.2.1. For any $0 < \xi_2 \le 1$, $0 < \rho \le \mathcal{O}(\frac{1}{L})$. Let there be two sets of training samples $\mathcal{P}_t, \mathcal{P}'_t$ with the unit-length and the $\rho$-separateness assumption, and let $\boldsymbol{\Theta}_t$ be the trained parameter on $\mathcal{P}_t$ while*

$\Theta_t'$ *is the trained parameter on* $\mathcal{P}_t'$. *Suppose* $m, \eta_1, \eta_2, J_1, J_2$ *satisfy the conditions in* ***Theorem*** *4.2. Then, with probability at least* $1 - \delta$, *we have*

$$|f(\boldsymbol{x}; \boldsymbol{\Theta}_t) - f(\boldsymbol{x}; \boldsymbol{\Theta}_t')| \leq$$
$$\left(1 + \mathcal{O}(\frac{tL^3 \log^{5/6}(m)}{\rho^{1/3}m^{1/6}})\right) \cdot \mathcal{O}(\frac{t^3 L}{\rho\sqrt{m}} \log(m)) + \mathcal{O}\left(\frac{t^4 L^2 \log^{11/6}(m)}{\rho^{4/3}m^{1/6}}\right)$$

*when given a new sample* $\boldsymbol{x} \in \mathbb{R}^d$.

**Proof.** First, based on the conclusion from Theorem 1 from (Allen-Zhu et al., 2019) and regarding the $t$ samples, the trained the parameters satisfy $\|\boldsymbol{\Theta}_t - \boldsymbol{\Theta}_0\|_2, \|\boldsymbol{\Theta}_t' - \boldsymbol{\Theta}_0\|_2 \leq \mathcal{O}(\frac{t^3}{\rho\sqrt{m}} \log(m)) = \omega$ where $\boldsymbol{\Theta}_0$ is the randomly initialized parameter. Then, we could have

$$\|\nabla_{\boldsymbol{\Theta}_t} f(\boldsymbol{x}; \boldsymbol{\Theta}_t)\|_2 \leq \|\nabla_{\boldsymbol{\Theta}_0} f(\boldsymbol{x}; \boldsymbol{\Theta}_0)\|_2 + \|\nabla_{\boldsymbol{\Theta}_t} f(\boldsymbol{x}; \boldsymbol{\Theta}_t) - \nabla_{\boldsymbol{\Theta}_0} f(\boldsymbol{x}; \boldsymbol{\Theta}_0)\|_2$$
$$\leq \left(1 + \mathcal{O}(\frac{tL^3 \log^{5/6}(m)}{\rho^{1/3}m^{1/6}})\right) \cdot \mathcal{O}(L)$$

w.r.t. the conclusion from Theorem 1 and Theorem 5 of (Allen-Zhu et al., 2019). Then, regarding the Lemma 4.1 from (Cao & Gu, 2019), we would have

$$|f(\boldsymbol{x}; \boldsymbol{\Theta}_t) - f(\boldsymbol{x}; \boldsymbol{\Theta}_t') - \langle \nabla_{\boldsymbol{\Theta}_t'} f(\boldsymbol{x}; \boldsymbol{\Theta}_t'), \boldsymbol{\Theta}_t - \boldsymbol{\Theta}_t' \rangle| \leq \mathcal{O}(\omega^{1/3} L^2 \sqrt{m \log(m)}) \cdot \|\boldsymbol{\Theta}_t - \boldsymbol{\Theta}_t'\|_2.$$

Therefore, the our target could be reformed as

$$|f(\boldsymbol{x}; \boldsymbol{\Theta}_t) - f(\boldsymbol{x}; \boldsymbol{\Theta}_t')| \leq \|\nabla_{\boldsymbol{\Theta}_t'} f(\boldsymbol{x}; \boldsymbol{\Theta}_t')\|_2 \|\boldsymbol{\Theta}_t - \boldsymbol{\Theta}_t'\|_2 + \mathcal{O}(\omega^{1/3} L^2 \sqrt{m \log(m)}) \cdot \|\boldsymbol{\Theta}_t - \boldsymbol{\Theta}_t'\|_2$$
$$\leq \left(1 + \mathcal{O}(\frac{tL^3 \log^{5/6}(m)}{\rho^{1/3}m^{1/6}})\right) \cdot \mathcal{O}(L) \cdot \omega + \mathcal{O}(\omega^{4/3} L^2 \sqrt{m \log(m)})$$

Substituting the $\omega$ with its value would complete the proof.

$\square$

## H  COMPUTATIONAL RESOURCES

All the experiments are conducted on a Windows machine with an Intel Core i7 CPU, 64GB RAM, and two RTX 5000 GPUs.

