# OpenReview forum: "Graph Neural Bandits"
_ICLR.cc/2023/Conference — Submitted to ICLR 2023_

### Official Review · Reviewer_tRgB · 2022-10-24

**Confidence:** 4
**Correctness:** 4
**Technical Novelty And Significance:** 2
**Empirical Novelty And Significance:** 2
**Recommendation:** 6

**Clarity, Quality, Novelty And Reproducibility:**

In my opinion, the contribution is limited as the paper is developed based on [1, 2], which made the contribution is incremental. It seems to me that the contribution is narrow down to instead of using exploitation and exploration neural networks [1], the authors constructed exploitation and exploration graph neural networks (GNNs) to solve the problem. Please see my comments/concerns in the Strength And Weaknesses section.

[1] Ee-net: Exploitation-exploration neural networks in contextual bandits. ICLR 2022.

[2] Neural collaborative filtering bandits via meta learning. ArXiv 2022.

**Strength And Weaknesses:**

Some strengths:

1. The paper tackles an interesting bandits problem by formulating the exploitation and exploration via graph neural networks. The paper has extensive experiments.
2. The proposed method seems to perform consistently across datasets according to Section 5 Experiments and Appendix B.
3. The paper is straightforward and easy to follow
4. It’s also good that the authors provided source code for reproducibility.


Some of my concerns/suggestions for improvements:

1. In my opinion, the contribution is limited as the paper is developed based on [1, 2], which made the contribution is incremental. It seems to me that the contribution is narrow down to instead of using exploitation and exploration neural networks [1], the authors constructed exploitation and exploration graph neural networks (GNNs) to solve the problem. Specifically, those graphs are weighed graphs in which each edge weight is an exploitation/exploration score.
2. Figure 2 shows cumulative regrets on four datasets: two from recommendation and two from classification. While they all show good results, it’s not trivial to understand why the ‘curves’ have much difference for the two recommendation datasets (i.e., almost linear) compared to the two classification datasets. Section 5.2 should give more detailed explanation and analyses. Why this method focuses particularly to personalized recommendation? Can it be generalized to other tasks?
3. While the authors provided multiple remarks and theoretical analyses on the complexity, I believe it would be better if the authors could also provide the run time of GNB and all the baselines. This is for us to have an overview if we should use the proposed method, or a single-bandit setting such as [1] is good enough, given that they all have similar bound. Moreover, I believe it could also give the readers a benchmark given the resources in Appendix I.
4. Figure 7 (Cumulative regrets for different exploration coefficients alpha) and Figure 8 (Figure 8: Cumulative regrets for different neighborhood hops k) show similar results. Thus, should we also study the relationships between alpha and k?
5. Also, according to Figure 8 and Appendix B.7, learning with 1-hop is good enough, and larger k (k=2,3) could potentially lead to over-smoothing problem. With this observation, it leads to a question that do we really need to build graphs for exploitation and exploration dilemma in the context of personalized recommendation? Building large-scale graphs in practice are also expensive even if we can use user neighbourhood approximation such as Appendix B.4. Please give more explanation/discussion.

Minor:

1. I think the title should be more specific towards recommendation instead of only “Graph Neural Bandits”
2. Appendix H Technical Lemmas is redundant in my opinion


Overall, I am personally a bit concerned about the paper’s novelty (i.e., contribution).

[1] Ee-net: Exploitation-exploration neural networks in contextual bandits. ICLR 2022.

[2] Neural collaborative filtering bandits via meta learning. ArXiv 2022.


**Summary Of The Paper:**

Summary

This paper introduces Graph Neural Bandits (GNB) in the context of personalized recommendation. Specifically, the authors leverage graph neural networks to learn users’ “coarse-grained” collaborative instead of “fine-grained” for bandits optimization. All in all, the proposed method shows good performance across nine algorithms/baselines on various datasets and the authors also provide the theoretical analysis for regret upper bound of complexity.


**Summary Of The Review:**

Overall, I am personally a bit concerned about the paper’s novelty (i.e., contribution). The authors did great job about the experiments and theoretical analyses, but some minor parts could be improved.

=====

Increase my score from 3 to 6 after the rebuttal

---

> ### Author Response · Authors · 2022-11-15
> **To Fourth Reviewer, tRgB (4)**
>
>
> **REFERENCES**
>
> Yikun Ban and Jingrui He. Local clustering in contextual multi-armed bandits. In Proceedings of
> the Web Conference 2021, pp. 2335–2346, 2021.
>
> Yikun Ban, Yunzhe Qi, Tianxin Wei, and Jingrui He. Neural collaborative filtering bandits via meta
> learning. arXiv preprint arXiv:2201.13395, 2022a.
>
> Yikun Ban, Yuchen Yan, Arindam Banerjee, and Jingrui He. Ee-net: Exploitation-exploration neural
> networks in contextual bandits. In International Conference on Learning Representations, 2022b.
>
> Claudio Gentile, Shuai Li, and Giovanni Zappella. Online clustering of bandits. In ICML, pp.
> 757–765, 2014.
>
> Yunzhe Qi, Yikun Ban, and Jingrui He. Neural bandit with arm group graph. arXiv preprint
> arXiv:2206.03644, 2022.
>
> Weitong Zhang, Dongruo Zhou, Lihong Li, and Quanquan Gu. Neural thompson sampling. In
> International Conference on Learning Representations, 2021.
>
> Dongruo Zhou, Lihong Li, and Quanquan Gu. Neural contextual bandits with ucb-based exploration.
> In International Conference on Machine Learning, pp. 11492–11502. PMLR, 2020.

---

> > ### Comment · Reviewer_tRgB · 2022-11-23
> > **Thanks for the response**
> >
> > Thank you for the detailed responses. I'm still skeptical about the need of using k-hop with k > 1. As I pointed out, 1-hop seems to provide good enough results, but I do not see clear explanation on why 2,3-hop did not yield much better performance. In addition, I also shared the same concern with Reviewer WB9s about real application (thus, that's why I asked about run time). I have read the authors' rebuttal but I do not see directly how real world systems would benefit from this setting. Besides the provided complexity, I believe it would be more convincing if the authors could provide the 'link' between the proposed framework and real world production system.
> >
> > Nevertheless, I increased my score after carefully reviewing the revised version.

---

> > > ### Author Response · Authors · 2022-11-25
> > > **Further Discussion with Fourth Reviewer (3)**
> > >
> > > ***(2) [Online Multi-task Learning]***
> > >
> > > Our proposed GNB can potentially be extended to other application scenarios, such as the online multi-task learning (Dekel et al., 2007; Xu et al., 2017).
> > > For a total of $n$ different tasks, we can construct a task graph, where each node corresponds to a single task and the task correlations are formulated by the edge weights among task nodes.
> > > Here, the classification model will be trained incrementally instead of being trained on a given training dataset.
> > >
> > > For instance, each task here can be an image classification problem. In this case, suppose we have three tasks (nodes) which are the image classification problems on polar animals, domestic animals, and terrestrial animals separately.
> > > Intuitively, the task of domestic animals is more correlated with the task of terrestrial animals, compared with the polar animals task.
> > > Thus, this offers us a great opportunity to leverage the knowledge from the task of domestic animals to help improve our prediction results for the task of terrestrial animals, or vice versa. GNB has the ability to formulate and leverage these correlations.
> > >
> > > Suppose the task of domestic animal classification involves $c$ different classes. After receiving a sample $x$ for the task of domestic animals, we can transform $x$ into $c$ different vectors following our empirical approach of dealing with classification data sets (Subsection 5.1) and feed into the GNB models. Here, each of the $c$ vectors can be deemed as an arm, and the corresponding class of the chosen arm will be our predicted image class for sample $x$.
> > > In this way, GNB can leverage the collaboration among different tasks to improve the prediction performance, especially for the tasks with insufficient samples.
> > >
> > > Since the online multi-task learning setting is out of the scope of this paper, we will explore this direction in future works.
> > >
> > >
> > >
> > >
> > > **REFERENCES**
> > >
> > >
> > > Ofer Dekel, Philip M Long, and Yoram Singer. Online learning of multiple tasks with a shared loss.
> > > Journal of Machine Learning Research, 8(10), 2007.
> > >
> > > Claudio Gentile, Shuai Li, and Giovanni Zappella. Online clustering of bandits. In ICML, pp.
> > > 757–765, 2014.
> > >
> > > Felix Wu, Amauri Souza, Tianyi Zhang, Christopher Fifty, Tao Yu, and Kilian Weinberger. Simplifying graph convolutional networks. In International conference on machine learning, pp.
> > > 6861–6871. PMLR, 2019.
> > >
> > > Jianpeng Xu, Pang-Ning Tan, Jiayu Zhou, and Lifeng Luo. Online multi-task learning framework
> > > for ensemble forecasting. IEEE Transactions on Knowledge and Data Engineering, 29(6):1268–
> > > 1280, 2017.3

---

> > > ### Author Response · Authors · 2022-11-25
> > > **Further Discussion with Fourth Reviewer (2)**
> > >
> > > **Q2: The link between GNB and real-world production systems?**
> > >
> > > To better bridge our proposed GNB with the real-world applications, we would like to provide more details for our proposed GNB under two potential real-world application scenarios.
> > >
> > > ***(1) [Online Recommendation with User Collaborations]***
> > >
> > > Let us consider an application scenario for movie recommendation.
> > > Suppose there are a total of $n$ users in the online movie recommendation platform. At each time step $t\in [T]$, we will receive a user $u_{t}$ and a set of candidate arms $\mathcal{X_{t}}$ (i.e., movies). The learner will then need to select one arm $x_{t} \in \mathcal{X}_{t}$ as the recommendation to the user.
> > >
> > > As we have discussed, the preference of the target user $u_{t}$ can be influenced by other correlated users (e.g., his/her friends) due to the user collaborative effect (Gentile et al., 2014).
> > > Therefore, given an arm $x_{i, t}\in \mathcal{X_{t}}$, we can apply GNB to form the user exploitation / exploration graphs as the representation of user correlations. Here, the nodes within the user graphs can be either: (1) the target user $u_{t}$ and his/her friends; or, (2) based on Remark 3.2, the target user $u_{t}$ and a batch of "representative users".
> > > In this case, the "fine-grained" user correlation magnitude between target user $u_{t}$ and a user $u$ in the user graph is preserved by the edge weight between these two nodes, which is utilized by GNB for the downstream reward / potential gain estimations.
> > >
> > > As we only have limited interactions with each single user, one natural approach is to leverage the correlations among different users to improve the recommendation quality.
> > > Moreover, utilizing the user collaborative effects can also help to reduce the recommendation error when we are serving users with insufficient interactions, such as new users who just join the platform. This motivates us to design GNB.
> > >
> > >
> > >
> > > Another major challenge in online recommendation is the exploitation-exploration dilemma.
> > > For example, if we only greedily choose the movies with plenty of user interactions based on the existing knowledge, we may neglect a considerable number of good movies with less user interactions, and it can also be hard for the learner to adapt to new movies with no user iterations.
> > > In this case, we need to decide whether to choose the arms with sufficient user interactions (e.g., popular action movies) so that we will be certain about our decision; or, it is better to explore audience-specific arms (e.g., artistic films) with relatively less user interactions.
> > >
> > > Therefore, we use the second GNN model, the exploration GNN model, to achieve adaptive exploration with the help of estimated user exploration correlations. Intuitively, when the exploration user correlation between two users $u_{i}, u_{j}$ is strong, we may want to apply similar exploration strategies for both of them. For instance, given arm $x$, if neither of these two users have past interactions with a similar arm $x$, we will need to explore both of these two user-arm pairs $(x, u_{i}), (x, u_{j})$ more for additional knowledge.
> > > Compared with existing works, it is more adaptive since GNB does not impose the same exploration strategy for all the users within the same rigid user group.
> > > This is also the key design that enables GNB to both achieve a sharper theoretical regret bound and outperform all the baselines across real data sets.
> > >
> > > Finally, to tackle the potentially large number of users, we also provide two heuristic methods to reduce the computational cost of GNB: (1) sampling strategy to reduce the input dimensionality of $f_{gnn}^{(2)}(\cdot)$ as in Remark 3.1;  and, (2) the representative users to reduce the size of user graphs as in Remark 3.2.
> > > The effectiveness of these two approaches is supported by experiments (we apply Remark 3.1 for all the experiments, and conduct experiments for Remark 3.2 in appendix subsection B.5).

---

> > > ### Author Response · Authors · 2022-11-25
> > > **Further Discussion with Fourth Reviewer (1)**
> > >
> > > We sincerely appreciate the fruitful discussion with the reviewer and want to thank you again for your comments.
> > > Here, we would like to further clarify the remaining of your concerns.
> > >
> > > **Q1:  About the aggregation hops $k$ and the benefit of including the user graphs?**
> > >
> > > We agree with you that the necessity of using $k > 1$ needs further discussion.
> > > Thus, we conduct the experiments on the MovieLens data set with larger user graphs (100 users) to further investigate the influence of the parameter $k$. Meanwhile, given two input vectors $w, v$, we apply the RBF kernel as the mapping functions $\Psi^{(1)}(w, v) = \Psi^{(2)}(w, v) = \exp(-\gamma\cdot \|\|w - v\|\|^{2})$ where $\gamma$ is the kernel bandwidth parameter.
> > > The experiment results are shown in the table below, and the value in the brackets "[]" is the element standard deviation of the normalized adjacency matrix of user exploitation graphs.
> > >
> > > ***Table 1: Cumulative regrets on MovieLens dataset with 100 users (different k / kernel bandwidth). The value in the brackets "[]" is the element standard deviation of the normalized adjacency matrix.***
> > >
> > > | ($k$ \ Bandwidth $\gamma$) |0.1| 1| 2| 5|
> > > |---  |--- |--- |--- |--- |
> > > |1    | 7276 [$1.6\times 10^{-4}$]   |  7073 [$1.4\times 10^{-3}$] |  7151 [$2.2\times 10^{-3}$] |  7490 [$3.9\times 10^{-3}$]
> > > | 2  |   6968 [$1.0\times 10^{-4}$] | 6966 [$7.7\times 10^{-4}$] | 7074 [$1.3\times 10^{-3}$] |  7087 [$2.5\times 10^{-3}$]
> > > | 3  |  7006 [$7.1\times 10^{-5}$] | 7018 [$7.0\times 10^{-4}$] |  6940 [$1.2\times 10^{-3}$] |  7167 [$1.9\times 10^{-3}$]
> > >
> > > Here, we have the following remarks:
> > >
> > > (1) Increasing the value of parameter $k$ will make the normalized adjacency matrix elements more "smooth", as we can see from the decreasing standard deviation values. This matches the low-pass nature of multi-hop feature propagation (Wu et al., 2019).
> > > With a larger $k$ value for information propagation, GNB will be able to incorporate the multi-hop user neighborhood. But overly large $k$ values can lead to the "over-smoothing" problem.
> > > In contrast, with a smaller $k$ value, it is possible that the target user is "heavily influenced" by only several specific users.
> > > Therefore, the practitioner may need to choose $k$ value properly under different application scenarios.
> > >
> > >
> > > (2) For a larger user graph, propagating multiple hops (larger $k$ values) seems to be more beneficial. One possible explanation can be that with a larger user graph, the user correlations will become more complex.
> > > Thus, instead of only utilizing the information directly from the target user's neighbors, we need to involve the neighborhood information of the target user's neighbors for a global perspective over the users. In this case, properly increasing the $k$ value will indeed enable GNB to obtain a more comprehensive neighborhood information around the target user, which leads to a better performance when working with the large user graph.
> > > However, as we have mentioned, overly large $k$ values can also lead to additional computational costs and the ``over-smoothing'' problem, which can impair the model performance based on the experiments above.
> > >
> > >
> > > We will add these experiment results and discussions to the manuscript.

---

> ### Author Response · Authors · 2022-11-15
> **To Fourth Reviewer, tRgB (3)**
>
> **Q: The running time of GNB and all the baselines?**
>
> Thanks for your suggestion. We have included the running time of GNB and baselines in the updated manuscript (Appendix B.8).
>
> Here, we have the following remarks:
>
> (1)
> Although the other baselines, especially the linear baselines tend to run much faster compared with neural algorithms, their experimental performance is not comparable with our proposed GNB as their linear assumption is too strong for most application scenarios.
> In particular, for the data set with large arm context dimension $d$ (e.g., the MNIST data set with $d=784$), the mapping from the arm context to the reward will be much more complicated. In this case, the neural algorithms manage to achieve the considerable improvement over the linear algorithms with reasonable running time.
>
> (2)
> From Table 1, we see that compared with the most closely related work, Meta-Ban (Ban et al.,
> 2022a), our proposed GNB is generally faster (GNB takes at most 75\% running time of Meta-Ban on various data sets), since GNB does not required to re-train the model for each candidate arm.
>
> (3)
> Based on the running time in the brackets “[]”, we see that for the two recommendation tasks, GNB takes approximately $\sim 0.4$ second per round to make the arm recommendation.
> In particular, since we are actually estimating the rewards for all the nodes (users) within the user graph (or the “approximated” user graph), GNB is able to handle multiple users in each round simultaneously without running the recommendation procedure multiple times for each target user, which is more efficient in real-world cases.
>
> (4)
> In all the experiments, we train the GNB framework per 100 rounds after $T > 1000$ and still manage to achieve good performance. Thus, the running time of GNB in the long run could be further significantly improved by reducing the training frequency since we already have enough data and an accurate framework.
>
> **Q: Correlations between exploration parameter $\alpha$ and the aggregation hops $k$?**
>
> Thanks for your suggestion. We have included the cumulative regret results on the MNIST data set in the updated manuscript (Section B.3, Figure 6) with different $k$, $\alpha$ value pairs.
>
>
> **Q: Is it necessary to build the user graphs?**
>
> Different from the local clustering methods, e.g., Ban & He, 2021, where they directly learn from the target user's neighborhood and the neighbors are treated the same, our proposed GNB allows the other users to contribute differently to the decision making process by modeling the magnitude of user collaborations through edge weights.
> Meanwhile, since user graphs also involve the neighborhood information of users' neighbors, GNB manages to provide a global perspective over the users by encoding such information into the gradient $\nabla f_{gnn}^{(1)}(\cdot)$ as the input of the exploration GNN model $f_{gnn}^{(2)}(\cdot)$. In this way, we achieve adaptive exploration by leveraging the user exploration correlation.
> Compared with existing works, modeling the fine-grained user correlations with user graphs is also the key design that enables GNB to both achieve a sharper theoretical regret bound and outperform all the baselines across real data sets.
>
> As we discussed in Remark 3.2 and related experiments, when facing a large number of users, we can always construct the user graphs for the received target user and the “representative users” only (i.e., with a total of $\tilde{n} + 1$ users / nodes, where $\tilde{n} << n$), which will be much smaller in size. In this case, we can significantly reduce the running time and maintain decent performance.
>
> As we have discussed in the previous question, for the single-bandit algorithms (e.g., Zhou et al.,
> 2020; Ban et al., 2022b), we can only assume: (1) all users have the same preference and will be treated the same; or (2) all the users are different and they will have distinct preferences. The first formulation is clearly unrealistic under the real-world application scenarios. On the other hand, if we assign each user with his/her own bandit model, these individual bandit estimators will be trained based on this user's information only with no collaborations among different estimators, which will make the whole system vulnerable to the data sparsity and the cold-start problem (Gentile et al., 2014). Meanwhile, as we mentioned before, assigning each user with a single bandit estimator will also inevitably lead to an additional $\sqrt{n}$ term in the theoretical regret bound.

---

> ### Author Response · Authors · 2022-11-15
> **To Fourth Reviewer, tRgB (2)**
>
>
>
> **Q: “Linear curve” for recommendation data sets?**
>
> It is usual to exhibit linear curves in experiments because the datasets often contain inherent noise, which makes the prediction accuracy relatively low. This leads to the “linear-like” curves in the figures.
> For instance, in existing works (Zhou et al., 2020; Zhang et al., 2021; Ban et al., 2022a; Qi
> et al., 2022; Ban et al., 2022b), we can all observe this kind of “linear-like” cumulative regrets for the recommendation datasets.
> One possible reason is that these two recommendation data sets contain considerable inherent noise, which makes the algorithm hard to learn the correct reward mapping function. Thus, the experimental improvement on these two data sets is non-trivial. As in the existing works, the baselines as well as their proposed algorithms will behave similarly, and the performance difference is small.
>
>
>
> **Q: Can GNB be generalized to other tasks?**
>
> In this paper, we mainly focus on solving the classical user collaboration problem in contextual bandits, where each user will be a node in the user graph.
> Intuitively, our framework can also be generalized to other tasks such as multi-task learning. For example, for image classification, we can consider each class to be a node in the image-class graph, where the class correlations are represented by edge weights. Here, when a new image comes, we can proceed to evaluate the edge weights of the image-class graph and utilize the derived graph for classification. In particular, under the settings of few-shot learning, our adaptive exploration strategy can also offer great help.
> But this direction is beyond the scope of this paper and will be explored as the future work.

---

> ### Author Response · Authors · 2022-11-15
> **To Fourth Reviewer, tRgB (1)**
>
> We sincerely thank the reviewer for the detailed comments and suggestions, and we will address the questions and concerns in the form of Q\&A.  Please also refer to our updated “Rebuttal Revision” manuscript for updates.
>
>
>
> **Q: Novelty concern of the paper?**
>
> In the original manuscript, we have discussed how this work is different from the mentioned two works EE-Net (Ban et al., 2022b) and Meta-Ban (Ban et al., 2022a) from various perspectives including problem definition, architecture, theoretical results and experiments.
> The novelty and the contributions of our work have been acknowledged by the other three reviewers.
> Here, we would like to clarify our contributions from the following perspectives:
>
> (1) [Problem Definition] To the best of our knowledge, this is the first work that proposes to model the “fine-grained” user correlations under the contextual bandits settings. All the existing user clustering works with “coarse-grained” user correlations, e.g., Meta-Ban (Ban et al., 2022a), impose fixed user clusters where users within the same cluster are forced to share the identical preferences, and users across different clusters are not correlated at all. This assumption is too strong for real application scenarios. Meanwhile, single-bandit algorithms like EE-Net (Ban et al., 2022b) does not incorporate any user collaborations, which means that either (i) all users are forced to share the same preference; or (ii) we will need to apply individual bandit models for each single user, with no collaboration across different models. This also does not fit into the real-world cases.
>
> Compared with them, we allow users to contribute differently to the final decision-making process instead of applying rigid user clusters. Therefore, our problem definition is more generic, and can be easily generalized to existing problem settings.
> Meanwhile, our non-parametric reward mapping function is also more generic compared with existing works, as it does not assume a fixed form of the reward mapping function (which can be linear / non-linear).
>
>
>
> (2) [Framework Design] To model “fine-grained” user correlations, we need to make the proposed neural framework incorporate both the arm contexts and the user collaborative effects. Here, we adopt two separate GNN models to separately leverage the user collaborative effects in terms of exploitation and exploration. Compared with existing works with UCB-based exploration strategy, e.g., Meta-Ban (Ban et al., 2022a), or algorithms with no user collaborations, our GNB framework needs more careful design to accommodate the generic reward mapping function to pave the way for the following theoretical analysis, which is distinct from existing works.
>
> (3) [Empirical Improvement] To support our claim, we conduct extensive experiments to demonstrate the effectiveness of our proposed framework, and GNB manages to achieve better performance compared with nine SOTA baselines, including the two SOTA works mentioned by the reviewer, i.e., EE-Net (Ban et al., 2022b) and Meta-Ban (Ban et al., 2022a).
>
> (4) [Theoretical Improvement]
> As our proposed GNB is distinct from existing user clustering works, we need to formulate theoretical analysis from a new perspective, which is non-trivial considering GNB is the first of its kind.
> With the proposed GNB framework, we have achieved non-trivial theoretical improvements over existing works (Remarks 4.3-4.4, Remark E.1), and our regret bound is sharper than existing works. First, our martingale-based theoretical analysis helps us to remove the i.i.d. assumption of arms (conventional for user clustering works, such as such as (Ban et al., 2022a; Gentile et al., 2014; Ban & He, 2021)) and also remove the terms $d$ and $\tilde{d}$ (typical for linear algorithms and neural algorithms with regression-based analysis (Zhou et al., 2020; Zhang et al., 2021)). Moreover, with the design of the GNB framework as well as our new theoretical analysis, we manage to reduce the cost of involving user collaborations from $\sqrt{n}$  to $\sqrt{\log(n)}$ ($n$ is the number of users), which has not been achieved by existing works.
>
>
> Moreover, existing user clustering works all demand a $\sqrt{n}$ term as the cost to incorporate user collaborative effects, because they are applying separate bandit models for each estimated rigid user cluster. Note that if we try to assign each user with an individual single-bandit model, e.g., EE-Net (Ban et al., 2022b), we all also include a $\sqrt{n}$ term as the cost.
> On the other hand, as we discussed above, our carefully designed GNB framework only utilizes two GNN models to achieve exploitation and adaptive exploration, which enables us to reduce the cost to $\sqrt{\log(n)}$.

---

### Official Review · Reviewer_WB9s · 2022-10-24

**Confidence:** 3
**Correctness:** 3
**Technical Novelty And Significance:** 3
**Empirical Novelty And Significance:** 2
**Recommendation:** 6

**Clarity, Quality, Novelty And Reproducibility:**

The proposed problem and the learning process are relatively novel.
The paper is well written.
The code is supplied by authors.

**Details Of Ethics Concerns:**

none.

**Strength And Weaknesses:**

Although the motivation of this paper is well and the idea of the proposed framework is well defined, However, I have the following concerns about its learning process, running and space complexity.

I feel that its complexity seems to be very high, which will hinder its real application.
1.	This paper tries to model the fine-grained collaborative effects of individual users and the arms. This motivation is straightforward, however, it may make the scale of the network be larger significantly. In a real online recommendation system, there would be several billions of requests from millions of different users. These users are evolving as the time. If we try to model the individual user in the bandit setting, the network should be larger extremely, which may highlight the sparsity problem of the data. Is it necessary to model the individual user after taken the large n into consideration? On the other hand, if we have a small number of users for an application scenario, the data may enough for the exploitation. In this situation, the advantage of exploration is relatively limited.
2.	In the learning framework, the method will construct two graphs continuously. The space and running time complexity should be considered. I feel the complexity would be larger significantly to obtain the fine-grained reward improvement. So I suggest to compare the complexity except cumulative reward.
3.	Although the learning framework is clear, the convergence of the exploration and exploitation GNN should be given. These two GNN are dependent. Does the structure of exploitation gnn have effect on the convergence of exploration gnn and vice versa? Will the prediction of the exploration gnn be smaller as the increase of the data?


**Summary Of The Paper:**

This paper proposes a new model called Graph Neural Bandits (GNB) to leverage the collaborative nature among users empowered by graph neural networks (GNNs). Instead of estimating clusters of users, the method models the “fine-grained” collaborative effects through estimated user graphs in terms of exploitation and exploration individually. Based on EE networks, the paper utilizes separate GNN-based models for exploitation and adaptive exploration respectively. Theoretical analysis and experimental results on multiple real data sets in comparison with state-of-the-art baselines are provided to demonstrate the effectiveness of the proposed framework.

**Summary Of The Review:**

see detailed Strength And Weaknesses section.

---

> ### Author Response · Authors · 2022-11-15
> **To Third Reviewer, WB9s (2)**
>
>
>
> **Q: Space and time complexity?**
>
> In each round $t$, we will need to iterate through user models and construct the user graphs. To calculate the output of user models, we will need $O(n)$ time. Although we need $O(n^{2})$ time to calculate the exploitation/exploration scores for user pairs based on the user model output, this procedure will not take too much time since applying the kernel functions on scalar values is fast and can be highly parallelized.
> Moreover, as we discussed in Remark 3.2, we can adopt the “representative users” to approximately model the “fine-grained” user correlations, which will significantly reduce the running time as we will end up with a small number of users (nodes) for the user graphs.
>
> Suppose that for neural models with no user collaborations (Zhou et al., 2020; Qi et al., 2022; Zhang
> et al., 2021), we assign each user with a separate neural model. In this case, for neural algorithms using the UCB (Zhou et al., 2020; Qi et al., 2022) and Thompson sampling (Zhang et al., 2021) to achieve exploration, they will need the space complexity of $O(n\cdot p^{2})$ ($p$ is the number of parameters in the neural model, which is a considerably large number) to store their gigantic gradient matrix, which demands large amount of running memory especially when the neural model is large. EE-Net (Ban et al., 2022b) has the space complexity of $O(n\cdot p\cdot T)$.
> Compared with them, our space complexity is no larger than $ 2T\cdot n^{2}  + O(n\cdot p\cdot T)$, and applying the approximation method in Remark 3.1 can further significantly reduce the space complexity. Here the first term $2T\cdot n^{2}$ is the cost of storing past user graph adjacency matrices.
>
>
> **REFERENCES**
>
> Yikun Ban and Jingrui He. Local clustering in contextual multi-armed bandits. In Proceedings of
> the Web Conference 2021, pp. 2335–2346, 2021.
>
> Yikun Ban, Yunzhe Qi, Tianxin Wei, and Jingrui He. Neural collaborative filtering bandits via meta
> learning. arXiv preprint arXiv:2201.13395, 2022a.
>
> Yikun Ban, Yuchen Yan, Arindam Banerjee, and Jingrui He. Ee-net: Exploitation-exploration neural
> networks in contextual bandits. In International Conference on Learning Representations, 2022b.
>
> Yunzhe Qi, Yikun Ban, and Jingrui He. Neural bandit with arm group graph. arXiv preprint
> arXiv:2206.03644, 2022.
>
> Weitong Zhang, Dongruo Zhou, Lihong Li, and Quanquan Gu. Neural thompson sampling. In
> International Conference on Learning Representations, 2021.
>
> Dongruo Zhou, Lihong Li, and Quanquan Gu. Neural contextual bandits with ucb-based exploration.
> In International Conference on Machine Learning, pp. 11492–11502. PMLR, 2020.3

---

> ### Author Response · Authors · 2022-11-15
> **To Third Reviewer, WB9s (1)**
>
> We sincerely thank the reviewer for the detailed comments and suggestions, and we will address the questions and concerns in the form of Q\&A.  Please also refer to our updated “Rebuttal Revision” manuscript for updates.
>
>
>
>
> **Q: Is it necessary to model the individual user? The data sparsity problem?**
>
> If the number of users $n$ is large, but we are at an early stage of recommendation ($T$ is small), there will exist the data sparsity problem. However, this is the exact reason that we apply only two GNN models to make the arm recommendations, where these two GNN models can leverage the past records for all the users from past rounds $t\in [T]$.
> On the contrary, if we assign each user a separate bandit model, it will make recommendations based on the records from one single user only ($T/n$ records in expectation).
> This is also the one major reason that GNB manages to outperform all the baselines in the experiments, since it can both leverage all the past records from different users and preserve the user collaborative effect simultaneously.
> But, the existing methods including user clustering algorithms (e.g., Ban et al., 2022a; Ban & He, 2021) and single-bandit algorithms (e.g., Zhou et al., 2020) will not be able to achieve these two objectives at the same time.
>
> Meanwhile, the design of GNB with only two GNN models also enables us to make the nontrivial improvements of removing the effective NTK dimension $\tilde{d}$, and reducing $\sqrt{n}$ to $\sqrt{\log(n)}$ in the theoretical regret bound.
>
>
>
>
>
> **Q: With a small number of users for an application scenario, the advantage of exploration is relatively limited?**
>
> Even with a small number of users (i.e., $n$ is small), the exploration is still necessary since we may continuously have the unseen user-arm pairs showing up.
> For instance, for the news streaming platform, we will need to decide on what article should be recommended to the target user in real time, where the articles are usually unobserved before.
> Since the platform usually needs to recommend different types of articles to the target user, the past interactions cannot necessarily be directly generalized to the current recommendation.
> If we want to decide whether to recommend a piece of political news to the target user, who has been browsing the political articles on our platform, we will likely be certain about our decision.
> However, if we only have the interactions with this target user over sports news, we will be less certain about our recommendation. In this case, our adaptive exploration strategy will take effect to help us better predict the target user's preference based on past interactions.
>
>
>
> **Q: Convergence of the GNN models?**
>
>
> In our theoretical analysis, we also provided the convergence analysis for the user models (Lemma F.2, F.4) and GNN models (Lemma G.2, G.3), which show that the generalization error of the components of our framework is bounded.
> Based on our theoretical analysis, when the network depth $L$ grows, we will need smaller learning rates $\eta_{1}, \eta_{2}$ and more iterations to make the framework converge. Similarly, with larger network width $m$, we will also need to reduce the learning rate to ensure the network convergence. Meanwhile, as we shown in Lemma G.2, G.3, since the cumulative estimation error of the exploitation GNN and exploration GNN models will grow in a sub-linear manner along with rounds $T$, the prediction of exploration will become smaller in expectation along with more rounds $T$ of the recommendation.

---

> ### Author Response · Authors · 2022-12-01
> **Looking forward to more discussions**
>
> Dear Reviewer WB9s,
>
> We would like to thank you again for your detailed comments and questions. As it has been around two weeks since we posted the response to your question and the deadline for discussion is approaching, we are wondering if you can kindly let us know whether our response has addressed your questions and concerns properly. Please kindly let us know if you have further questions or concerns. We will try our best to address them in order to make ourselves clear.
>
> Thanks again for your time and the discussion.
>
> Sincerely,
>
> Authors

---

### Official Review · Reviewer_hbM8 · 2022-10-25

**Confidence:** 4
**Correctness:** 4
**Technical Novelty And Significance:** 3
**Empirical Novelty And Significance:** Not applicable
**Recommendation:** 5

**Clarity, Quality, Novelty And Reproducibility:**

The contribution of this work is important since it is the first to analyze the fine-grained user collaboration. The GNB framework proposed in this paper is also novel. This paper is also technically solid and well-organized.

**Strength And Weaknesses:**

Strength:

1. The idea of using the user graph to model the fine-grained user collaborative effects is new. The proposed framework GNB is also novel.

2. The regret bound illustrated in this paper is sharper than the existing results. They remove the term $d$ and reduce $\sqrt{n}$ to $\sqrt{\log(n)}$.

Weakness:

1. I’m confused about the term $d$ in the regret bound. Can you give more explanation about how to remove the term $d$ in the analysis?

2. The regret theorem requires $m > \Omega(poly(T))$, which might be very large when round $T$ is sufficiently large.

3. Can you give more explanations about the exploration user graph? How about estimating the potential gain based only on a single user and arm?

4. The experimental results in the MovieLens dataset and Yelp dataset seem not to converge. Maybe the algorithm needs to run more rounds to show the convergence.

5. It will be better if the author could discuss more the comparison between the linear algorithm and neural architecture algorithm, and the comparison between the algorithm considering user collaboration and the algorithm considering no user collaboration.

**Summary Of The Paper:**

This paper proposes a novel framework named Graph Neural Bandits (GNB) to model the fine-grained user collaborative effects where they model the user correlation by the user graph. A GNN-based model is proposed to achieve adaptive exploration. They also show an O(\sqrt{T\log(Tn)}) regret upper bound of the GNB algorithm. Extensive experiments show the efficiency of GNB compared with state-of-the-art algorithms.

**Summary Of The Review:**

This paper proposes a novel framework named GNB to model fine-grained user collaboration. It also gives a sharper regret bound of the GNB algorithm compared with existing results. It will be better if the author could give more discussions about the experiments.

---

> ### Author Response · Authors · 2022-11-15
> **To Second Reviewer, hbM8 (4)**
>
>
> **REFERENCES**
>
> Zeyuan Allen-Zhu, Yuanzhi Li, and Zhao Song. A convergence theory for deep learning via over-
> parameterization. In International Conference on Machine Learning, pp. 242–252. PMLR, 2019.
>
> Yikun Ban and Jingrui He. Local clustering in contextual multi-armed bandits. In Proceedings of
> the Web Conference 2021, pp. 2335–2346, 2021.
>
> Yikun Ban, Jingrui He, and Curtiss B Cook. Multi-facet contextual bandits: A neural network
> perspective. In Proceedings of the 27th ACM SIGKDD Conference on Knowledge Discovery &
> Data Mining, pp. 35–45, 2021.
>
> Yikun Ban, Yunzhe Qi, Tianxin Wei, and Jingrui He. Neural collaborative filtering bandits via meta
> learning. arXiv preprint arXiv:2201.13395, 2022a.
>
> Yikun Ban, Yuchen Yan, Arindam Banerjee, and Jingrui He. Ee-net: Exploitation-exploration neural
> networks in contextual bandits. In International Conference on Learning Representations, 2022b.
>
> Yuan Cao and Quanquan Gu. Generalization bounds of stochastic gradient descent for wide and
> deep neural networks. Advances in Neural Information Processing Systems, 32:10836–10846,
> 2019.
>
> Claudio Gentile, Shuai Li, and Giovanni Zappella. Online clustering of bandits. In ICML, pp.
> 757–765, 2014.
>
> Claudio Gentile, Shuai Li, Purushottam Kar, Alexandros Karatzoglou, Giovanni Zappella, and
> Evans Etrue. On context-dependent clustering of bandits. In ICML, pp. 1253–1262, 2017.
>
> Parnian Kassraie, Andreas Krause, and Ilija Bogunovic. Graph neural network bandits. arXiv
> preprint arXiv:2207.06456, 2022.
>
> Lihong Li, Wei Chu, John Langford, and Robert E Schapire. A contextual-bandit approach to
> personalized news article recommendation. In WWW, pp. 661–670, 2010.
>
> Shuai Li, Wei Chen, Shuai Li, and Kwong-Sak Leung. Improved algorithm on online clustering of
> bandits. In IJCAI, pp. 2923–2929, 2019.
>
> Yunzhe Qi, Yikun Ban, and Jingrui He. Neural bandit with arm group graph. arXiv preprint
> arXiv:2206.03644, 2022.
>
> Michal Valko, Nathan Korda, R ́emi Munos, Ilias Flaounas, and Nello Cristianini. Finite-time anal-
> ysis of kernelised contextual bandits. In Uncertainty in Artificial Intelligence, 2013.
>
> Weitong Zhang, Dongruo Zhou, Lihong Li, and Quanquan Gu. Neural thompson sampling. In
> International Conference on Learning Representations, 2021.
>
> Dongruo Zhou, Lihong Li, and Quanquan Gu. Neural contextual bandits with ucb-based exploration.
> In International Conference on Machine Learning, pp. 11492–11502. PMLR, 2020.3

---

> > ### Comment · Reviewer_hbM8 · 2022-12-04
> > **Reply to the author**
> >
> > Thank you for providing the detailed response.
> >
> > First, I am still concerned about the “linear-like” curves of cumulative regrets shown in the experiments. I understand that this might be due to the modeling bias for the data sets, but I would like to see the convergence of the algorithm. I would suggest to try other measures such as cumulative reward or averaged reward to show the convergence and comparisons.
> >
> > Second, I understand it is common that the width $m$ needs to be $poly(T)$ in the over-parameterized neural network setting, but this might not match the interactive setting very well where we care more about the asymptotic rate on T. Perhaps it would be better to consider a pure exploration setting to identify the $\epsilon$-approximate best arm. Then the parameter $m$ might need to be $poly(1/\epsilon)$, which seems more natural.
> >
> > Third, I agree with the reviewer WB9s’s comments about the space and time complexity when the number of users $n$ is large. As mentioned in remark 3.2, the algorithm can adopt the “representative users” to approximately model the “fine-grained” user correlations. Perhaps a comparison with this approximation method would better illustrate the differences of performances and efficiencies.
> >
> > Based on these considerations, I would keep my score.

---

> > > ### Author Response · Authors · 2022-12-08
> > > **Further Discussion with Reviewer hbM8 (3)**
> > >
> > > ***(1) [Time complexity]***
> > >
> > > Suppose we have a total of $n$ users.
> > > In each round $t\in [T]$, we will need to iterate through user models and construct the user graphs. Originally, we will need $O(n)$ time to calculate the output of user models, and $O(n^{2})$ kernel evaluations on scalar values to calculate the exploitation/exploration scores for user pairs based on the user model output. The kernel evaluation of scalar values is fast, and the whole can be highly parallelized by using multiple machines where each machine is in charge of one / multiple users.
> > > After adopting $\widetilde{n}$ ($\widetilde{n} << n$) “representative users” instead to approximately model the “fine-grained” user correlations (Remark 3.2), we will then need $O(\widetilde{n})$ time to calculate the output of user models, and $O(\widetilde{n}^{2})$ time to calculate the exploitation/exploration scores for user pairs with kernel evaluations. In this way, the time complexity can be significantly reduced.
> > >
> > >
> > > ***(2) [Space complexity]***
> > >
> > > After adopting $\widetilde{n}$ “representative users”, our space complexity is no larger than $ 2T\cdot \widetilde{n}^{2}  + O(\widetilde{n}\cdot p\cdot T) + O(n\cdot p)$, and applying the approximation method in Remark 3.1 can further significantly reduce the space complexity. Here the first term $2T\cdot \widetilde{n}^{2}$ is the cost of storing past user graph adjacency matrices of selected arms, the second term is for storing past contexts and network gradients, and the third term is the cost of storing individual user models.
> > >
> > > To model user behaviors, each user is assigned with a separate neural single-bandit model with no user collaborations (Zhou et al., 2020; Qi et al., 2022; Zhang et al., 2021). For neural algorithms using the UCB (Zhou et al., 2020; Qi et al., 2022) and Thompson sampling (Zhang et al., 2021) to achieve exploration, they need the space complexity of $O(n\cdot p^{2})$ ($p$ is the number of parameters in the neural model, which is a considerably large number) to store their gigantic gradient matrices, which demands large amount of running memory especially when the neural model is large. EE-Net (Ban et al., 2022b) has the space complexity of $O(n\cdot p\cdot T)$.
> > >
> > >
> > >
> > >
> > > **REFERENCES**
> > >
> > > Yikun Ban, Yunzhe Qi, Tianxin Wei, and Jingrui He. Neural collaborative filtering bandits via meta
> > > learning. arXiv preprint arXiv:2201.13395, 2022a.
> > >
> > > Yikun Ban, Yuchen Yan, Arindam Banerjee, and Jingrui He. Ee-net: Exploitation-exploration neural
> > > networks in contextual bandits. In International Conference on Learning Representations, 2022b.
> > >
> > > Tanner Fiez, Lalit Jain, Kevin G Jamieson, and Lillian Ratliff. Sequential experimental design for
> > > transductive linear bandits. Advances in neural information processing systems, 32, 2019.
> > >
> > > Yunzhe Qi, Yikun Ban, and Jingrui He. Neural bandit with arm group graph. arXiv preprint
> > > arXiv:2206.03644, 2022.
> > >
> > > Michal Valko, Nathan Korda, R ́emi Munos, Ilias Flaounas, and Nello Cristianini. Finite-time analysis of kernelised contextual bandits. In Uncertainty in Artificial Intelligence, 2013.
> > >
> > > Weitong Zhang, Dongruo Zhou, Lihong Li, and Quanquan Gu. Neural thompson sampling. In
> > > International Conference on Learning Representations, 2021.
> > >
> > > Dongruo Zhou, Lihong Li, and Quanquan Gu. Neural contextual bandits with ucb-based exploration.
> > > In International Conference on Machine Learning, pp. 11492–11502. PMLR, 2020.
> > >
> > > Yinglun Zhu, Dongruo Zhou, Ruoxi Jiang, Quanquan Gu, Rebecca Willett, and Robert Nowak. Pure
> > > exploration in kernel and neural bandits. Advances in Neural Information Processing Systems, 34:
> > > 11618–11630, 2021.4

---

> > > ### Author Response · Authors · 2022-12-08
> > > **Further Discussion with Reviewer hbM8 (2)**
> > >
> > >
> > >
> > >
> > >
> > >
> > > ***(2) [Learning objective]***
> > >
> > > Under the pure exploration settings for contextual bandits, its task is to identify the approximately optimal arms with the minimum budget. Therefore, the performance of algorithms is usually measured by the sample complexity (i.e., after how many rounds, the algorithm will be able to output approximately optimal arms with a high probability).
> > >
> > > On the other hand, for the contextual bandits for user modeling, our objective is to minimize the cumulative regret $R(T)$ over $T$ rounds, and the cumulative regrets will be considered as the evaluation criterion.
> > > Therefore, the learning objectives of our user modeling settings and the pure exploration settings are distinct. For the user modeling in contextual bandits, the cumulative regret is consistently applied as the evaluation criterion since it measures the difference between the trained learner and the optimal policy after a finite time steps.
> > >
> > >
> > >
> > >
> > > ***(3) [Single / multiple bandit(s)]***
> > >
> > > In existing pure exploration works, the reward for each arm in the fixed arm pool is governed by one single mapping function, which can be linear (Fiez et al., 2019) or non-parametric (Zhu et al., 2021). This is analogous to the existing contextual bandits works under the single-bandit settings (e.g., Valko et al., 2013; Zhou et al., 2020; Ban et al., 2022b).
> > >
> > > However, for the existing user clustering works as well as this paper, the reward mapping function can be distinct for different users, which is more consistent with real-world applications. This is also one major difference between algorithms modeling user collaborations and the single-bandit algorithms. We have discussed this issue in our previous response, where we compare algorithms modeling user collaborations and algorithms with no collaboration.
> > >
> > >
> > >
> > > ***(4) [Network width $m$ of the same order]***
> > >
> > > Under the pure exploration settings, for state-of-the-art neural algorithms of identifying optimal arms, we also need network width $m \geq \text{Poly}(K, \epsilon^{-1})$ (e.g., Theorem 3 in Zhu et al. (2021)) where $K$ is the total number of arms in the fixed candidate pool and $\epsilon$ is the error parameter for defining the $\epsilon$-optimal arm. Note that under the recommendation settings, $K$ will be equal to the total number of items in the item pool (e.g., the total number of movies available across the whole platform), which is a gigantic number and can be much larger than the number of rounds $T$.
> > > This is because the width of over-parameterized networks needs to be significantly larger than the number of samples (i.e., size of the arm pool $K$).
> > > Meanwhile, since the analysis in this paper (Zhu et al. (2021)) relies on the NTK regime, the bound for sample complexity also has the effective dimension term $\widetilde{d}$ of the NTK gram matrix.
> > >
> > >
> > > Therefore, the pure exploration settings are distinct from our current problem of user modeling and it is beyond the scope of this paper. For those application scenarios with a fixed arm pool, it can be valuable to combine the pure exploration settings with the user collaborations. This can be considered as one possible future direction.
> > >
> > >
> > >
> > >
> > >
> > >
> > > **Further discussion: (3) More discussions for Remark 3.2?**
> > >
> > >
> > > We would like to clarify that we have included the experiments for the approximated neighborhood to Subsection B.5 in the manuscript appendix, which is currently available on the OpenReview platform. In addition, with the number of users $n=500$ on the MovieLens data set, we include the experiments given different numbers of “representative users” $\widetilde{n}$ to better show the performance of approximated neighborhoods.
> > > Here, increasing the number of “representative users” $n$ can lead to a better performance of GNB, and it also shows that a small number of “representative users” can indeed enable GNB to achieve good performances.
> > >
> > > We will add these experimental results to the manuscript.
> > >
> > >
> > >
> > >
> > >
> > >
> > > ***Table 2: With the number of users $n=500$ for the MovieLens data set, the comparison between GNB and baselines on average regret per round.***
> > >
> > > | (Method \ Time step) | 2000  | 4000 | 6000 | 8000 | 10000 |
> > > |---  |--- |--- |--- |--- |--- |
> > > | CLUB   | 0.7691 | 0.7513 | 0.7464 | 0.7468 | 0.7496
> > > | Neural-UCB-Ind  |   0.8901 |  0.8808 |  0.8790  |  0.8754 | 0.8741
> > > | Neural-UCB-Pool   | 0.7681 | 0.7526 | 0.7405 | 0.7362 | 0.7334
> > > | EE-Net  |   0.7886 | 0.7723 | 0.7642 | 0.7618 | 0.7582
> > > | Meta-Ban  |  0.7811 | 0.7761 | 0.7754 | 0.7729 | 0.7708
> > > | GNB ($\widetilde{n}=50$)  |  0.7760 | 0.7245 | 0.7190 | 0.7265 | 0.7140
> > > | GNB ($\widetilde{n}=100$)   | 0.7406 | 0.7178 | 0.7172 | 0.7110 | 0.7104
> > > | GNB ($\widetilde{n}=150$)  |  0.7291 | 0.7228 | 0.7129 | 0.7105 | 0.7085

---

> > > ### Author Response · Authors · 2022-12-08
> > > **Further Discussion with Reviewer hbM8 (1)**
> > >
> > > We sincerely appreciate the discussion with the reviewer and would like to thank you again for your questions and comments.
> > > Here, we would like to further clarify the remaining of your concerns.
> > >
> > > **Further discussion Q1: Convergence of GNN models, and the change of average cumulative regrets?**
> > >
> > > Thank you for the suggestion.
> > > Here, for each separate time step interval of 2000 rounds, we show the average cumulative regrets results on MovieLens and Yelp data sets within this interval. We also include another three neural baselines (EE-Net (Ban et al., 2022b), Neural-UCB (Zhou et al., 2020) and Meta-Ban (Ban et al., 2022a)) for comparison.
> > >
> > >
> > > **Table 1: In different time step intervals, the average cumulative regrets per round (with the standard deviation) on MovieLens and Yelp data sets.**
> > >
> > > | (Dataset (Algorithm) \ Time step intervals) | 0-2k  | 2k-4k | 4k-6k | 6k-8k | 8k-10k |
> > > |---  |--- |--- |--- |--- |--- |
> > > | ***MovieLens (GNB)***   | 0.7050 |  0.6895 | 0.6795  |  0.6770 | 0.6685
> > > | ***Yelp  (GNB)***  |   0.7705 | 0.7685 | 0.7485 | 0.7450 | 0.7330
> > > | MovieLens (EE-Net)    | 0.7320 | 0.7120 | 0.7072 | 0.6945 | 0.7070
> > > | Yelp  (EE-Net)  |   0.8480 | 0.8420 | 0.7965 | 0.7950 | 0.7865
> > > | MovieLens (Neural-UCB)    | 0.8090 | 0.7115 | 0.7165 | 0.6945 | 0.6865
> > > |  Yelp  (Neural-UCB)  |   0.8380 | 0.8115 | 0.8185 | 0.7940 | 0.8025
> > > | MovieLens (Meta-Ban)    | 0.7760 | 0.7245 | 0.7190 | 0.7265 | 0.7140
> > > |  Yelp  (Meta-Ban)  |   0.8525| 0.8035 | 0.7930 | 0.7600 | 0.7620
> > >
> > >
> > >
> > >
> > > As we can see from the table, the average regret per round of GNB is decreasing, although the decreasing rate is small. Meanwhile, compared with baselines, GNB manages to achieve the best prediction accuracy across different time step intervals by modeling the fine-grained user correlations and applying the adaptive exploration strategy.
> > >
> > >
> > >
> > > We will add more discussion about the model convergence issue to the manuscript.
> > >
> > >
> > >
> > >
> > >
> > >
> > >
> > >
> > >
> > > **Further discussion Q2: Adapting to the pure exploration settings?**
> > >
> > > Thank you for your suggestion. Here, we would like to clarify several intrinsic differences between our settings and the pure exploration settings. Meanwhile, we also note that for neural algorithms identifying optimal arms under the pure exploration settings, they will also need the network width $m \geq \text{Poly}(K, \epsilon^{-1})$ (Zhu et al., 2021), where $K$ is the total number of arms in the fixed candidate pool (e.g., the total number of movies available across the whole movie recommendation platform, which is a considerably large number), and $\epsilon$ is the error parameter for defining the $\epsilon$-optimal arm.
> > >
> > > ***(1) [Fixed / dynamic arm pool]***
> > >
> > > For the optimal arm identification task under the pure exploration settings, we need to assume that the arm pool is fixed (Fiez et al., 2019; Zhu et al., 2021). Here, if we adapt this setting to the user modeling scenario such as the movie recommendation, we will end up with a fixed arm (movie) pool.
> > > In real-world application scenario, the arm pool is always dynamic, since we will have new items continuously showing up while the obsolete items (e.g., expired goods) will be deleted from the arm pool.
> > > Moreover, the interactions of one particular user-item are usually scarce, which is also not consistent with the pure exploration setting.
> > > Taking this into consideration, both the existing user clustering algorithms and our proposed GNB have no assumption for the arm pool, which is distinct from the pure exploration settings.

---

> ### Author Response · Authors · 2022-11-15
> **To Second Reviewer, hbM8 (3)**
>
>
>
> **Q: Comparison between algorithms modeling user collaborations and algorithms with no collaboration?**
>
>
>
> Our proposed GNB framework differs from the linear user clustering algorithms from the following major aspects:
>
> (1) Regarding the definition of the reward mapping function $h(\cdot)$: For algorithms with no collaboration, e.g., Zhou et al., 2020; Ban et al., 2022b, they consider the reward mapping function to be a direct mapping from the arm $x$ to the expected reward $E[r]$, i.e., $E[r] = h(x)$. Therefore, with no user collaborations, all the users will be treated the same way given an item since they are sharing the identical reward mapping function, which is not realistic in real-world cases.
>
> On the other hand, given an user $u_{t}$, existing user clustering works with “coarse-grained” user collaborations, e.g., (Ban et al., 2022a), assume $E[r] = h_{N_{t}}(x)$ where $u_{t}$ belongs to the user cluster $N_{t}$ ($u_{t} \in N_{t}$). In this case, all the users within the same user cluster $u\in N_{t}$ are treated the same, which is too strong for real applications as users within the same user group (e.g., friends) may have different preferences.
> Moreover, by their definition, users across different groups will have no collaborations, which is also too strong in the real-world cases because two users from different preference groups can also be mutually influenced. In this case, modeling “coarse-grained” user correlations will lead to extra estimation error.
>
> Finally, our formulation of the “fine-grained” user collaborations considers a much more generic problem definition as
> $E[r] = h(x, u_{t}, \Lambda)$ where $\Lambda$ is the user affinity matrix induced by arm $x$. In this way, our work is able to model the pair-wise user correlation instead of imposing rigid user groups, which is more reasonable especially when concrete user groups do not even exist.
> In particular, our formulation can be easily generalized to existing problem settings. For instance, for the user clustering scenario, we can consider the affinity matrix $\Lambda$ to be a block matrix where each block corresponds to a single user group.
>
> In the experiments, linear algorithms that model the user collaborations will occasionally outperform the neural single-bandit algorithms (e.g., in the “Yelp” data set) despite the huge representation power gap between the linear models and neural models. This phenomenon provides the clear evidence that modeling user correlations is necessary in real-world application scenarios.
>
>
> (2) The theoretical analysis of single-bandit algorithms, i.e., the algorithms with no user collaborations, will not involve the term $n$ (the number of users) in their regret bound, because their strong assumption of the reward mapping function considers all the users are sharing the same preference.
> On the other hand, if we assign each user with his/her own single-bandit algorithm, the final regret bound will also include a $\sqrt{n}$ term (Li et al., 2010).
>
> Analogously, existing user clustering works need to assign each user group a separate bandit model. This will naturally lead to a $\sqrt{n}$ term for the regret bound (Gentile et al., 2014; Li et al., 2019; Gentile et al., 2017; Ban & He, 2021; Ban et al., 2022a).
>
> On the other hand, we apply two separate GNN models for exploitation and exploration individually. With the help of framework design and the carefully designed theoretical analysis based on martingale difference sequences and over-parameterized neural networks, we will only include a $\sqrt{\log(n)}$ term in this regret bound, which is sharper than existing works.

---

> ### Author Response · Authors · 2022-11-15
> **To Second Reviewer, hbM8 (2)**
>
> **Q: Comparison between linear algorithms and neural algorithms?**
>
> Apart from proposing to model the “fine-grained user correlations”, our proposed GNB framework differ from the linear user clustering algorithms from the following major aspects.
>
> (1) For linear algorithms, they all assume that the expected reward is generated by a linear reward mapping function, i.e., $E[r] = \langle x, \theta \rangle$ where $\theta$ is the unknown parameter. This assumption is strong since the reward mapping is non-linear in many cases under the real-world application scenario (Valko et al., 2013). On the other hand, our GNB and other neural algorithms, consider the reward mapping function to be non-parametric, which means that the reward mapping function can be either linear or non-linear. The power of neural algorithms is also supported by our experiments, where the neural methods generally tend to outperform linear methods.
>
> (2) Regarding the theoretical analysis, since the linear user clustering algorithms are modeling the “coarse-grained” user correlations, they will need to use separate linear models for each user group, which will inevitably lead to a $\sqrt{n}$ ($n$ is the number of users) term in their regret bound. Our proposed GNB manages to reduce the $\sqrt{n}$ term to $\sqrt{\log(n)}$ by using two GNN models. This improvement is nontrivial and has not been achieved by existing works. Meanwhile, since linear algorithms tend to apply regression-based analysis, their regret bound will naturally include a term $d$ (the dimensionality of the arm context $x$). The theoretical analysis of GNB is based on martingale difference sequences and the generalization results of over-parameterized neural networks, which are free of this term $d$.

---

> ### Author Response · Authors · 2022-11-15
> **To Second Reviewer, hbM8 (1)**
>
> We sincerely thank the reviewer for the detailed comments and suggestions, and we will address the questions and concerns in the form of Q\&A.  Please also refer to our updated “Rebuttal Revision” manuscript for updates.
>
>
> **Q: Removing input dimension $d$ in the theoretical analysis?**
>
> Recall that $d$ stands for the dimension of arm features. As we have discussed in the paper, the generalization ability and the convergence rate of over-parameterized networks are not affected by the dimension of the input data $d$ (Allen-Zhu et al., 2019; Cao & Gu, 2019).
> This is a major difference between linear algorithms, e.g., Gentile et al., 2014; Ban & He, 2021, and the neural bandit algorithms. Meanwhile, compared with existing works with NTK regression-based theoretical analysis, e.g., Zhou et al., 2020; Zhang et al., 2021; Qi et al., 2022; Kassraie et al., 2022, our theoretical analysis based on martingale difference sequences also enables us to remove the effective dimension $\tilde{d}$ (Lemma F.2, F.4, G.2, G.3), which can be as large as the number of rounds $T$ in the worst-case scenario.
> Therefore, our final regret bound is free of the terms $d$ and $\tilde{d}$.
>
>
>
>
>
> **Q: Large network width $m$?**
>
> Under the over-parameterized neural network settings, the network width $m$ needs to be as large as the polynomial order of the number of samples (in the bandit case, the number of samples depends on $T$) (Allen-Zhu et al., 2019; Cao & Gu, 2019). In this way, the neural network will be generic enough to properly approximate the non-parametric reward mapping function $h(\cdot)$.
> This is also a standard assumption for neural bandits (e.g., Zhou et al., 2020; Qi et al., 2022; Ban et al., 2021; Zhang et al., 2021; Ban et al., 2022a; Kassraie et al., 2022; Ban et al., 2022b)
>
>
>
>
>
> **Q: Explanation about the user exploration graph?**
>
> As we discussed in the paper, to achieve the adaptive exploration by taking the user collaborative effects into consideration, the user exploration graph $G_{i, t}^{(2)}$ aims to model the user collaborative effects from the exploration perspective in terms of the given arm $x_{i, t} \in \mathcal{X_{t}}$. This is different from conventional UCB-based / Thompson-Sampling-based exploration strategies, where the user collaborative effects are not considered.
>
> Intuitively, in the exploration graph $G_{i, t}^{(2)}$, when the edge weight $w_{i, t}^{(2)}(u, u')$ between two users $u, u'$ is high, we may want to apply similar exploration strategies for both users $u, u'$.
> For instance, given arm $x_{i, t}$, if the prediction uncertainty (of their user models) is large for both $u$ and $u'$, we will need to explore both of these two user-arm pairs $(u, x_{i, t}), (u', x_{i, t})$ more for additional knowledge.
>
>
>
>
>
> **Q: Model convergence for the recommendation data sets?**
>
> During our experiments, we have noticed that the prediction accuracy (i.e., the accuracy of pulling the optimal arm) is consistently increasing, but the increasing rate is relatively low, which leads to the “linear-like” curves of cumulative regrets.
>
> One possible reason is that these two recommendation data sets contain considerable inherent noise, which makes the algorithm hard to learn the correct reward mapping function. In this case, the experimental improvement on these two data sets is non-trivial. As in the existing works, the baselines as well as their proposed algorithms will behave similarly, and the performance difference is small.
> For instance, in existing works (Zhou et al., 2020; Zhang et al., 2021; Ban et al., 2022a; Qi et al.,
> 2022; Ban et al., 2022b), we can all observe this kind of “linear-like” cumulative regret curves for the recommendation data sets.

---

> ### Author Response · Authors · 2022-12-01
> **Looking forward to more discussions**
>
> Dear Reviewer hbM8,
>
> We really appreciate your detailed comments and questions. As it has been a while since we replied to your questions, we are wondering if you could kindly let us know whether our response has properly answered your questions. If you have any further questions or concerns, please kindly let us know and we will try our best to address them.
>
> Thanks again for your time and the discussion.
>
> Sincerely,
>
> Authors

---

### Official Review · Reviewer_ENq3 · 2022-10-27

**Confidence:** 4
**Correctness:** 3
**Technical Novelty And Significance:** 3
**Empirical Novelty And Significance:** 2
**Recommendation:** 6

**Clarity, Quality, Novelty And Reproducibility:**

The presented approach is novel.
The paper requires some rethinking of the structure.
The authors did not mention anything about the availability of the code corresponding to the experiments they run.

**Strength And Weaknesses:**

The topic and the methodology proposed are for sure of value to the ICLR community. I have concerns mainly about two points.
1) The presentation of the paper is poor. Before considering it for publication, I think that the authors should revise the structure of the paper and to better present the elements composing the proposed approach.
2) The paper is not self-contained. Indeed, while it is clear that the proofs cannot be included in the main paper, most of the details about the experimental setting are left in the appendix. This also holds for some parts of the description of the methodology (pseudocode). My concern is that the paper requires the appendix to be fully understood.

Moreover, I think that the authors should provide statistical significance for the experiments they provided.

Details:
- please add a formal definition of what you call volume (before Equation 1)
- check the punctuation of the formulas
- please add a more complete definition of the \Lambda^*_i,t matrix. For instance, is it changing over time?
- The presentation of the user graphs is not clear. In my opinion, you should have presented a general definition for a graph and subsequently instantiate it for your exploitation and exploration graphs. I suggest rethinking this paragraph.
- Is the function \Psi^{(1)} known? It is mentioned in the following part of the paper, but it should require a discussion just after its definition. The same holds with \Psi^{(2)}.
- there are minor grammar errors (in -> at, missing 'the')
- Overall, the presentation of the problem formulation can be improved. I think that all the elements required for further discussion are present, but the order and the comments provided may be unclear to a generic ML researcher who is reading the paper, while an expert of the bandit setting is able to understand even if they are presented poorly.
- Remark 3.1: do we also have some bounds on the loss we might incur in terms of accuracy or the bias we are introducing with this approach?
- Remark 3.2: even in this case, it might be interesting to understand if there is a factor in terms of regret implied by this choice.
- I think that deferring the details about the algorithm to the appendix would make the paper non-self-contained. Maybe the complexity of the description of the proposed algorithm requires more space to include all the material, therefore, I suggest evaluating the possibility of sending the paper to a venue where there is no constraint on the page length.
- It seems that the experiments are provided without confidence intervals/standard deviation bounds. I think that the significance of the results can be assessed only if they are explicitly added to the figures.
- The details about the experimental setting are provided in the appendix. I think that the authors should also distribute the code corresponding to their method.

------------------------------------------------------------------------------
After rebuttal: I think that the paper is of interest but should be accepted only if the authors are able to make it self-contained. I raised my score, assuming that the authors can implement all the improvements cited in the rebuttal.

After the second rebuttal response: I think that the quality of the paper presentation has been improved during the discussion. I raised further my score.

**Summary Of The Paper:**

The authors present a novel graph neural bandit approach to tackle the problem of estimating the reward corresponding to the selection of options for users based on their contextual information. Differently from previous works, they adopt a graph structure among users to exploit the correlations existing among them. The authors propose and analyse a new algorithm theoretically. Finally, they apply their methods to real-world settings and show the superior performance of what has been presented w.r.t. state-of-the-art algorithms.

**Summary Of The Review:**

The topic is interesting, and the developed approach is novel, but some concerns about the presentation and self-completeness of the main paper make it not yet ready for publication.

---

> ### Author Response · Authors · 2022-11-15
> **To First Reviewer, ENq3**
>
> We sincerely thank the reviewer for the detailed comments and suggestions, and we will address the questions and concerns in the form of Q\&A.  Based on your suggestion regarding the presentation issues, we have updated the “Problem Definition” and the “Experiments” section to include more details. Please refer to our updated “Rebuttal Revision” manuscript for revisions and updates.
>
>
>
> **Q: Why not formulate the user graphs first, and use the user graphs to define the reward mapping function?**
>
> Thanks for your helpful suggestion.
> We have updated the problem formulation by including the user graphs into the reward mapping function definition. Please refer to the updated manuscript.
>
>
>
> **Q: Definition of the $\Lambda_{i, t}^{*}$ matrix?**
>
> As we discussed in the paper, the true correlation matrix $\Lambda_{i, t}^{*}$ (in the updated version, we use the true user graph $G_{i, t}^{(1), *}$ instead) only depends on the given arm $x_{i, t} \in \mathcal{X}_{t}$.
> Therefore, the unknown true user correlations w.r.t. the arms are independent from the change of time. This is a standard argument used in closely related works (e.g., (Gentile et al., 2017; Ban et al., 2022)).
>
>
>
> **Q: Deferred details about the algorithm?**
>
> We would like to clarify that although the pseudo code for arm graph construction (Algorithm 2) and the model training (Algorithm 3) is presented in the appendix, we have included all the detailed descriptions for the proposed framework, such as text descriptions, equations, and the illustration (Figure 1) in the main body.  Similar arrangements can also be found in some existing works, e.g., (Kassraie et al., 2022).
> Therefore, the main body itself is self-contained as the details of the framework, including the problem definition and the methodology, have been clearly stated before the appendix. The goal of providing the pseudo code in the appendix is to help the readers easily implement our proposed framework.
> This arrangement is also acknowledged by the other three reviewers.
>
>
> **Q: Theoretical analysis for Remark 3.1, 3.2?**
>
> Thanks for your suggestion.
> In this paper, we consider Remarks 3.1 and 3.2 as two heuristic approaches to practically expedite the running speed of GNB and scale GNB to real-world application scenarios.
>
> In this paper, we focus on deriving the theoretical regret bound for our proposed GNB framework, which is sharper than existing works.
> Since the theoretical analysis for these two heuristic methods will require considerably more additional assumptions, which are beyond the scope of this paper, we choose to prove their effectiveness through experiments.
>
>
>
>
>
>
> **Q: Statistical significance of the experiments?**
>
> Thanks for the suggestion.
> We have included the standard deviation of the cumulative regret results for GNB on real data sets.
>
>
>
>
>
> **Q: Availability of code?**
>
> We would like to clarify that our code has been included in the “supplementary materials” section along with our initial submission.
>
>
>
>
>
> **REFERENCES**
>
> Yikun Ban, Yunzhe Qi, Tianxin Wei, and Jingrui He. Neural collaborative filtering bandits via meta
> learning. arXiv preprint arXiv:2201.13395, 2022.
>
> Claudio Gentile, Shuai Li, Purushottam Kar, Alexandros Karatzoglou, Giovanni Zappella, and
> Evans Etrue. On context-dependent clustering of bandits. In ICML, pp. 1253–1262, 2017.
>
> Parnian Kassraie, Andreas Krause, and Ilija Bogunovic. Graph neural network bandits. arXiv
> preprint arXiv:2207.06456, 2022

---

> ### Author Response · Authors · 2022-11-26
> **Second-round Reply to First Reviewer: Improving paper presentation and making it more self-contained (2)**
>
>
> **(4) [Theoretical Analysis (Section 4)]**
>
> In this section, we first give the prerequisites for the theoretical analysis, including the background settings as well as the assumptions. After showing the main theorem of the theoretical regret bound (Theorem 4.2), which includes the details about the network width, learning rate and the corresponding regret bound, we proceed to compare our derived regret bound with those of existing works. Our regret bound as well as our theoretical improvements are then summarized by the following two remarks (Remark 4.3, 4.4). ***Apart from the improvements, we have also added more details and intuitions on how we derive the regret bound for the proposed GNB framework to these two remarks, so that the readers will have a more comprehensive understanding about our theoretical analysis.***
> Interested readers can then go to the appendix for the details of the regret bound derivation.
>
>
> **(5) [Experiments (Section 5)]**
>
> ***To offer the reader with more details of the experiments, we have moved the data set descriptions as well as the implementation details, including the setting of GNB and baselines, back to the main body. To achieve this, we have properly reshape the figures to reduce the page space occupied and make them both clear and concise. Meanwhile, we also move the lengthy URLs of the data sets to the appendix, since these URLs are for the reproduction purposes and will not affect the conclusions for our experiments.***
> Afterwards, we include detailed discussion for the experiment results on the four data sets, including the findings, insights and the comparison with the baselines. Apart from the experiments on the four real data sets, we have also included brief descriptions for the complementary experiments (e.g., the parameter study) located in the appendix. In addition, each subsection of the complementary experiments is accompanied with one-sentence summary, so that readers can rapidly learn the insights of these complementary experiments, and go to the appendix for details if they are interested.
>
> **(6) [Conclusion (Section 6)]**
>
> Finally, we conclude the paper in Section 6. Here, we summarize the motivations as well as the main contributions of the paper.
>
>
>
> **We will continue working on the manuscript and try our best to improve its presentation quality. If you have any further comments or suggestions, please kindly let us know.**

---

> ### Author Response · Authors · 2022-11-26
> **Second-round Reply to First Reviewer: Improving paper presentation and making it more self-contained (1)**
>
> We sincerely appreciate your constructive comments and suggestions on improving the presentation and the structure of the paper. To better address the remaining of your concerns, we have been continuously working on improving the paper presentation, and we would like to briefly introduce the current paper structure and contents. Here, additional improvements have been included to make the manuscript more self-contained, compared with the version that is now available on the "OpenReview" platform.
>
> **(1) [Introduction (Section 1)]**
>
> We introduce the motivation of this work and the background information for the problem we trying to solve. ***Meanwhile, we have also added more discussion regarding the contribution of this paper, as well as the difference between this paper and the existing works in terms of problem formulation, proposed method, theoretical analysis and the experimental results.***
>
> **(2) [Problem Definition (Section 2)]**
>
> We first introduce our new problem definition followed by the detailed comparison between our problem settings with existing works. Then, we propose two kinds of user graphs to model the user collaborations in terms of exploitation / exploration individually, and also give the formal definition of these two kinds of user graphs as well as the definition for edge weights, which measure the magnitude of the user correlations.
> ***Here, instead of using the user affinity matrix to define the reward mapping function, we have changed the reward definition with the user exploitation graph based on your suggestion.***
> After that, we also describe the learning objective, i.e., minimizing the cumulative pseudo-regret, and list the notation used in this paper.
>
> **(3) [Proposed Framework (Section 3)]**
>
> We describe the details of our proposed GNB framework. Here, we begin with the user models with which we estimate the two kinds of user graphs. Then, we proceed to introduce the two GNN models, the exploitation GNN model and the exploration GNN model. For the user models and the GNN models, we apply text descriptions and equations to show their architecture, inputs and outputs in detail. Meanwhile,
>
> ***To improve the paper presentation and make it more self-contained, we adjust the pseudo-code style to accommodate more contents. Here, we include the pseudo-code of constructing the user graphs (i.e., the pseudo-code in Algorithm 2 originally) to Algorithm 1 in the main body. Therefore, the current pseudo-code in Algorithm 1 have covered both the estimation of user graphs and the reward / potential estimation process.***
>
> ***On the other hand, since the model training with gradient descent (originally corresponds pseudo-code in Algorithm 3) is the standard knowledge, which is not the main contribution of this work, we choose to describe the training process with text descriptions as well as equations (e.g., the input data and corresponding labels for the gradient descent optimization) for each module in their own subsections. Thus, although the pseudo-code of model training (Algorithm 3, originally) is left to the appendix for reproduction purposes, readers can still be able to get sufficient knowledge of the model training process based on the descriptions in the main body.***
>
>
> Finally, to reduce the computational cost of GNB in practice, we propose two heuristic methods to reduce the computational cost of GNB. Their effectiveness is be supported by the corresponding experiments.

---

### Decision · Program_Chairs · 2023-01-20

**Decision:**

Reject

**Justification For Why Not Higher Score:**

There are multi-round extensive discussions between authors and reviewers. The overall rating increased from 4.25 to 5.75 during the discussion period. There are some remaining concerns such as 1. space and time complexity are high, which may prevent the algorithm to be applied in practice; 2. the 'linear-like' regret curves are hard to explain; 3. the design and advantage of k-hop for information propagation are unclear. Others concerns are mostly resolved and reviewers increased scores accordingly.

**Justification For Why Not Lower Score:**

N/A

**Metareview: Summary, Strengths And Weaknesses:**

The authors present a novel Graph Neural Bandit (GNB) method to model the fine-grained user collaborative effects where the user correlation is formulated on the user graph.  A GNN-based bandit algorithm is proposed to achieve adaptive exploration. The authors proved an $O(\sqrt{T\log(Tn}) $ regret bound of the proposed algorithm. Experiments on real-world data showed its efficiency compared with state-of-the-art algorithms.

There are multi-round extensive discussions between authors and reviewers. The reviewers all agree that the setting is interesting and proposed GNN-based bandit algorithm is novel. The reviewers raised several concerns.  While some concerns are resolved and reviewers increased scores for that, the following concerns are remaining: 1. space and time complexity are high, which may prevent the algorithm to be applied in practice; 2. the 'linear-like' regret curves are hard to explain; 3. the design and advantage of k-hop for information propagation are unclear. The authors are encouraged to improve the paper according.